# One-Pass Feature Evolvable Learning with Theoretical Guarantees

Cun-Yuan Xing [*1]   Meng-Zhang Qian [*1]   Wu-Yang Chen [1]   Wei Gao [1]   Zhi-Hua Zhou [1]

## Abstract

Feature evolvable learning studies the scenario where old features will vanish and new features will emerge when learning with data streams, and various methods have been developed by utilizing some useful relationships from old features to new features, rather than re-training from scratch. In this work, we focus on two fundamental problems: How to characterize the relationships between two different feature spaces, and how to exploit those relationships for feature evolvable learning. We introduce the Kernel Ortho-Mapping (KOM) discrepancy to characterize relationships between two different feature spaces via kernel functions, and correlate with the optimal classifiers learned from different feature spaces. Based on this discrepancy, we develop the one-pass algorithm for feature evolvable learning, which requires going through all instances only once without storing the entire or partial training data. Our basic idea is to take online kernel learning with the random Fourier features and incorporate some feature and label relationships via the KOM discrepancy for feature evolvable learning. We finally validate the effectiveness of our proposed method both theoretically and empirically.

## 1. Introduction

Conventional machine learning generally works with the assumption that the data comes from a fixed feature space (Valiant, 1984; Shalev-Shwartz & Ben-David, 2014; Goodfellow et al., 2016; Mohri et al., 2018; Alpaydin, 2021). In some real-world applications, however, we may face more open scenarios; for example, we deploy sensors to collect data in an environmental monitoring task, and each sensor corresponds to a feature. Due to finite lifespan of sensors,

we need to deploy new sensors since old sensors will wear out, i.e., features corresponding to old sensors vanish but features corresponding to new sensors emerge.

Feature evolvable learning has been proposed to study the scenarios, where old features will vanish and new features will emerge when learning with streaming data. Recent years have witnessed increasing attention on this direction (Zhang et al., 2016; Hou et al., 2019; Beyazit et al., 2019; Gu et al., 2022; Hou et al., 2023; Schreckenberger et al., 2023; Chen & Liu, 2024). For feature evolvable learning, it is crucial to update model adaptively to accommodate new feature space but retain information of old feature space.

Various methods have been developed for feature evolvable learning by utilizing useful relationships from old features to new features, rather than re-training from scratch (Hou & Zhou, 2018; Zhang et al., 2020; Dong et al., 2022; Hou et al., 2022; Lian et al., 2023; Sajedi & Razzazi, 2024). This is helpful to prevent unnecessary wastes of computational resources and useful information from previous models over old features, and sometimes we may not collect sufficient training data to learn a stable model over new feature space.

Despite successes on the designs of practical algorithms, there are still some fundamental problems unsolved for feature evolvable learning. For example, how to present a formalization on the relationship characterization between different feature spaces, as well as correlations with model performance. Another problem is how to utilize useful relationships and information to improve the performance for feature evolvable learning from a theoretical view.

This work focuses on the one-pass algorithm for feature evolvable learning with theoretical guarantees, and the main contributions can be summarized as follows:

- We introduce the Kernel Ortho-Mapping (KOM) discrepancy to characterize the relationships between two different feature spaces via kernel functions, which essentially reflects kernels' gap under the rotational invariance. We compare our KOM discrepancy with prior characterizations such as kernel alignment and $\ell_2$ distance (Cristianini et al., 2001; Romero et al., 2015).

- Based on the KOM discrepancy, we develop the one-pass algorithm for feature evolvable data streams, which requires going through all instances only once

---

[*]Equal contribution  [1]National Key Laboratory for Novel Software Technology, Nanjing University, China; School of Artificial Intelligence, Nanjing University, China. Correspondence to: Wei Gao <gaow@nju.edu.cn>.

*Proceedings of the 42nd International Conference on Machine Learning*, Vancouver, Canada. PMLR 267, 2025. Copyright 2025 by the author(s).

without storing the entire or partial training data. Our basic idea is to take online kernel learning with the random Fourier features and incorporate feature and label relationships via the KOM discrepancy[1].

- Theoretically, we establish the intrinsic relationship between the KOM discrepancy and optimal classifiers learned from different feature spaces, and present the convergence analysis to show better regret bounds via some good model initializations and relationships from old feature space and models.

- We finally conduct extensive experiments to validate the effectiveness of our OPFES method in comparison with the state-of-the-art methods on feature evolvable learning, i.e., our method achieves better performance and the fastest convergence simultaneously.

The rest of this work is constructed as follows: Section 2 presents some preliminaries. Section 3 characterizes the relationship between two different feature spaces. Section 4 develops the OPFES method. Section 5 conducts extensive experiments. Section 6 concludes with future work.

## 2. Preliminaries

**Feature evolvable learning** studies evolvable feature spaces over time, where old features will vanish and new features will emerge. Let $\mathcal{X}^{[1]} \subseteq \mathbb{R}^{d^{[1]}}$ and $\mathcal{X}^{[2]} \subseteq \mathbb{R}^{d^{[2]}}$ be the old and new feature spaces, respectively. Feature evolvable learning includes three stages as follows:

- Previous stage: receive instances $\boldsymbol{x}_t^{[1]}$ from the old space $\mathcal{X}^{[1]}$ for $t = 1, \cdots, T_1$;

- Evolving stage: receive instances $\boldsymbol{x}_t^{[1]}$ and $\boldsymbol{x}_t^{[2]}$ from the old space $\mathcal{X}^{[1]}$ and new space $\mathcal{X}^{[2]}$, respectively, for $t = T_1 + 1, \cdots, T_1 + T_e$ with small positive $T_e$;

- Current stage: receive instances from the new space $\mathcal{X}^{[2]}$ for $t = T_1 + T_e + 1, \cdots, T_1 + T_e + T_2$.

Figure 1 presents an illustration of single feature evolvable learning (Hou et al., 2017; 2022; Lian et al., 2023), and we could make a similar analysis for multiple cases.

Let $\mathcal{K}(\cdot, \cdot)$ be a positive-definite and symmetric kernel with a mapping $\boldsymbol{\varphi} : \mathcal{X} \to \mathbb{H}$ from a feature space $\mathcal{X}$ to an RKHS $\mathbb{H}$. This work focuses on the shift-invariant kernels $\mathcal{K}(\boldsymbol{x}_1, \boldsymbol{x}_2) = \kappa(\boldsymbol{x}_1 - \boldsymbol{x}_2)$, such as Gaussian kernel and Laplacian kernel (Schölkopf & Smola, 2002).

**Online kernel learning** trains classifiers $h_1, \cdots, h_T \in \mathbb{H}$ from a streaming sample $S_T = \{(\boldsymbol{x}_1, y_1), \cdots, (\boldsymbol{x}_T, y_T)\}$

[1]The code is available at github.com/WeltXing/opfes

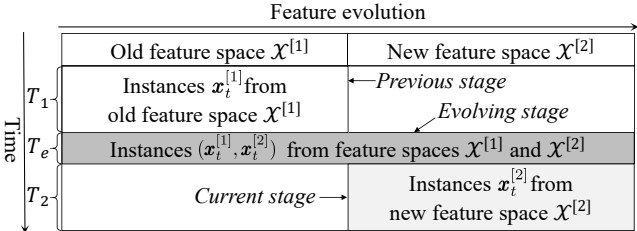

Figure 1. An illustration of feature evolvable stream.

with $y_i \in \{-1, +1\}$, by minimizing the following loss

$$h_t \in \arg\min_{h \in \mathbb{H}} \left\{ \frac{1}{t} \sum_{i=1}^{t} \ell(h, (\boldsymbol{x}_i, y_i)) + \frac{\lambda}{2} \|h\|_{\mathbb{H}}^2 \right\} ,$$

where $\ell(h_t, (\boldsymbol{x}, y)) = \max\{0, 1 - y h_t(\boldsymbol{x})\}$. Based on the representer theorem (Schölkopf & Smola, 2002), we have

$$h_t(\boldsymbol{x}) = \sum_{i=1}^{t} \alpha_i \mathcal{K}(\boldsymbol{x}, \boldsymbol{x}_i) \text{ and } \|h_t\|_{\mathbb{H}}^2 = \sum_{i,j=1}^{t} \alpha_i \alpha_j \mathcal{K}(\boldsymbol{x}_i, \boldsymbol{x}_j).$$

This shows that online kernel learning requires storing the entire training sample $S_T$, which makes it difficult for the large-scale datasets (Shen et al., 2019; Hong & Chae, 2021).

**Online kernel learning with random Fourier features** has been an efficient way for large-scale online kernel learning (Rahimi & Recht, 2007; Lu et al., 2016), which approximates high (or infinite) dimensional mapping $\boldsymbol{\varphi}(\cdot)$ with the finite-dimensional random Fourier features. Specifically, we approximate the kernel function of $\mathcal{K}$ as

$$\mathcal{K}(\boldsymbol{x}_i, \boldsymbol{x}_j) = \langle \boldsymbol{\varphi}(\boldsymbol{x}_i), \boldsymbol{\varphi}(\boldsymbol{x}_j) \rangle$$
$$\approx \sum_{k=1}^{d} p(\boldsymbol{u}_k) \phi(\boldsymbol{x}_i, \boldsymbol{u}_k, b_k) \phi(\boldsymbol{x}_j, \boldsymbol{u}_k, b_k) , \quad (1)$$

where $\phi(\boldsymbol{x}, \boldsymbol{u}, b) = \sqrt{2} \cos(\langle \boldsymbol{x}, \boldsymbol{u} \rangle + b)$, and $p(\cdot)$ is the spectral density function of $\mathcal{K}$. Here, random vectors $\boldsymbol{u}_k$ are sampled i.i.d. from standard normal distribution $\mathcal{N}(\boldsymbol{0}, \mathbf{I}_d)$, and $b_k$ are randomly selected independently and uniformly over $[0, 2\pi]$. For shift-invariant kernel $\mathcal{K}(\boldsymbol{x}_1, \boldsymbol{x}_2)$, we have

$$p(\boldsymbol{u}) = \int_{\mathbb{R}^d} \kappa(\boldsymbol{x}) \exp(-\mathbf{i}\langle \boldsymbol{x}, \boldsymbol{u} \rangle) / (2\pi)^d \mathrm{d}\boldsymbol{x} ,$$

where $\mathbf{i}$ is the imaginary unit. By random Fourier features, we can approximate a kernel classifier

$$h(\boldsymbol{x}) = \langle \tilde{\boldsymbol{w}}, \boldsymbol{\varphi}(\boldsymbol{x}) \rangle \approx \langle \boldsymbol{w}, \boldsymbol{z}(\boldsymbol{x}) \rangle ,$$

where the finite-dimensional approximated vector

$$\boldsymbol{z}(\boldsymbol{x}) = \left( \sqrt{p(\boldsymbol{u}_1)} \phi(\boldsymbol{x}, \boldsymbol{u}_1, b_1), \cdots, \sqrt{p(\boldsymbol{u}_d)} \phi(\boldsymbol{x}, \boldsymbol{u}_d, b_d) \right) .$$

We update the classifier according to Fourier online gradient descent (Lu et al., 2016) as follows:

$$\boldsymbol{w}_t = \boldsymbol{w}_{t-1} - \tau_t \nabla \ell_t(\boldsymbol{w}_{t-1}) , \quad (2)$$

where $\tau_t$ is the stepsize and loss function

$$\ell_t(\boldsymbol{w}_{t-1}) = \ell\left(\boldsymbol{w}_{t-1}, (\boldsymbol{z}(\boldsymbol{x}_t), y_t)\right) + \frac{\lambda}{2}\|\boldsymbol{w}_{t-1}\|_2^2 .$$

This work focuses on random Fourier feature technique over shift-invariant kernels, and it is natural to make similar approximation and online algorithm for other kernels such as polynomial kernel (Pennington et al., 2015) and linear kernel (Wacker et al., 2024).

We introduce some notations throughout this work. Bold uppercase and lowercase letters denote matrices and vectors, respectively. We denote by $\|\cdot\|_p$ the $\ell_p$-norm of a vector, and $\|\cdot\|_F$ and $\|\cdot\|_*$ denote the Frobenius norm and nuclear norm of a matrix, respectively. Denote by $\|\cdot\|_{\text{HS}}$ the Hilbert-Schmidt norm of an operator, which is an extension of the Frobenius norm in Hilbert space. Let $(\boldsymbol{v}_i)_{i=1}^n$ be the $d \times n$ concatenation matrix of vectors $\boldsymbol{v}_1, \cdots, \boldsymbol{v}_n \in \mathbb{R}^d$.

Let $\mathbf{I}_d$ be an $d \times d$ identity matrix, and $\text{diag}(\boldsymbol{v})$ is a diagonal matrix with diagonal elements $\boldsymbol{v}$. Denote by $\mathbf{1}_d$ and $\mathbf{0}_d$ $d$-dimensional vectors with all-one and all-zero elements, respectively. Let $\mathcal{N}(\boldsymbol{\mu}, \boldsymbol{\Sigma})$ be a Gaussian distribution with parameters $\boldsymbol{\mu}$ and $\boldsymbol{\Sigma}$. Let $\mathcal{U}_d = \{\mathbf{U} \in \mathbb{R}^{d \times d} : \mathbf{U}\mathbf{U}^\top = \mathbf{I}_d\}$ be the set of $d \times d$ orthogonal matrices, and $\sqrt{\mathbf{A}}$ is the square root of positive-semidefinite matrix $\mathbf{A}$, i.e., $\sqrt{\mathbf{A}}\sqrt{\mathbf{A}} = \mathbf{A}$.

# 3. On the Exploration of Feature Relationship

In this section, we introduce a general framework on the characterization of relationship between different feature spaces based on kernel functions, while previous studies can be viewed as some special selections of different kernels (Hou et al., 2017; 2021; Zhou et al., 2024). We further correlate it with the distance between classifiers trained from different feature spaces and then develop the one-pass learning algorithm for optimization.

## 3.1. Charactering Relationship between Feature Spaces

Our basic idea is to map the original raw feature spaces into Reproducing Kernel Hilbert Spaces (RKHSs) with kernel functions, which could provide plentiful and implicit non-linear representations for original feature spaces.

We focus on positive-definite kernel functions $\mathcal{K}^{[1]}$ and $\mathcal{K}^{[2]}$ over old feature space $\mathcal{X}^{[1]}$ and new feature space $\mathcal{X}^{[2]}$, respectively. For $S_n = \{(\boldsymbol{x}_1^{[1]}, \boldsymbol{x}_1^{[2]}), \cdots, (\boldsymbol{x}_n^{[1]}, \boldsymbol{x}_n^{[2]})\}$ with $\boldsymbol{x}_i^{[1]} \in \mathcal{X}^{[1]}$ and $\boldsymbol{x}_i^{[2]} \in \mathcal{X}^{[2]}$, we define their Gram matrices

$$\mathbf{K}^{[k]} = \left[\mathcal{K}^{[k]}\big(\boldsymbol{x}_i^{[k]}, \boldsymbol{x}_j^{[k]}\big)\right]_{n \times n} \quad \text{for} \quad k = 1, 2 .$$

For $\mathcal{K}^{[1]}$ and $\mathcal{K}^{[2]}$, we introduce a new distance to measure the difference between two feature spaces as follows.

**Definition 3.1.** We define Kernel Ortho-Mapping (KOM) discrepancy between $\mathcal{K}^{[1]}$ and $\mathcal{K}^{[2]}$ over sample $S_n$ as

$$\hat{\mathcal{E}}(S_n, \mathcal{K}^{[1]}, \mathcal{K}^{[2]}) = \min_{\mathbf{U} \in \mathcal{U}_n} \left\{\left\|\mathbf{U}\sqrt{\mathbf{K}^{[1]}} - \sqrt{\mathbf{K}^{[2]}}\right\|_F / \sqrt{n}\right\} .$$

In this definition, the empirical kernel mapping is introduced to deal with different dimensionalities of kernel mappings as done by Schölkopf & Smola (2002) and Marukatat (2016), and the minimum is taken for the uniqueness of kernel mapping from rotational invariance.

**Lemma 3.2.** *We have the closed-form solution for the KOM discrepancy (in Definition 3.1) as*

$$\hat{\mathcal{E}}\left(S_n, \mathcal{K}^{[1]}, \mathcal{K}^{[2]}\right)$$
$$= \left(\text{Tr}(\mathbf{K}^{[1]} + \mathbf{K}^{[2]})/n - 2\|\sqrt{\mathbf{K}^{[1]}}\sqrt{\mathbf{K}^{[2]}}\|_*/n\right)^{1/2} .$$

The detailed proof is given in Appendix A.1, and the basic idea is to take the polar decomposition and upper bound the trace of an orthogonal matrix over $\mathcal{U}_n$.

For loss function $\ell(h, (\boldsymbol{x}, y)) = \max\{0, 1 - yh_t(\boldsymbol{x})\}$, we define the optimal kernel classifiers over sample $S_n$ in the old and new feature spaces as follows:

$$h_*^{[k]} \in \underset{h^{[k]} \in \mathbb{H}^{[k]}}{\arg\min} \sum_{i=1}^n \frac{\ell(h^{[k]}, (\boldsymbol{x}_i^{[k]}, y_i))}{n} + \frac{\lambda}{2}\|h^{[k]}\|_{\mathbb{H}^{[k]}}^2 , \quad (3)$$

where $k = 1$ and $k = 2$ correspond to the old and new feature spaces, respectively. We measure the gap between two optimal classifiers $h_*^{[1]}$ and $h_*^{[2]}$ over sample $S_n$ as follows:

$$\hat{\rho}_{S_n}(h_*^{[1]}, h_*^{[2]}) = \frac{1}{n} \sum_{i=1}^n \left|h_*^{[1]}(\boldsymbol{x}_i^{[1]}) - h_*^{[2]}(\boldsymbol{x}_i^{[2]})\right| . \quad (4)$$

We now present the first main result to correlate our KOM discrepancy with two optimal classifiers as follows:

**Theorem 3.3.** *Given sample $S_n$, we have*

$$\hat{\rho}_{S_n}(h_*^{[1]}, h_*^{[2]}) \leq \frac{r}{\lambda}\hat{\mathcal{E}}(S_n, \mathcal{K}^{[1]}, \mathcal{K}^{[2]}) + \frac{r}{\lambda}\left(2r\hat{\mathcal{E}}(S_n, \mathcal{K}^{[1]}, \mathcal{K}^{[2]})\right)^{\frac{1}{2}}$$

*for two kernels $\mathcal{K}^{[1]}$ and $\mathcal{K}^{[2]}$ bounded by $r^2$, where $\lambda$ is the regularization parameter in Eqn. (3).*

In this theorem, we upper bound the distance between two optimal classifiers with our KOM discrepancy, and this may shed some new insights to develop feature evolvable learning algorithms based on our KOM discrepancy, which essentially measures the relationships between two different feature spaces. The detailed proof of Theorem 3.3 is given in Appendix A.2, which linearizes the kernel classifiers via empirical kernel mapping and constructs KOM discrepancy.

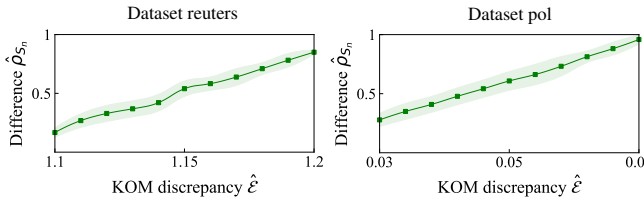

*Figure 2.* Illustrations of the relationship between $\hat{\mathcal{E}}$ and $\hat{\rho}_{S_n}$.

Theorem 3.3 is limited to binary classification, while it is easy to make a similar analysis for multi-class learning (Crammer & Singer, 2002) and regression (Murphy, 2012).

Figure 2 presents an intuitive illustration on the relationship between KOM discrepancy $\hat{\mathcal{E}}(S_n, \mathcal{K}^{[1]}, \mathcal{K}^{[2]})$ and classifiers' difference $\hat{\rho}_{S_n}(h_*^{[1]}, h_*^{[2]})$ via Gaussian kernels over datasets pol and reuters. As we can see, $\hat{\rho}_{S_n}(h_*^{[1]}, h_*^{[2]})$ is positively relevant to KOM discrepancy $\hat{\mathcal{E}}(S_n, \mathcal{K}^{[1]}, \mathcal{K}^{[2]})$, i.e., the bigger the KOM discrepancy $\hat{\mathcal{E}}(S_n, \mathcal{K}^{[1]}, \mathcal{K}^{[2]})$, the larger the distance $\hat{\rho}_{S_n}(h_*^{[1]}, h_*^{[2]})$. This is nicely in accordance with our Theorem 3.3 empirically.

Our KOM discrepancy is defined over the sample $S_n$ in Definition 3.1. We can also define the KOM discrepancy w.r.t. distribution $\mathcal{D}$ over $\mathcal{X}^{[1]} \times \mathcal{X}^{[2]}$ as follows

$$\mathcal{E}(\mathcal{D}, \mathcal{K}^{[1]}, \mathcal{K}^{[2]})$$
$$= \min_{\mathbf{U} \in \mathcal{U}} \left\{ \sqrt{\mathbb{E}_{\mathcal{D}} \left[ \|\mathbf{U}\boldsymbol{\varphi}^{[1]}(\boldsymbol{x}^{[1]}) - \boldsymbol{\varphi}^{[2]}(\boldsymbol{x}^{[2]})\|_{\mathrm{HS}}^2 \right]} \right\},$$

where $\mathcal{U}$ is a unitary operator set on a real Hilbert space. We present the following convergence analysis.

**Theorem 3.4.** *Let $\mathcal{K}^{[1]}$ and $\mathcal{K}^{[2]}$ be two kernels bounded by $r^2$. For $\delta \in (0, 1)$ and for some constant $c_1 > 0$, we have, with probability at least $1 - \delta$ over sample $S_n$*

$$\left| \hat{\mathcal{E}}(S_n, \mathcal{K}^{[1]}, \mathcal{K}^{[2]}) - \mathcal{E}(\mathcal{D}, \mathcal{K}^{[1]}, \mathcal{K}^{[2]}) \right| \leq c_1 r \sqrt{\frac{1}{n} \ln \frac{2}{\delta}}.$$

The detailed proof is presented in Appendix A.3, in which the main techniques include McDiarmid's inequality (McDiarmid et al., 1989) and operator Khintchine's inequality in non-commutative probability (Vershynin, 2018).

**Relevant to previous relationship characterizations**

Kernel alignment has been used to characterize relationship between two kernels (Cristianini et al., 2001; Cortes et al., 2012; Zhou et al., 2024), which essentially calculates the cosine similarity between two Gram matrices $\mathbf{K}^{[1]}$ and $\mathbf{K}^{[2]}$

$$\hat{A}(\mathbf{K}^{[1]}, \mathbf{K}^{[2]}) = \frac{\mathrm{Tr}(\mathbf{K}^{[1]}\mathbf{K}^{[2]})}{\|\mathbf{K}^{[1]}\|_F \|\mathbf{K}^{[2]}\|_F}.$$

We could present the following relationship between our KOM discrepancy and kernel alignment, and the detailed proof is given in Appendix A.4.

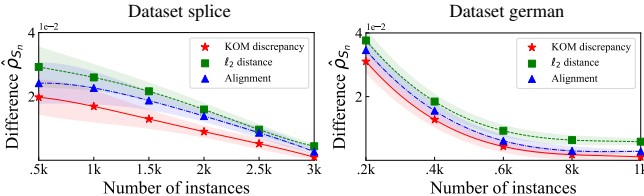

*Figure 3.* An illustration of the difference between two optimal classifiers on two feature spaces, by optimizing KOM discrepancy and previous kernel alignment and $\ell_2$ distance, respectively.

**Lemma 3.5.** *For two normalized kernel matrices $\mathbf{K}^{[1]}$ and $\mathbf{K}^{[2]}$ with $\|\mathbf{K}^{[1]}\|_F = \|\mathbf{K}^{[2]}\|_F \leq r^2$, we have*

$$\hat{\mathcal{E}}(S_n, \mathcal{K}^{[1]}, \mathcal{K}^{[2]}) \leq r \sqrt[4]{2(1 - \hat{A}(\mathbf{K}^{[1]}, \mathbf{K}^{[2]}))}.$$

The $\ell_2$ distance has also been used to align the features of two kernel mappings (Romero et al., 2015; Heo et al., 2019), which solves a finite-dimensional kernel mapping $\hat{\boldsymbol{\varphi}}^{[2]}$ from the following optimization problem:

$$\hat{\boldsymbol{\varphi}}^{[2]} \in \arg \min_{\boldsymbol{\varphi}^{[2]} \in \mathcal{F}} \left\{ \frac{1}{n} \sum_{i=1}^{n} \|\boldsymbol{\varphi}^{[1]}(\boldsymbol{x}_i^{[1]}) - \boldsymbol{\varphi}^{[2]}(\boldsymbol{x}_i^{[2]})\|_2^2 \right\}.$$

Here, function space $\mathcal{F} \subseteq \{\boldsymbol{\varphi}^{[2]} : \mathbb{R}^{d^{[2]}} \to \mathbb{R}^{\dim(\boldsymbol{\varphi}^{[1]})}\}$, and $\boldsymbol{\varphi}^{[1]}$ is the finite-dimensional mapping of kernel $\mathcal{K}^{[1]}$.

The $\ell_2$ distance has been successfully applied for feature evolvable learning. For example, Hou et al. (2017; 2022) selected linear kernels $\mathcal{K}^{[1]}$ and $\mathcal{K}^{[2]}$, while Chen & Liu (2024) considered Gaussian kernel $\mathcal{K}^{[1]}$ and Mahalanobis kernel $\mathcal{K}^{[2]}$. We can also present the following relationship between the KOM discrepancy and $\ell_2$ distance, and the detailed proof is given in Appendix A.5.

**Lemma 3.6.** *For kernel $\mathcal{K}^{[1]}$ with mapping $\boldsymbol{\varphi}^{[1]}$, we have*

$$\left(\hat{\mathcal{E}}(S_n, \mathcal{K}^{[1]}, \hat{\mathcal{K}}^{[2]})\right)^2 \leq \frac{1}{n} \sum_{i=1}^{n} \left\|\boldsymbol{\varphi}^{[1]}(\boldsymbol{x}_i^{[1]}) - \hat{\boldsymbol{\varphi}}^{[2]}(\boldsymbol{x}_i^{[2]})\right\|_2^2,$$

*where kernel $\hat{\mathcal{K}}^{[2]}(\boldsymbol{x}^{[2]}, \boldsymbol{x}^{[2]\prime}) = \langle \hat{\boldsymbol{\varphi}}^{[2]}(\boldsymbol{x}^{[2]}), \hat{\boldsymbol{\varphi}}^{[2]}(\boldsymbol{x}^{[2]\prime}) \rangle$.*

Figure 3 presents an illustration of the difference between two optimal classifiers over two feature spaces by optimizing our KOM discrepancy, previous $\ell_2$ distance and kernel alignment, respectively. Here, we focus on simple linear mapping spaces on two datasets splice and german, and the trends are similar for other datasets.

From Figure 3, it is observable that we could get a smaller difference between two optimal classifiers by optimizing the KOM discrepancy, rather than kernel alignment and $\ell_2$ distance. Therefore, our KOM discrepancy presents a better characterization of relationships between different feature spaces via kernel functions, and this is partially consistent with Lemma 3.5 and Lemma 3.6.

---

**Algorithm 1** One-pass optimization of Eqn. (6)

---

**Input**: Streaming sample $S_{T_e}^{[e]}$, number of iterations $T_M$, and stepsize $\eta_M$

**Output**: Spectral density $\boldsymbol{p}^{(T_M)}$

1: Initialize $\mathbf{M}^{(T_1)} = \mathbf{0}$, $\boldsymbol{v}^{(T_1)} = \mathbf{0}$ and $\boldsymbol{p}^{(0)} = \mathbf{1}/d_2$
2: **for** $t = T_1 + 1, \cdots, T_1 + T_e$ **do**
3:     Update $\mathbf{M}^{(t)}$ and $\boldsymbol{v}^{(t)}$ according to Eqns. (7)-(8)
4: **end for**
5: **for** $i = 1, \cdots, T_M$ **do**
6:     Calculate $\boldsymbol{q} = \boldsymbol{p}^{(i-1)} \exp(-\eta_M \nabla f(\boldsymbol{p}^{(i-1)}))$
7:     Update $\boldsymbol{p}^{(i)} = \boldsymbol{q}/\|\boldsymbol{q}\|_1$
8: **end for**
9: **return:** Spectral density $\boldsymbol{p}^{(T_M)}$

---

### 3.2. One-Pass Optimization for our KOM discrepancy

During the evolving stage, we reach a streaming sample $S_{T_e}^{[e]} = \{(\boldsymbol{x}_{T_1+1}^{[1]}, \boldsymbol{x}_{T_1+1}^{[2]}), \cdots, (\boldsymbol{x}_{T_1+T_e}^{[1]}, \boldsymbol{x}_{T_1+T_e}^{[2]})\}$, and get the kernel $\mathcal{K}^{[1]}$ learned from the previous stage. Motivated by Theorem 3.3, we learn the kernel $\mathcal{K}^{[2]}$ by minimizing the KOM discrepancy as follows

$$\min_{\mathcal{K}^{[2]}} \left\{ \hat{\mathcal{E}}(S_{T_e}^{[e]}, \mathcal{K}^{[1]}, \mathcal{K}^{[2]}) \right\} . \tag{5}$$

For steaming sample $S_{T_e}^{[e]}$, it is not allowed to store kernel Gram matrices in memory, and we can not directly optimize the above optimization as in (Cortes et al., 2012; Liu, 2024).

We present one-pass optimization for Eqn. (5) via random Fourier features. The basic idea is to transform the original optimization into a convex problem on a simplex, and then solve it w.r.t. streaming data $S_{T_e}^{[e]}$. For the spectral density $\boldsymbol{p} = (p^{[2]}(\boldsymbol{u}_1^{[2]}), \cdots, p^{[2]}(\boldsymbol{u}_{d_2}^{[2]}))$ of $\mathcal{K}^{[2]}$, we approximate the KOM discrepancy by random Fourier features as

$$\hat{\mathcal{E}}(S_{T_e}^{[e]}, \mathcal{K}^{[1]}, \mathcal{K}^{[2]})$$
$$\approx \sqrt{(\mathrm{Tr}(\mathbf{K}^{[1]}) + \langle \boldsymbol{p}, \boldsymbol{v} \rangle - 2\|\mathbf{M}\mathrm{diag}(\sqrt{\boldsymbol{p}})\|_*)/T_e} ,$$

where $\mathbf{M} = [m_{kl}]_{d_1 \times d_2}$ and $\boldsymbol{v} = (v_1, v_2 \cdots, v_{d_2})$ with

$$m_{kl} = \sum_{i=T_1+1}^{T_1+T_e} p^{[1]}(\boldsymbol{u}_k^{[1]}) \Phi_{ik}^{[1]} \Phi_{il}^{[2]} \text{ and } v_k = \sum_{i=T_1+1}^{T_1+T_e} (\Phi_{ik}^{[2]})^2 .$$

Here, $\Phi_{ij}^{[1]} = \phi(\boldsymbol{x}_i^{[1]}, \boldsymbol{u}_j^{[1]}, b_j^{[1]})$, $\Phi_{ij}^{[2]} = \phi(\boldsymbol{x}_i^{[2]}, \boldsymbol{u}_j^{[2]}, b_j^{[2]})$, and $d_1$ and $d_2$ are the numbers of random Fourier features of kernels $\mathcal{K}^{[1]}$ and $\mathcal{K}^{[2]}$, respectively. From random features approximation, Eqn. (5) is essentially equivalent to

$$\min_{\boldsymbol{p} \in \Delta} \left\{ f(\boldsymbol{p}) = \frac{1}{2}\langle \boldsymbol{p}, \boldsymbol{v} \rangle - \|\mathbf{M}\mathrm{diag}(\sqrt{\boldsymbol{p}})\|_* \right\} , \tag{6}$$

where $\Delta = \{\boldsymbol{p}: \boldsymbol{p} \geq \mathbf{0}, \|\boldsymbol{p}\|_1 = 1\}$. For Eqn. (6), we initialize $\mathbf{M}^{(T_1)} = [\mathbf{0}]_{d_1 \times d_2}$ and $\boldsymbol{v}^{(T_1)} = \mathbf{0}_{d_2}$, and in the

$t$-th round ($t \geq T_1 + 1$), we update $\mathbf{M}^{(t)}$ and $\boldsymbol{v}^{(t)}$ w.r.t. instances $\boldsymbol{x}_t^{[1]}$ and $\boldsymbol{x}_t^{[2]}$, respectively, as

$$m_{kl}^{(t)} = m_{kl}^{(t-1)} + p^{[1]}(\boldsymbol{u}_k^{[1]}) \Phi_{tk}^{[1]} \Phi_{tl}^{[2]} \tag{7}$$
$$v_k^{(t)} = v_k^{(t-1)} + (\Phi_{tk}^{[2]})^2 , \tag{8}$$

with $\Phi_{ij}^{[1]} = \phi(\boldsymbol{x}_i^{[1]}, \boldsymbol{u}_j^{[1]}, b_j^{[1]})$ and $\Phi_{ij}^{[2]} = \phi(\boldsymbol{x}_i^{[2]}, \boldsymbol{u}_j^{[2]}, b_j^{[2]})$.

We finally take the mirror descent method (Bubeck, 2015; Hazan et al., 2016) to solve Eqn. (6).

Algorithm 1 presents the details of our proposed method, and we have the convergence analysis as follows.

**Theorem 3.7.** *For Algorithm 1, we have*

$$\frac{1}{T_M} \sum_{t=1}^{T_M} f(\boldsymbol{p}^{(t)}) - f(\boldsymbol{p}^*) \leq O\left(\frac{1}{\sqrt{T_M}}\right) ,$$

*by setting stepsize $\eta_M = \Theta(\sqrt{\ln d_2/T_M})$, where $f(\cdot)$ is defined by Eqn. (6) and $\boldsymbol{p}^* \in \arg\min_{\boldsymbol{p} \in \Delta} f(\boldsymbol{p})$.*

The detailed proof is given in Appendix B, which presents an operation to preserve convexity and then derives the convergence analysis of mirror descent on a simplex.

## 4. The OPFES Approach

This section presents the one-pass optimization for feature evolvable learning in the current stage as shown in Figure 1. Our idea is to incorporate feature and label information from previous relationships and stages, and reuse prior models, rather than re-training a new model from scratch.

**i) Incorporation of feature information via kernel $\mathcal{K}^{[2]}$**

Notice that the kernel $\mathcal{K}^{[2]}$ is learned by minimizing the KOM discrepancy in Section 3. We could train an online kernel model based on $\mathcal{K}^{[2]}$, which incorporate implicitly some information from old feature space.

Specifically, we reach an example $(\boldsymbol{x}_t^{[2]}, y_t)$ in the $t$-th round for $T_1 + T_e + 1 \leq t \leq T_1 + T_e + T_2$, and learn an online model $h_t^{[2]}(\boldsymbol{x}^{[2]})$ based on kernel $\mathcal{K}^{[2]}$ as follows

$$h_t^{[2]}(\boldsymbol{x}^{[2]}) \approx \langle \boldsymbol{w}_t^{[2]}, \boldsymbol{z}^{[2]}(\boldsymbol{x}^{[2]}) \rangle ,$$

where $\boldsymbol{z}^{[2]}(\boldsymbol{x}^{[2]}) = ((p^{[2]}(\boldsymbol{u}_k^{[2]}))^{1/2}\phi(\boldsymbol{x}^{[2]}, \boldsymbol{u}_k^{[2]}, b_k^{[2]}))_{k=1}^{d_2}$. Here, we take the representer theorem (Schölkopf & Smola, 2002) and random features approximation from Eqn. (1). We take online gradient descent with stepsize $\tau_t^{[2]}$ as

$$\boldsymbol{w}_t^{[2]} = \boldsymbol{w}_{t-1}^{[2]} - \tau_t^{[2]} \nabla \ell_t^{[2]}(\boldsymbol{w}_{t-1}^{[2]}) , \tag{9}$$

where the loss function $\ell_t^{[2]}(\boldsymbol{w}_{t-1}^{[2]})$ is given by

$$\max\left\{0, 1 - y_t\langle \boldsymbol{w}_{t-1}^{[2]}, \boldsymbol{z}^{[2]}(\boldsymbol{x}_t^{[2]}) \rangle\right\} + \frac{\lambda}{2}\|\boldsymbol{w}_{t-1}^{[2]}\|_2^2 .$$

**ii) Incorporation of label information via ideal kernel**

For label information, we exploit the *ideal kernel* from kernel selections (Cristianini et al., 2001; Kwok et al., 2003). Essentially, the ideal kernel is helpful to learn models with correct predictions for training data (Cristianini et al., 2001). We define the ideal kernel $\mathcal{K}^*$ over feature space $\mathcal{X}^{[2]}$ as

$$\mathcal{K}^*(\boldsymbol{x}_i^{[2]}, \boldsymbol{x}_j^{[2]}) = y_i y_j \ \text{ for } \ i, j \in \{T_1 + 1, \cdots, T_1 + T_e\} \ .$$

We can not obtain the ideal kernel matrix owing to streaming data. Our idea is to learn a new kernel $\mathcal{K}^l$ aligning with the ideal kernel $\mathcal{K}^*$ to incorporate label information, i.e.,

$$\mathcal{K}^l = \arg\min_{\mathcal{K}^l} \left\{ \hat{\mathcal{E}}\left( S_{T_e}^{[e]}, \mathcal{K}^*, \mathcal{K}^l \right) \right\} \ .$$

In the $t$-th round, we learn a new online model by representer theorem and random features approximation again, i.e.,

$$h_t^l(\boldsymbol{x}^{[2]}) \approx \langle \boldsymbol{w}_t^l, \boldsymbol{z}^l(\boldsymbol{x}^{[2]}) \rangle \ ,$$

where $\boldsymbol{z}^l(\boldsymbol{x}^{[2]}) = ((p^l(\boldsymbol{u}_k^l))^{1/2}\phi(\boldsymbol{x}^{[2]}, \boldsymbol{u}_k^l, b_k^l))_{k=1}^{d_2}$ with the density $\boldsymbol{p}^l$ of kernel $\mathcal{K}^l$. Given stepsize $\tau_t^l$, we update

$$\boldsymbol{w}_t^l = \boldsymbol{w}_{t-1}^l - \tau_t^l \nabla \ell_t^l(\boldsymbol{w}_{t-1}^l) \ , \tag{10}$$

where the loss function $\ell_t^l(\boldsymbol{w}_{t-1}^l)$ is given by

$$\max\left\{0, 1 - y_t \langle \boldsymbol{w}_{t-1}^l, \boldsymbol{z}^l(\boldsymbol{x}_t^{[2]}) \rangle \right\} + \frac{\lambda}{2} \left\| \boldsymbol{w}_{t-1}^l \right\|_2^2 \ .$$

**iii) Previous model reuse**

We exploit some good model initializations in the current stage, rather than re-training from scratch. Our basic idea is to obtain a new model on the space spanned by $\boldsymbol{z}^{[2]}(\boldsymbol{x}^{[2]})$, which takes similar predictions with the previous model.

Specifically, we consider the model for the new feature space via an orthogonal transformation

$$\boldsymbol{w}_{T_1+T_e}^{[2]} = \mathbf{U}_*^\top \boldsymbol{w}_{T_1}^{[1]} \ , \tag{11}$$

where $\mathbf{U}_*$ is in the set of

$$\arg\min_{\mathbf{U} \in \mathcal{U}_{d_1}} \left\{ \left\| \mathbf{U}\left(\boldsymbol{z}^{[1]}(\boldsymbol{x}_{T_1+i}^{[1]})\right)_{i=1}^{T_e} - \left(\boldsymbol{z}^{[2]}(\boldsymbol{x}_{T_1+i}^{[2]})\right)_{i=1}^{T_e} \right\|_F \right\} \ .$$

In the following, we present an effective solution for $\mathbf{U}_*$, and the detailed proof is given in Appendix C.1.

**Proposition 4.1.** *We have* $\mathbf{U}_* = \mathbf{V}\mathbf{W}^\top$ *for the optimal solution in Eqn. (11), where $\mathbf{V}$ and $\mathbf{W}$ are left and right singular vectors of* $\mathbf{M}^{(T_1+T_e)}diag(\sqrt{\boldsymbol{p}^{(T_M)}})$ *in Algorithm 1.*

Based on previous analysis, we should learn and update two online kernel models $\boldsymbol{w}_t^{[2]}$ and $\boldsymbol{w}_t^l$ from Eqns. (9) and (10)

---

**Algorithm 2** The OPFES method

**Input**: Feature evolvable stream sample $S_{T_1+T_e+T_2}$, kernel $\mathcal{K}^{[1]}$, stepsize $\tau_t^{[1]}, \tau_t^{[2]}$ and $\tau_t^l$, sensitivity parameter $\gamma$
**Initialize**: $\boldsymbol{w}_0^{[1]} = \boldsymbol{0}$
**Output**: classifier $h_{T_1+T_e+T_2}$

1: Obtain random Fourier features $(\boldsymbol{u}_k^{[1]}, b_k^{[1]})_{k=1}^{d_1}$ and $(\boldsymbol{u}_k^{[2]}, b_k^{[2]}, \boldsymbol{u}_k^l, b_k^l)_{k=1}^{d_2}$ via Eqn. (1)
2: **for** $t = 1, \cdots, T_1$ **do**
3:   Update $\boldsymbol{w}_t^{[1]}$ by online gradient descent in Eqn. (2)
4: **end for**
5: Obtain $\boldsymbol{p}^{[2]}$ and $\boldsymbol{p}^l$ from Algorithm 1
6: Compute $\boldsymbol{w}_{T_1+T_e}^{[2]}$ by Eqn. (11)
7: **for** $t = T_1 + T_e + 1, \cdots, T_1 + T_e + T_2$ **do**
8:   Update $\boldsymbol{w}_t^{[2]}$ and $\boldsymbol{w}_t^l$ by Eqns. (9)-(10), respectively
9:   Update the combined classifier $h_t$ by Eqn. (12)
10: **end for**
11: **return:** classifier $h_{T_1+T_e+T_2}$

---

in the current stage, respectively. From (Hou et al., 2021), we combine two models, for $t \geq T_1 + T_e + 1$,

$$h_t(\boldsymbol{x}_t^{[2]}) = \omega_t \langle \boldsymbol{w}_t^{[2]}, \boldsymbol{z}^{[2]}(\boldsymbol{x}_t^{[2]}) \rangle$$
$$+ (1 - \omega_t) \langle \boldsymbol{w}_t^l, \boldsymbol{z}^l(\boldsymbol{x}_t^{[2]}) \rangle \ , \tag{12}$$

where $\omega_t$ is relevant to a sensitivity parameter $\gamma > 0$, i.e.,

$$\omega_t = \frac{\omega_{t-1}e^{-\gamma\ell_t^{[2]}(\boldsymbol{w}_{t-1}^{[2]})}}{\omega_{t-1}e^{-\gamma\ell_t^{[2]}(\boldsymbol{w}_{t-1}^{[2]})} + (1-\omega_{t-1})e^{-\gamma\ell_t^l(\boldsymbol{w}_{t-1}^l)}} \ .$$

Algorithm 2 presents the detailed description of our OPFES approach, which goes through all instances only once without storing the entire or partial training data, while previous methods require storing the entire dataset or partial dataset (Orabona et al., 2008; Jin et al., 2010; Hou et al., 2021; 2022; He et al., 2021a; Wu et al., 2023).

**Theoretical guarantee**

We begin with the upper bounds for prediction difference between $\boldsymbol{w}_{T_1}^{[1]}$ and $\boldsymbol{w}_{T_1+T_e}^{[2]}$ via our KOM discrepancy, and the detailed proof is presented in Appendix C.2.

**Lemma 4.2.** *For bounded kernels, we have, for previous model $\boldsymbol{w}_{T_1}^{[1]}$ and reused model $\boldsymbol{w}_{T_1+T_e}^{[2]}$ from Eqn. (11),*

$$\frac{1}{T_e} \sum_{t=T_1+1}^{T_1+T_e} \left| \langle \boldsymbol{w}_{T_1}^{[1]}, \boldsymbol{z}^{[1]}(\boldsymbol{x}_t^{[1]}) \rangle - \langle \boldsymbol{w}_{T_1+T_e}^{[2]}, \boldsymbol{z}^{[2]}(\boldsymbol{x}_t^{[2]}) \rangle \right|$$
$$\leq \sqrt{2}\hat{\mathcal{E}}\left(S_{T_e}^{[e]}, \mathcal{K}^{[1]}, \mathcal{K}^{[2]}\right)/\lambda \ ,$$

*where $\lambda$ is the regularization parameter in Eqn. (3).*

*Table 1.* Details of datasets

| Dataset | # Inst. | # Feat. | Dataset | # Inst. | # Feat. | Dataset | # Inst. | # Feat. | Dataset | # Inst. | # Feat. | Dataset | # Inst. | # Feat. |
|---|---|---|---|---|---|---|---|---|---|---|---|---|---|---|
| jungle | 2351 | 87 | svmguide1 | 7089 | 4 | elevators | 16599 | 18 | nomao | 34465 | 118 | higgs | 98049 | 28 |
| splice | 3175 | 60 | usps | 9298 | 25 | magic | 19020 | 10 | adult | 48842 | 108 | miniboone | 130064 | 50 |
| bioresponse | 3751 | 1776 | aileron | 13750 | 40 | letter | 20000 | 16 | acoustic | 78823 | 50 | ijcnn1 | 141691 | 22 |
| christine | 5418 | 1636 | pol | 15000 | 44 | house | 22784 | 16 | runwalk | 88588 | 6 | covtype | 581012 | 54 |

Denote by the optimal model in the current stage

$$\boldsymbol{w}_*^{[2]} \in \arg\min_{\boldsymbol{w}} \left\{ \mathcal{L}_{T_2}^{[2]}(\boldsymbol{w}) = \frac{1}{T_2} \sum_{t=T_1+T_e+1}^{T_1+T_e+T_2} \ell_t^{[2]}(\boldsymbol{w}) \right\} ,$$

and the cumulative loss

$$\hat{\mathcal{L}}_{T_2}^{[2]} = \frac{1}{T_2} \sum_{t=T_1+T_e+1}^{T_1+T_e+T_2} \ell_t^{[2]}(\boldsymbol{w}_t^{[2]}) ,$$

with $\ell_t^{[2]}(\boldsymbol{w}) = \max\{0, 1 - y_t \langle \boldsymbol{w}, \boldsymbol{z}^{[2]}(\boldsymbol{x}_t^{[2]}) \rangle\} + \lambda \|\boldsymbol{w}\|_2^2/2$.

Let $S_{T_2} = \{(\boldsymbol{x}_i^{[2]}, y_i)\}_{i=T_1+T_e+1}^{T_1+T_e+T_2}$ be a streaming sample drawn i.i.d. from a distribution. For Algorithm 2, we have

**Theorem 4.3.** *For kernels $\mathcal{K}^{[1]}$ and $\mathcal{K}^{[2]}$ with bound $r^2$, and for $\delta \in (0, 1)$, the following holds with probability at least $1 - \delta$ over sample $S_{T_2}$*

$$\hat{\mathcal{L}}_{T_2}^{[2]} - \mathcal{L}_{T_2}^{[2]}(\boldsymbol{w}_*^{[2]}) \leq \frac{4r^2}{\lambda\sqrt{T_2}} \left( \frac{\mathcal{E}}{r} + \sqrt{\frac{\mathcal{E}}{r}} \right)^{1/2}$$

$$+ \frac{c_2 r^2}{\lambda\sqrt{T_2}} \left[ \left( \frac{1}{\sqrt{T_1}} + \frac{1}{\sqrt{T_e}} + \frac{1}{\sqrt[4]{T_2}} \right) \sqrt{\ln\frac{6}{\delta}} \right]^{1/2} ,$$

*where $\mathcal{E} = \mathcal{E}(\mathcal{D}, \mathcal{K}^{[1]}, \mathcal{K}^{[2]})$ and $c_2 > 0$ is some constant.*

Theorem 4.3 gives the convergence analysis for Algorithm 2. We obtain tighter bound as for smaller KOM discrepancy, i.e., a closer relationship between old and new feature spaces. It is also useful to exploit information and model from old feature space theoretically, because of tighter bounds as for larger $T_1$ and $T_e$. The detailed proof is given in Appendix C.3, which is motivated from regret analysis (Hazan et al., 2016), generalization bounds via Rademacher complexity (Bartlett & Mendelson, 2002), some online-to-batch conversion techniques (Cesa-Bianchi et al., 2004).

Denote by the cumulative loss for $\boldsymbol{w}^l$

$$\hat{\mathcal{L}}_{T_2}^l = \frac{1}{T_2} \sum_{t=T_1+T_e+1}^{T_1+T_e+T_2} \ell_t^l(\boldsymbol{w}_t^l) ,$$

with $\ell_t^l(\boldsymbol{w}) = \max\{0, 1 - y_t \langle \boldsymbol{w}, \boldsymbol{z}^l(\boldsymbol{x}_t^l) \rangle\} + \lambda \|\boldsymbol{w}\|_2^2/2$. We also analyze the cumulative error rate for Algorithm 2.

**Theorem 4.4.** *For kernels $\mathcal{K}^{[2]}$ and $\mathcal{K}^l$ with bound $r^2$ and for parameter $\gamma = \sqrt{\ln 2/((1 + 3r^2/2\lambda)T_2)}$, we have*

$$\sum_{t=T_1+T_e+1}^{T_1+T_e+T_2} \frac{\mathbb{I}[y_t h_t(\boldsymbol{x}_t^{[2]}) \leq 0]}{T_2} \leq \min\{\hat{\mathcal{L}}_{T_2}^{[2]}, \hat{\mathcal{L}}_{T_2}^l\} + \sqrt{\frac{2\ln 2}{T_2}}$$

*where $\mathbb{I}[\cdot]$ is the indicator function, which returns $1$ if the argument is true, and $0$ otherwise.*

Theorem 4.4 shows that the cumulative error rate of our OPFES method converges to the minimum of the cumulative loss of two classifiers. The detailed proof is presented in Appendix C.4, and the basic idea is to construct a potential function and apply Hoeffding's lemma (Hoeffding, 1963).

# 5. Empirical Study

We conduct experiments on 20 datasets[2], and the details are summarized in Table 1. Most of the datasets have been well-studied for previous feature evolvable learning, and all features have been scaled to $[0, 1]$. We compare our OPFES with state-of-the-art methods on feature evolvable learning.

- align-FESL: Feature evolvable method of random features and kernel alignment for feature and label relationships (Sinha & Duchi, 2016).

- rff-ROGD: Feature evolvable method of random features and $\ell_2$ distance for feature relationships (Lu et al., 2016);

- rff-FESL: Online ensemble of rff-ROGD and random feature models learned from scratch (Hou et al., 2021);

- ker-ROGD: Feature evolvable method with kernel model and $\ell_2$ distance for feature relationships (Hou et al., 2021);

- ker-FESL: Online ensemble of ker-NOGD and kernel models learned from scratch (Hou et al., 2021);

- lin-ROGD: Feature evolvable method with linear models and $\ell_2$ distance for feature relationships (Hou et al., 2017);

- lin-FESL: Online ensemble of lin-ROGD and a linear model learned from scratch (Hou et al., 2017);

- OCDS: Capricious streaming method with a linear model via generative graphical model (He et al., 2021b).

For each dataset, we randomly split the feature space into old feature space $\mathcal{X}^{[1]}$ and new feature space $\mathcal{X}^{[2]}$ with almost equal number of features, following (Gu et al., 2022; Ni et al., 2024). We set $T_e = 1000$ for datasets with a size larger than 10000; otherwise, set $T_e$ as 10% of the dataset's size. We also set $T_1$ and $T_2$ as half of the amount of dataset

---

[2]Downloaded from OpenML and UCI datasets repository

*Table 2.* Cumulative error rate (CER) evaluation of our OPFES and compared methods (mean±std). ●/○ indicates that our OPFES is significantly better/worse than the corresponding algorithms (pairwise t-tests at 95% significance level).

| Dataset | Our OPFES | align-FESL | rff-FESL | rff-ROGD | ker-FESL | ker-ROGD | lin-FESL | lin-ROGD | OCDS |
|---|---|---|---|---|---|---|---|---|---|
| jungle | .0097 ± .0047 | .0099 ± .0028 | .0161 ± .0035● | .0246 ± .0069● | .0276 ± .0055● | .0329 ± .0061● | .1084 ± .0152● | .1471 ± .0144● | .1106 ± .0138● |
| splice | .3070 ± .0079 | .3126 ± .0136 | .3234 ± .0087● | .3662 ± .0215● | .4192 ± .0160● | .4240 ± .0188● | .3447 ± .0097● | .4307 ± .0213● | .3547 ± .0156● |
| bioresponse | .2763 ± .0117 | .2921 ± .0106● | .3051 ± .0137● | .4285 ± .0192● | .3690 ± .0095● | .4454 ± .0116● | .2938 ± .0093● | .3684 ± .0102● | .2951 ± .0112● |
| christine | .3192 ± .0095 | .3205 ± .0090 | .3316 ± .0108● | .3503 ± .0092● | .3858 ± .0096● | .4506 ± .0117● | .3443 ± .0098● | .3439 ± .0098● | .3663 ± .0116● |
| svmguide1 | .1614 ± .0052 | .1617 ± .0056 | .1632 ± .0054● | .2295 ± .0107● | .1900 ± .0061● | .2316 ± .0050● | .2399 ± .0062● | .2451 ± .0102● | .2442 ± .0070● |
| usps | .1684 ± .0044 | .1875 ± .0073● | .1658 ± .0051 | .2184 ± .0061● | .2267 ± .0084● | .2857 ± .0063● | .2654 ± .0081● | .2839 ± .0073● | .2746 ± .0085● |
| aileron | .1963 ± .0034 | .2144 ± .0081● | .2139 ± .0059● | .2344 ± .0098● | .2531 ± .0076● | .2523 ± .0076● | .2466 ± .0066● | .2465 ± .0066● | .3026 ± .0047● |
| pol | .0654 ± .0036 | .0692 ± .0044● | .0686 ± .0023● | .0807 ± .0028● | .0865 ± .0023● | .0956 ± .0034● | .1484 ± .0035● | .1655 ± .0041● | .1515 ± .0038● |
| elevators | .2422 ± .0039 | .2419 ± .0038 | .2467 ± .0037● | .2619 ± .0045● | .2963 ± .0042● | .3003 ± .0051● | .3073 ± .0043● | .3045 ± .0040● | .3073 ± .0042● |
| magic | .2154 ± .0039 | .2206 ± .0039● | .2121 ± .0046 | .2434 ± .0057● | .2656 ± .0033● | .3119 ± .0074● | .2535 ± .0040● | .2988 ± .0073● | .2554 ± .0045● |
| letter | .1354 ± .0043 | .1568 ± .0086● | .1557 ± .0037● | .2311 ± .0067● | .3139 ± .0060● | .3373 ± .0076● | .3380 ± .0038● | .3565 ± .0071● | .3390 ± .0034● |
| house | .1849 ± .0040 | .1927 ± .0030● | .1894 ± .0043● | .2001 ± .0093● | .2598 ± .0112● | .2597 ± .0113● | .2623 ± .0084● | .2658 ± .0120● | .2853 ± .0037● |
| nomao | .0646 ± .0026 | .0680 ± .0024● | .0778 ± .0017● | .0845 ± .0041● | .1302 ± .0034● | .1355 ± .0032● | .0860 ± .0019● | .1107 ± .0039● | .0882 ± .0023● |
| adult | .1875 ± .0023 | .1942 ± .0027● | .1906 ± .0021● | .1932 ± .0029● | .2277 ± .0019● | .2218 ± .0031● | .2050 ± .0033● | .2036 ± .0042● | .2303 ± .0026● |
| acoustic | .3074 ± .0024 | .3227 ± .0036● | .2967 ± .0045○ | .2977 ± .0043○ | .4168 ± .0073● | .4107 ± .0072● | .4321 ± .0079● | .4317 ± .0075● | .4668 ± .0022● |
| runwalk | .2602 ± .0033 | .2890 ± .0055● | .2578 ± .0016○ | .3496 ± .0130● | .3558 ± .0021● | .4355 ± .0061● | .4945 ± .0021● | .4963 ± .0033● | .4972 ± .0027● |
| higgs | .3946 ± .0045 | .4135 ± .0061● | .3803 ± .0074○ | .3807 ± .0080○ | .4577 ± .0054● | .4570 ± .0055● | .4309 ± .0028● | .4481 ± .0139● | .4366 ± .0021● |
| miniboone | .1602 ± .0036 | .2804 ± .0011● | .1729 ± .0029● | .1603 ± .0029 | .2488 ± .0047● | .2484 ± .0047● | .2384 ± .0039● | .2384 ± .0039● | .2803 ± .0011● |
| ijcnn1 | .0616 ± .0115 | .0746 ± .0038● | .0673 ± .0028● | .0747 ± .0083● | .0957 ± .0009● | .0957 ± .0009● | .0951 ± .0007● | .0957 ± .0009● | .0957 ± .0009● |
| covtype | .3782 ± .0008 | .3813 ± .0025● | .3783 ± .0009 | .3795 ± .0012● | .4095 ± .0025● | .4093 ± .0025● | .3790 ± .0007● | .3920 ± .0034● | .3792 ± .0007● |
| Win/Tie/Loss | | 15/5/0 | 14/3/3 | 17/1/2 | 20/0/0 | 20/0/0 | 20/0/0 | 20/0/0 | 20/0/0 |

size subtracting $T_e$. For ker-ROGD and ker-FESL, we set the buffer size to $10\%$ of dataset size and consider reservoir sampling as done by Hou et al. (2021).

For rff-ROGD, rff-FESL, align-FESL and our OPFES, we fix the dimensionality of random Fourier feature as 1000. In the previous stage, we employ Gaussian kernels with widths in $2^{[-6:6]}$ for all methods. For OPFES, we set $T_M = 1000$ of the optimal stepsize from Theorem 3.7. The stepsize $\tau_t$ is constrained within $10^{[-4:2]}/\sqrt{t}$, and the regularization parameter $\lambda$ is selected from $10^{[-10:1]}$. For OCDS, $\alpha$ and $\beta$ are chosen form $10^{[-5:0]}$ by cross validations.

The performance of the compared methods is evaluated by 50 times on each dataset with random partitions and random ordering in the previous, evolving and current stages, where the cumulative error rate (CER) is obtained by averaging over these 50 runs, as summarized in Table 2.

It is observable that, from Table 2, our OPFES method takes significantly better performance than three linear methods lin-ROGD, lin-FESL, and OCDS, since these methods rely on simple linear classifiers and $\ell_2$ distance to characterize feature relationships. Our OPFES outperforms ker-ROGD, ker-FESL, rff-ROGD most times because of exploration on feature relationships via our KOM discrepancy, while other methods fix Gaussian or Mahalanobis kernels.

Our OPFES is also better than align-FESL, since the KOM discrepancy is more effective in capturing some feature relationships than kernel alignment, as shown in Lemma 3.5.

*Table 3.* Ablation studies for our OPFES method (mean±std): (i) OPFES without KOM discrepancy; (ii) OPFES without ideal kernel; (iii) OPFES without initialization from previous model.

| Datasets | OPFES | (i) | (ii) | (iii) |
|---|---|---|---|---|
| Pol | .0654 ± .0036 | .1290 ± .0044● | .0674 ± .0059 | .1418 ± .0072● |
| House | .1849 ± .0033 | .1976 ± .0032● | .1957 ± .0034● | .2350 ± .0033● |
| Nomao | .0646 ± .0026 | .0933 ± .0028● | .0750 ± .0047● | .1875 ± .0056● |
| Adult | .1875 ± .0023 | .2013 ± .0035● | .1988 ± .0043● | .2133 ± .0064● |

Our OPFES achieves better and comparable performance in contrast to rff-FESL, except for datasets acoustic, runwalk, and higgs. This is partially because of the class-imbalance problem on the three datasets, which results in the hardness of learning ideal kernel $\mathcal{K}^l$ for label information and the degrade of learning performance of base learners $\boldsymbol{w}^l$.

Table 3 presents the ablation study of OPFES to verify the effectiveness of KOM discrepancy, ideal kernel and initializations from previous models. Due to pages limit, we consider four datasets Pol, House, Nomao and Adult, while the trends are similar for other datasets. It is clear that the performance of our OPFES will decrease drastically without the consideration of KOM discrepancy and good initializations. Ideal kernel takes limited improvement from label correlation, in particularly for dataset Pol, where $\mathcal{K}^{[2]}$ possibly takes comparable performance to $\mathcal{K}^l$.

Figure 4 presents the convergence analysis of cumulative error rate for our OPFES. It is evident that our OPFES takes faster convergence than other methods in the current stage,

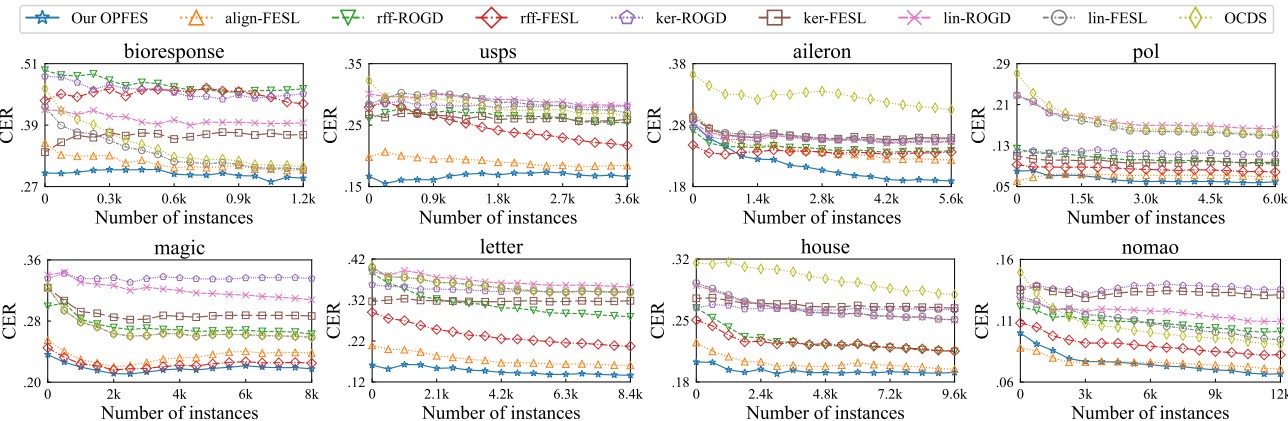

*Figure 4.* Cumulative error rate (CER) versus the number of instances in the current stage for our OPFES and compared methods. The lower the curve, the faster the convergence.

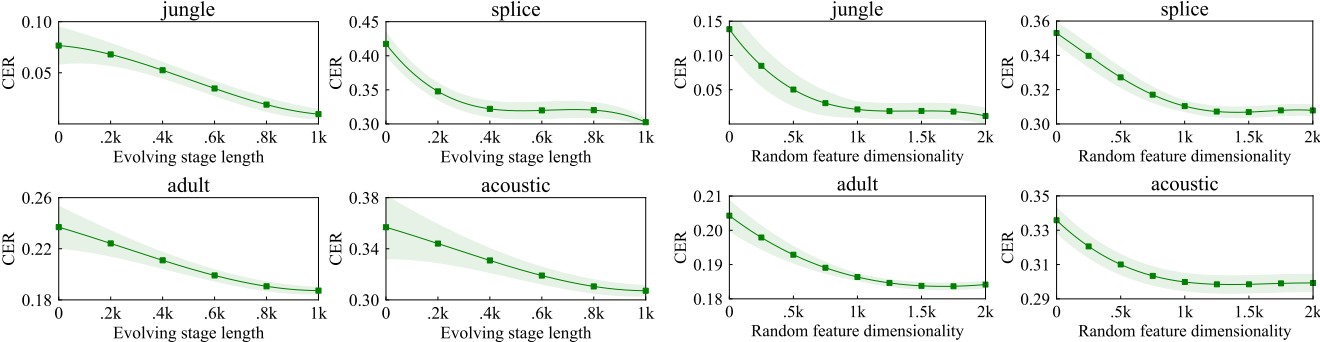

*Figure 5.* Cumulative error rate (CER) versus the length of evolving stage for our OPFES.

*Figure 6.* Cumulative error rate (CER) versus the dimensionality of random Fourier features for our OPFES.

partially because our OPFES can exploit feature relationships between two feature spaces and label information. It is observable that our method obtains lower cumulative error rates in the beginning of the current stage, indicating the importance of good model initializations from the evolving stage, which is consistent with Theorem 4.3.

Figure 5 empirically illustrates the influences of the evolving stage length $T_e$ in Algorithm 2. It is evident that larger $T_e$ could yield better performance, because of more available instances in training to capture feature relationships and label information, which is consistent with our theoretical analysis in Theorem 4.3. We finally present the influence of dimensionality of random Fourier features for our OPFES on 4 datasets in Figure 6, and trends are similar on other datasets. It is clear that our OPFES obtains stable performance if we set the dimensionality $d$ larger than 1000, while smaller dimensionality could yield heavy information loss.

## 6. Conclusion

This work focuses on two fundamental problems on feature evolvable learning. We propose the Kernel Ortho-Mapping (KOM) discrepancy to characterize intrinsic relationships

between two feature spaces via kernel functions, and then theoretically correlate it with optimal classifiers learned from different feature spaces. Based on this discrepancy, we develop one-pass algorithm for feature evolvable learning without storing the entire or partial training data. We verify the effectiveness of our porposed OPFES both theoretically and empirically. An interesting future work is to apply our KOM discrepancy to deep learning via neural tangent kernel, and exploit more effective tools to characterize the feature relationships for feature evolvable learning.

## Acknowledgements

The authors want to thank the reviewers for their helpful comments and suggestions. This research was supported by National Key R&D Program of China (2021ZD0112802) and NSFC (62376119).

## Impact Statement

This paper presents work whose goal is to advance the field of Machine Learning especially Feature Evolvable Learning. There are many potential societal consequences of our work, none which we feel must be specifically highlighted here.

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

# A. Proofs for Kernel Ortho-Mapping Discrepancy

## A.1. Proof of Lemma 3.2

For Frobenius norm and orthogonal matrix $\mathbf{U} \in \mathcal{U}_n$, we have

$$
\begin{aligned}
\left\| \sqrt{\mathbf{K}^{[1]}}\mathbf{U} - \sqrt{\mathbf{K}^{[2]}} \right\|_F^2 &= \mathrm{Tr}\left( \sqrt{\mathbf{K}^{[1]}}\mathbf{U} - \sqrt{\mathbf{K}^{[2]}} \right)\left( \sqrt{\mathbf{K}^{[1]}}\mathbf{U} - \sqrt{\mathbf{K}^{[2]}} \right)^\top \\
&= \mathrm{Tr}(\mathbf{K}^{[1]}) + \mathrm{Tr}(\mathbf{K}^{[2]}) - 2\,\mathrm{Tr}\left( \mathbf{U}^\top \sqrt{\mathbf{K}^{[2]}}\sqrt{\mathbf{K}^{[1]}} \right) .
\end{aligned}
\tag{13}
$$

The minimization of Kernel Ortho-Mapping (KOM) (in Definition 3.1) is equivalent to the following optimization:

$$
\max_{\mathbf{U}\in\mathcal{U}_n} \left\{ \mathrm{Tr}\left( \mathbf{U}^\top \sqrt{\mathbf{K}^{[2]}}\sqrt{\mathbf{K}^{[1]}} \right) \right\} .
\tag{14}
$$

Denote by $\mathbf{X} = \sqrt{\mathbf{K}^{[2]}}\sqrt{\mathbf{K}^{[1]}}$ and $|\mathbf{X}| = \sqrt{\mathbf{X}^\top\mathbf{X}} = \sqrt{\sqrt{\mathbf{K}^{[1]}}\mathbf{K}^{[2]}\sqrt{\mathbf{K}^{[1]}}}$. There exists $\mathbf{V} \in \mathcal{U}_n$ such that $\mathbf{X} = \mathbf{V}|\mathbf{X}|$, which is a polar decomposition of $\mathbf{X}$. We then have

$$
\mathrm{Tr}\left( \mathbf{U}^\top \sqrt{\mathbf{K}^{[2]}}\sqrt{\mathbf{K}^{[1]}} \right) = \mathrm{Tr}\left( \mathbf{U}^\top\mathbf{X} \right) = \mathrm{Tr}\left( \mathbf{U}^\top\mathbf{V}|\mathbf{X}| \right) .
$$

Denote by $\mathbf{W} = \mathbf{U}^\top\mathbf{V}$. We take the eigen-decomposition $|\mathbf{X}| = \mathbf{P}\mathbf{D}\mathbf{P}^\top$ for orthogonal $\mathbf{P}$ and diagonal matrix $\mathbf{D}$ with non-negative elements from the semi-positive definiteness of $|\mathbf{X}|$. From the cyclic invariance of matrix trace, we have

$$
\mathrm{Tr}\left( \mathbf{U}^\top \sqrt{\mathbf{K}^{[2]}}\sqrt{\mathbf{K}^{[1]}} \right) = \mathrm{Tr}\left( \mathbf{W}\mathbf{P}\mathbf{D}\mathbf{P}^\top \right) = \mathrm{Tr}\left( \mathbf{D}\mathbf{P}^\top\mathbf{W}\mathbf{P} \right) ,
$$

Denote by $\hat{\mathbf{W}} = \mathbf{P}^\top\mathbf{W}\mathbf{P}$ another unitary matrix in $\mathcal{U}_n$. The optimization problem in Eqn. (14) can solved by

$$
\max_{\hat{\mathbf{W}}\in\mathcal{U}_n} \left\{ \mathrm{Tr}\left( \mathbf{D}\hat{\mathbf{W}} \right) \right\} = \max_{\hat{\mathbf{W}}\in\mathcal{U}_n} \left\{ \sum_{i=1}^{n} \mathbf{D}_{ii}\hat{\mathbf{W}}_{ii} \right\} = \mathrm{Tr}(\mathbf{D}) ,
$$

where the last equality holds from $\hat{\mathbf{W}} = \mathbf{I}_n$. From nuclear norm and matrix trace of positive semi-definite matrix, we have

$$
\mathrm{Tr}(\mathbf{D}) = \mathrm{Tr}\left( \sqrt{\sqrt{\mathbf{K}^{[1]}}\mathbf{K}^{[2]}\sqrt{\mathbf{K}^{[1]}}} \right) = \sum_{i=1}^{n} \sqrt{\sigma_i\left( \sqrt{\mathbf{K}^{[1]}}\mathbf{K}^{[2]}\sqrt{\mathbf{K}^{[1]}} \right)} = \sum_{i=1}^{n} \sigma_i(\sqrt{\mathbf{K}^{[1]}}\sqrt{\mathbf{K}^{[2]}}) = \left\| \sqrt{\mathbf{K}^{[1]}}\sqrt{\mathbf{K}^{[2]}} \right\|_* ,
$$

which completes the proof by combining with Eqn. (13). $\qquad\square$

## A.2. Proof for Theorem 3.3.

We begin with the empirical feature mapping as follows:

**Lemma A.1.** *Let $\mathbf{K}$ be the Gram matrix w.r.t. kernel $\mathcal{K}$ and sample $S = \{\boldsymbol{x}_1, \cdots, \boldsymbol{x}_n\}$, and consider eigen decomposition $\mathbf{K} = \mathbf{S}\mathbf{D}\mathbf{S}^\top$. For kernel classifier $h(\boldsymbol{x}) = \sum_{j=1}^{n} \alpha_i\mathcal{K}(\boldsymbol{x}_j, \boldsymbol{x})$, we have*

$$
h(\boldsymbol{x}_i) = \left\langle \boldsymbol{w}, \mathbf{S}\sqrt{\mathbf{D}}\mathbf{S}^\top\boldsymbol{e}_i \right\rangle \quad \text{with} \quad \boldsymbol{w} = \mathbf{S}\sqrt{\mathbf{D}}\mathbf{S}^\top \sum_{j=1}^{n} \alpha_j\boldsymbol{e}_i ,
$$

*where $\boldsymbol{e}_i$ denotes a unit vector of the $i$-th element being $1$.*

*Proof.* From eigen decomposition $\mathbf{K} = \mathbf{S}\mathbf{D}\mathbf{S}^\top$, we have

$$
\mathcal{K}(\boldsymbol{x}_i, \boldsymbol{x}_j) = \mathbf{K}_{ij} = \left( \mathbf{S}\mathbf{D}\mathbf{S}^\top \right)_{ij} = \left\langle \mathbf{S}\sqrt{\mathbf{D}}\mathbf{S}^\top\boldsymbol{e}_i, \mathbf{S}\sqrt{\mathbf{D}}\mathbf{S}^\top\boldsymbol{e}_j \right\rangle ,
$$

and this gives one data-dependent feature mapping of $\mathcal{K}$ as $\hat{\boldsymbol{\varphi}} : \boldsymbol{x}_i \mapsto \mathbf{S}\sqrt{\mathbf{D}}\mathbf{S}^\top\boldsymbol{e}_i$ (Schölkopf & Smola, 2002). For kernel classifier $h(\boldsymbol{x}) = \sum_{j=1}^{n} \alpha_{1,j}\mathcal{K}(\boldsymbol{x}_j, \boldsymbol{x})$, we have

$$
h(\boldsymbol{x}_i) = \sum_{j=1}^{n} \alpha_{1,j}\mathcal{K}(\boldsymbol{x}_i, \boldsymbol{x}_j) = \sum_{j=1}^{n} \alpha_{1,j}\langle \hat{\boldsymbol{\varphi}}(\boldsymbol{x}_j), \hat{\boldsymbol{\varphi}}(\boldsymbol{x}_i) \rangle = \left\langle \mathbf{S}\sqrt{\mathbf{D}}\mathbf{S}^\top \sum_{j=1}^{n} \alpha_{1,j}\boldsymbol{e}_j, \mathbf{S}\sqrt{\mathbf{D}}\mathbf{S}^\top\boldsymbol{e}_i \right\rangle = \langle \boldsymbol{w}, \mathbf{S}\sqrt{\mathbf{D}}\mathbf{S}^\top\boldsymbol{e}_i \rangle,
$$

which completes the proof. $\qquad\square$

***Proof of Theorem 3.3.*** From Lemma A.1, the kernel learning problem in Eqn. (3) can be equivalently linearized as

$$\boldsymbol{w}_*^{[k]} \in \underset{\boldsymbol{w}^{[k]} \in \mathbb{R}^n}{\arg\min} \left\{ \hat{R}^{[k]}(\boldsymbol{w}^{[k]}) + \frac{\lambda}{2} \left\| \boldsymbol{w}^{[k]} \right\|_2^2 \right\}, \quad \text{for} \quad k = 1, 2 \,, \tag{15}$$

where

$$\hat{R}^{[k]}(\boldsymbol{w}^{[k]}) = \frac{1}{n} \sum_{i=1}^{n} \max \left\{ 0, 1 - y_i \left\langle \boldsymbol{w}^{[k]}, \hat{\boldsymbol{\varphi}}^{[k]}(\boldsymbol{x}_i^{[k]}) \right\rangle \right\} \,,$$

and $\hat{\boldsymbol{\varphi}}^{[k]}$ are the empirical kernel mappings of $\mathcal{K}^{[k]}$. For two classifiers and from strong convexity of Eqn. (15), we have

$$\frac{\lambda}{2} \left\| \boldsymbol{w}_*^{[1]} \right\|_2^2 + \hat{R}^{[1]}(\boldsymbol{w}_*^{[1]}) + \frac{\lambda}{2} \left\| \boldsymbol{w}_*^{[1]} - \boldsymbol{w}_*^{[2]} \right\|_2^2 \leq \hat{R}^{[1]}(\boldsymbol{w}_*^{[2]}) + \frac{\lambda}{2} \left\| \boldsymbol{w}_*^{[2]} \right\|_2^2 \,,$$

$$\frac{\lambda}{2} \left\| \boldsymbol{w}_*^{[2]} \right\|_2^2 + \hat{R}^{[2]}(\boldsymbol{w}_*^{[2]}) + \frac{\lambda}{2} \left\| \boldsymbol{w}_*^{[1]} - \boldsymbol{w}_*^{[2]} \right\|_2^2 \leq \hat{R}^{[2]}(\boldsymbol{w}_*^{[1]}) + \frac{\lambda}{2} \left\| \boldsymbol{w}_*^{[1]} \right\|_2^2 \,.$$

This holds that, from 1-Lipschitz continuous hinge loss and Cauchy-Schwarz inequality,

$$\left\| \boldsymbol{w}_*^{[1]} - \boldsymbol{w}_*^{[2]} \right\|_2^2 \leq \frac{1}{\lambda} \left( \left( \hat{R}^{[1]}(\boldsymbol{w}_*^{[2]}) - \hat{R}^{[2]}(\boldsymbol{w}_*^{[2]}) \right) + \left( \hat{R}^{[2]}(\boldsymbol{w}_*^{[1]}) - \hat{R}^{[1]}(\boldsymbol{w}_*^{[1]}) \right) \right)$$

$$= \frac{1}{n\lambda} \sum_{i=1}^{n} \left( \ell \left( \left\langle \boldsymbol{w}_*^{[2]}, \hat{\boldsymbol{\varphi}}^{[1]}(\boldsymbol{x}_i^{[1]}) \right\rangle, y_i \right) - \ell \left( \left\langle \boldsymbol{w}_*^{[2]}, \hat{\boldsymbol{\varphi}}_2(\boldsymbol{x}_i^{[2]}) \right\rangle, y_i \right) \right)$$

$$+ \frac{1}{n\lambda} \sum_{i=1}^{n} \left( \ell \left( \left\langle \boldsymbol{w}_*^{[1]}, \hat{\boldsymbol{\varphi}}^{[2]}(\boldsymbol{x}_i^{[2]}) \right\rangle, y_i \right) - \ell \left( \left\langle \boldsymbol{w}_*^{[1]}, \hat{\boldsymbol{\varphi}}^{[1]}(\boldsymbol{x}_i^{[1]}) \right\rangle, y_i \right) \right)$$

$$\leq \frac{1}{n\lambda} \left( \left\| \boldsymbol{w}_*^{[1]} \right\|_2 + \left\| \boldsymbol{w}_*^{[2]} \right\|_2 \right) \sum_{i=1}^{n} \left\| \hat{\boldsymbol{\varphi}}^{[1]}(\boldsymbol{x}_i^{[1]}) - \hat{\boldsymbol{\varphi}}^{[2]}(\boldsymbol{x}_i^{[2]}) \right\|_2 \,. \tag{16}$$

For $\|\boldsymbol{w}_*^{[1]}\|_2 + \|\boldsymbol{w}_*^{[2]}\|_2$, we have the following constraints on $\boldsymbol{w}_*^{[k]}$ from the KKT condition of Eqn. (15)

$$\boldsymbol{w}_*^{[k]} \in \left\{ -\frac{1}{n\lambda} \sum_{i=1}^{n} \boldsymbol{g}_i : \boldsymbol{g}_i \in \partial \max \left\{ 0, 1 - y_i \left\langle \boldsymbol{w}_*^{[k]}, \hat{\boldsymbol{\varphi}}^{[k]}(\boldsymbol{x}_i^{[k]}) \right\rangle \right\}, i \in [n] \right\} \quad \text{for} \quad k = 1, 2 \,,$$

where $\partial(\cdot)$ is the sub-gradient operator. We can upper bound $\|\boldsymbol{w}^{[1]*}\|_2 + \|\boldsymbol{w}^{[2]*}\|_2 \leq 2r/\lambda$ from

$$\left\| \boldsymbol{w}_*^{[k]} \right\|_2 \leq \max \left\{ \left\| -\frac{\sum_{i=1}^{n} \boldsymbol{g}_i}{n\lambda} \right\|_2 : \boldsymbol{g}_i \in \partial \max \left\{ 0, 1 - y_i \left\langle \boldsymbol{w}_k^*, \hat{\boldsymbol{\varphi}}_k(\boldsymbol{x}_i^{[k]}) \right\rangle \right\} \right\} \leq \frac{1}{n\lambda} \sum_{i=1}^{n} \left\| y_i \hat{\boldsymbol{\varphi}}^{[k]}(\boldsymbol{x}_i^{[k]}) \right\|_2 \leq \frac{r}{\lambda} \,.$$

It remains to bound $\sum_{i=1}^{n} \|\hat{\boldsymbol{\varphi}}^{[1]}(\boldsymbol{x}_i^{[1]}) - \hat{\boldsymbol{\varphi}}^{[2]}(\boldsymbol{x}_i^{[2]})\|_2$. We observe that the empirical kernel mapping is unitarily invariant in Lemma A.1, i.e., we have $\hat{\boldsymbol{\varphi}}(\boldsymbol{x}_i)^\top \hat{\boldsymbol{\varphi}}(\boldsymbol{x}_j) = (\mathbf{U}\hat{\boldsymbol{\varphi}}(\boldsymbol{x}_i))^\top (\mathbf{U}\hat{\boldsymbol{\varphi}}(\boldsymbol{x}_j))$ for $\mathbf{U} \in \mathcal{U}_n$ and $i, j \in [n]$, and the $i$-th column of $\mathbf{U}\sqrt{\mathbf{K}}$ is also a legal empirical kernel mapping of each $\boldsymbol{x}_i$. This follows that, for KOM discrepancy,

$$\sum_{i=1}^{n} \left\| \hat{\boldsymbol{\varphi}}^{[1]}(\boldsymbol{x}_i^{[1]}) - \hat{\boldsymbol{\varphi}}^{[2]}(\boldsymbol{x}_i^{[2]}) \right\|_2 \leq \sqrt{n} \min_{\mathbf{U}_1, \mathbf{U}_2 \in \mathcal{U}_n} \left\{ \left\| \sqrt{\mathbf{K}^{[1]}} \mathbf{U}_1 - \sqrt{\mathbf{K}^{[2]}} \mathbf{U}_2 \right\|_F \right\} = \sqrt{n} \hat{\mathcal{E}}(S_n, \mathcal{K}^{[1]}, \mathcal{K}^{[2]}) \,,$$

from the unitary invariance of Frobenius norm. From Eqn. (16), we have

$$\left\| \boldsymbol{w}_*^{[1]} - \boldsymbol{w}_*^{[2]} \right\|_2^2 \leq \frac{2r \hat{\mathcal{E}}(S_n, \mathcal{K}^{[1]}, \mathcal{K}^{[2]})}{\lambda^2} \,,$$

and we bound the difference between two optimal classifiers

$$
\begin{aligned}
& \left| h_*^{[1]}(\boldsymbol{x}_i^{[1]}) - h_*^{[2]}(\boldsymbol{x}_i^{[2]}) \right| \\
&= \left| \left\langle \boldsymbol{w}_*^{[1]}, \hat{\boldsymbol{\varphi}}^{[1]}(\boldsymbol{x}_i^{[1]}) \right\rangle - \left\langle \boldsymbol{w}_*^{[2]}, \hat{\boldsymbol{\varphi}}^{[2]}(\boldsymbol{x}_i^{[2]}) \right\rangle \right| \\
&\leq \left| \left\langle \boldsymbol{w}_*^{[1]}, \hat{\boldsymbol{\varphi}}^{[1]}(\boldsymbol{x}_i^{[1]}) \right\rangle - \left\langle \boldsymbol{w}_*^{[2]}, \hat{\boldsymbol{\varphi}}^{[1]}(\boldsymbol{x}_i^{[1]}) \right\rangle \right| + \left| \left\langle \boldsymbol{w}_*^{[2]}, \hat{\boldsymbol{\varphi}}^{[1]}(\boldsymbol{x}_i^{[1]}) \right\rangle - \left\langle \boldsymbol{w}_*^{[2]}, \hat{\boldsymbol{\varphi}}^{[2]}(\boldsymbol{x}_i^{[2]}) \right\rangle \right| \\
&\leq \left\| \hat{\boldsymbol{\varphi}}^{[1]}(\boldsymbol{x}_i^{[1]}) \right\|_2 \cdot \left\| \boldsymbol{w}_*^{[1]} - \boldsymbol{w}_*^{[2]} \right\|_2 + \left\| \boldsymbol{w}_*^{[2]} \right\|_2 \cdot \left\| \hat{\boldsymbol{\varphi}}^{[1]}(\boldsymbol{x}_i^{[1]}) - \hat{\boldsymbol{\varphi}}^{[2]}(\boldsymbol{x}_i^{[2]}) \right\|_2 \\
&\leq \frac{r}{\lambda} \sqrt{2r\hat{\mathcal{E}}(S_n, \mathcal{K}^{[1]}, \mathcal{K}^{[2]})} + \frac{r}{\lambda} \left\| \hat{\boldsymbol{\varphi}}^{[1]}(\boldsymbol{x}_i^{[1]}) - \hat{\boldsymbol{\varphi}}^{[2]}(\boldsymbol{x}_i^{[2]}) \right\|_2 ,
\end{aligned}
$$

from equivalent linearization of kernel classifier and bounded norm for optimal classifier. We finally have

$$
\begin{aligned}
\hat{\rho}_{S_n}(h_*^{[1]}, h_*^{[2]}) &\leq \frac{r}{\lambda} \sqrt{2r\hat{\mathcal{E}}(S_n, \mathcal{K}^{[1]}, \mathcal{K}^{[2]})} + \frac{r}{n\lambda} \sum_{i=1}^n \left\| \hat{\boldsymbol{\varphi}}^{[1]}(\boldsymbol{x}_i^{[1]}) - \hat{\boldsymbol{\varphi}}^{[2]}(\boldsymbol{x}_i^{[2]}) \right\|_2 \\
&\leq \frac{r}{\lambda} \left( \hat{\mathcal{E}}(S_n, \mathcal{K}^{[1]}, \mathcal{K}^{[2]}) + \sqrt{2r\hat{\mathcal{E}}(S_n, \mathcal{K}^{[1]}, \mathcal{K}^{[2]})} \right) ,
\end{aligned}
$$

which completes the proof. $\qquad\square$

### A.3. Proof of Theorem 3.4

We begin with some useful lemmas as follows:

**Lemma A.2.** *For a distribution $\mathcal{D} \sim \mathcal{X}^{[1]} \times \mathcal{X}^{[2]}$, we have*

$$
\begin{aligned}
\mathcal{E}(\mathcal{D}, \mathcal{K}^{[1]}, \mathcal{K}^{[2]}) &= \min_{\mathbf{U} \in \mathcal{U}} \left\{ \sqrt{\mathbb{E}_{\mathcal{D}} \left[ \| \mathbf{U}\boldsymbol{\varphi}^{[1]}(\boldsymbol{x}^{[1]}) - \boldsymbol{\varphi}^{[2]}(\boldsymbol{x}^{[2]}) \|_{HS}^2 \right]} \right\} \\
&= \left( \mathbb{E}_{\mathcal{D}} \left[ \mathcal{K}^{[1]}(\boldsymbol{x}^{[1]}, \boldsymbol{x}^{[1]}) + \mathcal{K}^{[2]}(\boldsymbol{x}^{[2]}, \boldsymbol{x}^{[2]}) \right] - 2 \left\| \mathbb{E}_{\mathcal{D}} \left[ \boldsymbol{\varphi}^{[1]}(\boldsymbol{x}^{[1]}) \boldsymbol{\varphi}^{[2]}(\boldsymbol{x}^{[2]})^\top \right] \right\|_* \right)^{1/2} .
\end{aligned}
$$

*Proof.* For Hilbert-Schmidt norm, we have

$$
\begin{aligned}
& \mathbb{E}_{\mathcal{D}} \left[ \left\| \mathbf{U}\boldsymbol{\varphi}^{[1]}(\boldsymbol{x}^{[1]}) - \boldsymbol{\varphi}^{[2]}(\boldsymbol{x}^{[2]}) \right\|_{HS}^2 \right] \\
&= \mathbb{E}_{\mathcal{D}} \left[ \mathcal{K}^{[1]}(\boldsymbol{x}^{[1]}, \boldsymbol{x}^{[1]}) + \mathcal{K}^{[2]}(\boldsymbol{x}^{[2]}, \boldsymbol{x}^{[2]}) \right] - 2\mathbb{E}_{\mathcal{D}} \left[ \boldsymbol{\varphi}^{[2]}(\boldsymbol{x}^{[2]})^\top \mathbf{U}\boldsymbol{\varphi}^{[1]} \right] \\
&= \mathbb{E}_{\mathcal{D}} \left[ \mathcal{K}^{[1]}(\boldsymbol{x}^{[1]}, \boldsymbol{x}^{[1]}) + \mathcal{K}^{[2]}(\boldsymbol{x}^{[2]}, \boldsymbol{x}^{[2]}) \right] - 2\mathbb{E}_{\mathcal{D}} \left[ \mathrm{Tr} \left( \boldsymbol{\varphi}^{[2]}(\boldsymbol{x}^{[2]})^\top \mathbf{U}\boldsymbol{\varphi}^{[1]} \right) \right] \\
&= \mathbb{E}_{\mathcal{D}} \left[ \mathcal{K}^{[1]}(\boldsymbol{x}^{[1]}, \boldsymbol{x}^{[1]}) + \mathcal{K}^{[2]}(\boldsymbol{x}^{[2]}, \boldsymbol{x}^{[2]}) \right] - 2\,\mathrm{Tr} \left( \mathbf{U}\mathbb{E}_{\mathcal{D}} \left[ \boldsymbol{\varphi}^{[1]}(\boldsymbol{x}^{[1]}) \boldsymbol{\varphi}^{[2]}(\boldsymbol{x}^{[2]})^\top \right] \right) ,
\end{aligned}
$$

where the last inequality holds from the linearity of operator trace w.r.t expectation, and independence between $\mathbf{U}$ and $\mathcal{D}$. We complete the proof from the similar derivations as in the proof of Lemma 3.2. $\qquad\square$

**Lemma A.3** (Perturbation bound for singular values (Bhatia, 2013))**.** *For $n \times n$ real matrices $\mathbf{A}$ and $\mathbf{B}$, we have*

$$
\sum_{i=1}^n |\sigma_i(\mathbf{A}) - \sigma_i(\mathbf{B})| \leq \| \mathbf{A} - \mathbf{B} \|_* ,
$$

*where $\sigma_i(\mathbf{A})$ and $\sigma_i(\mathbf{B})$ are their respective $i$-th singular values, i.e., $\sigma_1(\mathbf{A}) \geq \cdots \geq \sigma_n(\mathbf{A})$ and $\sigma_1(\mathbf{B}) \geq \cdots \geq \sigma_n(\mathbf{B})$.*

**Lemma A.4** (McDiarmid's inequality (McDiarmid et al., 1989))**.** *Let $X_1, X_2, \cdots, X_n$ be independent random variables taking values in a set $A$, and $f : A^n \to \mathbb{R}$ satisfies*

$$
\sup_{X_1, X_2, \cdots, X_n, X_i' \in A} |f(X_1, X_2, \cdots, X_n) - f(X_1, X_2, \cdots, X_{i-1}, X_i', X_{i+1}, X_n)| \leq c_i
$$

*for every $i \in [n]$. Then, for $t > 0$, we have*

$$\Pr\left[f(X_1, \cdots X_i, \cdots, X_n) - \mathbb{E}[f(X_1, \cdots, X_i', \cdots, X_n)] \geq t\right] \leq \exp\left(-\frac{2t^2}{\sum_{i=1}^n c_i^2}\right).$$

**Lemma A.5.** *Let $\mathcal{K}^{[1]}$ and $\mathcal{K}^{[2]}$ be two kernels bounded by $r^2$. For samples $S_n = \{(\boldsymbol{x}_1^{[1]}, \boldsymbol{x}_1^{[2]}), \cdots, (\boldsymbol{x}_n^{[1]}, \boldsymbol{x}_n^{[2]})\}$ and $S_n' = S_n \backslash \{(\boldsymbol{x}_k^{[1]}, \boldsymbol{x}_k^{[2]})\} \cup \{(\boldsymbol{x}_k^{[1]'}, \boldsymbol{x}_k^{[2]'})\}$ $(k \in [n])$, we have*

$$\left| \left\| \sqrt{\mathbf{K}^{[1]}} \sqrt{\mathbf{K}^{[2]}} \right\|_* - \left\| \sqrt{\mathbf{K}^{[1]'}} \sqrt{\mathbf{K}^{[2]'}} \right\|_* \right| \leq 2r^2 ,$$

*where $\mathbf{K}^{[1]}$ and $\mathbf{K}^{[1]'}$ are Gram matrices w.r.t kernel $\mathcal{K}^{[1]}$ over $S_n$ and $S_n'$, respectively, and define $\mathbf{K}^{[2]}$ and $\mathbf{K}^{[2]'}$ similarly.*

*Proof.* From Lemma A.3, we have

$$
\begin{aligned}
&\left| \left\| \sqrt{\mathbf{K}^{[1]}} \sqrt{\mathbf{K}^{[2]}} \right\|_* - \left\| \sqrt{\mathbf{K}^{[1]'}} \sqrt{\mathbf{K}^{[2]'}} \right\|_* \right| \\
\leq\;& \left| \left\| \sqrt{\mathbf{K}^{[1]}} \sqrt{\mathbf{K}^{[2]}} \right\|_* - \left\| \sqrt{\mathbf{K}^{[1]'}} \sqrt{\mathbf{K}^{[2]}} \right\|_* \right| + \left| \left\| \sqrt{\mathbf{K}^{[1]'}} \sqrt{\mathbf{K}^{[2]}} \right\|_* - \left\| \sqrt{\mathbf{K}^{[1]'}} \sqrt{\mathbf{K}^{[2]'}} \right\|_* \right| \\
=\;& \left| \sum_{i=1}^n \left| \sigma_i(\sqrt{\mathbf{K}^{[1]}} \sqrt{\mathbf{K}^{[2]}}) \right| - \left| \sigma_i(\sqrt{\mathbf{K}^{[1]'}} \sqrt{\mathbf{K}^{[2]}}) \right| \right| + \left| \sum_{i=1}^n \left| \sigma_i(\sqrt{\mathbf{K}^{[1]'}} \sqrt{\mathbf{K}^{[2]}}) \right| - \left| \sigma_i(\sqrt{\mathbf{K}^{[1]'}} \sqrt{\mathbf{K}^{[2]'}}) \right| \right| \\
\leq\;& \min_{\mathbf{U}, \mathbf{V} \in \mathcal{U}_n} \left\{ \sum_{i=1}^n \left| \sigma_i(\sqrt{\mathbf{K}^{[1]}} \sqrt{\mathbf{K}^{[2]}}) - \sigma_i(\mathbf{U}\sqrt{\mathbf{K}^{[1]'}} \sqrt{\mathbf{K}^{[2]}}) \right| + \left| \sigma_i(\sqrt{\mathbf{K}^{[1]'}} \sqrt{\mathbf{K}^{[2]}}) - \sigma_i(\mathbf{V}\sqrt{\mathbf{K}^{[1]'}} \sqrt{\mathbf{K}^{[2]'}}) \right| \right\} \\
\leq\;& \min_{\mathbf{U} \in \mathcal{U}_n} \left\{ \left\| \left( \sqrt{\mathbf{K}^{[1]}} - \mathbf{U}\sqrt{\mathbf{K}^{[1]'}} \right) \sqrt{\mathbf{K}^{[2]}} \right\|_* \right\} + \min_{\mathbf{V} \in \mathcal{U}_n} \left\{ \left\| \left( \sqrt{\mathbf{K}^{[2]}} - \mathbf{V}\sqrt{\mathbf{K}^{[2]'}} \right) \sqrt{\mathbf{K}^{[1]'}} \right\|_* \right\} . \quad (17)
\end{aligned}
$$

We now prove that there is an $\hat{\mathbf{U}} \in \mathcal{U}_n$ such that the difference between $\sqrt{\mathbf{K}^{[1]}}$ and $\hat{\mathbf{U}}\sqrt{\mathbf{K}^{[1]'}}$ lies only in the $k$-th column. Denote by $\boldsymbol{v} = \left[ \hat{\mathbf{U}}\sqrt{\mathbf{K}^{[1]'}} \right]_k$, i.e., the $k$-th column of $\hat{\mathbf{U}}\sqrt{\mathbf{K}^{[1]'}}$. The existence of $\hat{\mathbf{U}} \in \mathcal{U}_n$ is equivalent to solving the following underdetermined system of $n-1$ equations with $n$ variables

$$\left\langle \boldsymbol{v}, \left( \sqrt{\mathbf{K}^{[1]}} \right)_i \right\rangle = \frac{\mathcal{K}^{[1]}\left( \boldsymbol{x}_i^{[1]}, \boldsymbol{x}_k^{[1]'} \right)}{\left( \mathcal{K}^{[1]}\left( \boldsymbol{x}_k^{[1]'}, \boldsymbol{x}_k^{[1]'} \right) \right)^{1/2}} \quad \text{for} \quad i \in [n] \backslash \{k\} . \quad (18)$$

From the full-rank Gram matrix $\mathbf{K}^{[1]}$, there exists a solution $\boldsymbol{v}_0$ for the system in Eqn. (18) because of $n-1$ equations with $n$ variables. By setting $\boldsymbol{v} = \boldsymbol{v}_0$, we have an $\hat{\mathbf{U}} \in \mathcal{U}_n$ such that the difference between $\sqrt{\mathbf{K}^{[1]}}$ and $\hat{\mathbf{U}}\sqrt{\mathbf{K}^{[1]'}}$ lies only in the $k$-th column. This follows that, from the nuclear norm of the rank-1 matrix,

$$\min_{\mathbf{U} \in \mathcal{U}_n} \left\{ \left\| \left( \sqrt{\mathbf{K}^{[1]}} - \mathbf{U}\sqrt{\mathbf{K}^{[1]'}} \right) \sqrt{\mathbf{K}^{[2]}} \right\|_* \right\} \leq \left\| \left( [\sqrt{\mathbf{K}^{[1]}}]_k - [\hat{\mathbf{U}}\sqrt{\mathbf{K}^{[1]'}}]_k \right) \left[ \sqrt{\mathbf{K}^{[2]}} \right]_k^\top \right\|_* \leq 2r^2 .$$

This completes the proof from Eqn. (17) and similar analysis for $\min_{\mathbf{V} \in \mathcal{U}_n} \left\{ \left\| \left( \sqrt{\mathbf{K}^{[2]}} - \mathbf{V}\sqrt{\mathbf{K}^{[2]'}} \right) \sqrt{\mathbf{K}^{[1]'}} \right\|_* \right\}$. $\qquad \square$

**Lemma A.6** (Non-commutative Khintchine inequality, (Vershynin, 2018)). *For independent Rademacher random variables $\epsilon_1, \cdots, \epsilon_n$ and for real matrices $\mathbf{X}_1, \cdots, \mathbf{X}_n$ of the same size, we have*

$$\mathbb{E}\left[ \left\| \sum_{i=1}^n \epsilon_i \mathbf{X}_i \right\|_* \right] \leq C \max \left\{ \left\| \sqrt{\sum_{i=1}^n \mathbf{X}_i \mathbf{X}_i^\top} \right\|_*, \left\| \sqrt{\sum_{i=1}^n \mathbf{X}_i^\top \mathbf{X}_i} \right\|_* \right\} \quad \text{for some positive constant } C.$$

**Lemma A.7.** *For independent Rademacher random variables $\epsilon_1, \cdots, \epsilon_n$ and for real matrices $\mathbf{X}_1, \cdots, \mathbf{X}_n$ of the same size with $\mathbb{E}[\|\mathbf{X}_i\|_*] < \infty$, we have*

$$\mathbb{E}\left[ \left\| \sum_{i=1}^n (\mathbf{X}_i - \mathbb{E}[\mathbf{X}_i]) \right\|_* \right] \leq 2\mathbb{E}\left[ \left\| \sum_{i=1}^n \epsilon_i \mathbf{X}_i \right\|_* \right] .$$

*Proof.* Let $\{\mathbf{X}_1', \cdots, \mathbf{X}_n'\}$ denote an independent copy of the sequence $\{\mathbf{X}_1, \cdots, \mathbf{X}_n\}$. From i.i.d. assumption and Jensen's inequality, we have

$$
\mathop{\mathbb{E}}_{\mathbf{X}_1,\cdots,\mathbf{X}_n}\left[\left\|\sum_{i=1}^{n}(\mathbf{X}_i - \mathop{\mathbb{E}}_{\mathbf{X}_1,\cdots,\mathbf{X}_n}[\mathbf{X}_i])\right\|_*\right] = \mathop{\mathbb{E}}_{\mathbf{X}_1,\cdots,\mathbf{X}_n}\left[\left\|\sum_{i=1}^{n}(\mathbf{X}_i - \mathop{\mathbb{E}}_{\mathbf{X}_1,\cdots,\mathbf{X}_n}[\mathbf{X}_i]) - \mathbb{E}'\left[\mathbf{X}_i' - \mathop{\mathbb{E}}_{\mathbf{X}_1,\cdots,\mathbf{X}_n}[\mathbf{X}_i]\right]\right\|_*\right]
$$

$$
\leq \mathop{\mathbb{E}}_{\mathbf{X}_1,\cdots,\mathbf{X}_n}\left[\mathop{\mathbb{E}}_{\mathbf{X}_1',\cdots,\mathbf{X}_n'}\left[\left\|\sum_{i=1}^{n}(\mathbf{X}_i - \mathbb{E}[\mathbf{X}_i]) - (\mathbf{X}_i' - \mathbb{E}[\mathbf{X}_i])\right\|_*\right]\right] = \mathop{\mathbb{E}}_{\mathbf{X}_1,\cdots,\mathbf{X}_n,\mathbf{X}_1',\cdots,\mathbf{X}_n'}\left[\left\|\sum_{i=1}^{n}(\mathbf{X}_i - \mathbf{X}_i')\right\|_*\right] .
$$

We also have, from the triangle inequality of nuclear norm and symmetry of Rademacher random variables,

$$
\mathop{\mathbb{E}}_{\mathbf{X}_1,\cdots,\mathbf{X}_n,\mathbf{X}_1',\cdots,\mathbf{X}_n'}\left[\left\|\sum_{i=1}^{n}(\mathbf{X}_i - \mathbf{X}_i')\right\|_*\right] = \mathop{\mathbb{E}}_{\mathbf{X}_1,\cdots,\mathbf{X}_n,\mathbf{X}_1',\cdots,\mathbf{X}_n'}\left[\left\|\sum_{i=1}^{n}\epsilon_i(\mathbf{X}_i - \mathbf{X}_i')\right\|_*\right]
$$

$$
\leq \mathop{\mathbb{E}}_{\mathbf{X}_1,\cdots,\mathbf{X}_n}\left[\left\|\sum_{i=1}^{n}\epsilon_i\mathbf{X}_i\right\|_*\right] + \mathop{\mathbb{E}}_{\mathbf{X}_1',\cdots,\mathbf{X}_n'}\left[\left\|\sum_{i=1}^{n}-\epsilon_i\mathbf{X}_i'\right\|_*\right] = 2\mathop{\mathbb{E}}_{\mathbf{X}_1,\cdots,\mathbf{X}_n}\left[\left\|\sum_{i=1}^{n}\epsilon_i\mathbf{X}_i\right\|_*\right] ,
$$

which completes the proof. $\qquad\square$

***Proof of Theorem 3.4.*** By triangle inequality, we have

$$
\left|\hat{\mathcal{E}}(S_n, \mathcal{K}^{[1]}, \mathcal{K}^{[2]}) - \mathcal{E}(\mathcal{D}, \mathcal{K}^{[1]}, \mathcal{K}^{[2]})\right|
$$

$$
\leq \underbrace{\left|\hat{\mathcal{E}}(S_n, \mathcal{K}^{[1]}, \mathcal{K}^{[2]}) - \mathbb{E}_{S_n}\left[\hat{\mathcal{E}}(S_n, \mathcal{K}_1, \mathcal{K}_2)\right]\right|}_{\text{Concentration analysis}} + \underbrace{\left|\mathbb{E}_{S_n}\left[\hat{\mathcal{E}}(S_n, \mathcal{K}_1, \mathcal{K}_2)\right] - \mathcal{E}(\mathcal{D}, \mathcal{K}^{[1]}, \mathcal{K}^{[2]})\right|}_{\text{Random matrix analysis}} . \quad (19)
$$

For sample $S_n'$ with the $k$-th instance replaced by $(\boldsymbol{x}_k^{[1]'}, \boldsymbol{x}_k^{[2]'})$ from $S_n$, we have

$$
n\left|\left(\hat{\mathcal{E}}(S_n, \mathcal{K}^{[1]}, \mathcal{K}^{[2]})\right)^2 - \left(\hat{\mathcal{E}}(S_n', \mathcal{K}^{[1]}, \mathcal{K}^{[2]})\right)^2\right|
$$

$$
\leq \left|\Delta\left(\left\|\sqrt{\mathbf{K}^{[1]}}\sqrt{\mathbf{K}^{[2]}}\right\|_*\right)\right| + \left|\mathcal{K}^{[1]}(\boldsymbol{x}_k^{[1]}, \boldsymbol{x}_k^{[1]}) - \mathcal{K}^{[1]'}(\boldsymbol{x}_k^{[1]'}, \boldsymbol{x}_k^{[1]'})\right| + \left|\mathcal{K}^{[2]}(\boldsymbol{x}_k^{[2]}, \boldsymbol{x}_k^{[2]}) - \mathcal{K}^{[2]'}(\boldsymbol{x}_k^{[2]'}, \boldsymbol{x}_k^{[2]'})\right| \leq 6r^2 ,
$$

from bounded kernel and Lemma A.5. Based on the McDiarmid's inequality (Lemma A.4), the following holds with probability at least $1 - \delta$,

$$
\left|\hat{\mathcal{E}}(S_n, \mathcal{K}^{[1]}, \mathcal{K}^{[2]}) - \mathbb{E}_{S_n}\left[\hat{\mathcal{E}}(S_n, \mathcal{K}^{[1]}, \mathcal{K}^{[2]})\right]\right| \leq 3r\sqrt{\frac{2}{n}\ln\frac{2}{\delta}} . \quad (20)
$$

For the second term in Eqn. (19), we consider the random matrix analysis with nuclear norm (Vershynin, 2018). Denote by $\boldsymbol{\Phi} = \mathbb{E}_{\mathcal{D}}[\boldsymbol{\varphi}^{[1]}(\boldsymbol{x}^{[1]})\boldsymbol{\varphi}^{[2]}(\boldsymbol{x}^{[2]})^\top]$ and operator $\mathbf{X}_i = (\boldsymbol{\varphi}^{[1]}(\boldsymbol{x}_i^{[1]})\boldsymbol{\varphi}^{[2]}(\boldsymbol{x}_i^{[2]})^\top - \boldsymbol{\Phi})/n$, and we have

$$
\mathbb{E}_{S_n}\left[\frac{1}{n}\operatorname{Tr}(\mathbf{K}^{[k]})\right] = \mathbb{E}_{\mathcal{D}}[\mathcal{K}^{[k]}(\boldsymbol{x}^{[k]}, \boldsymbol{x}^{[k]})] \quad \text{for} \quad k \in [2] .
$$

This follows that, from the linearity of expectation and Lemma A.3,

$$
\left|\left(\mathcal{E}(\mathcal{D}, \mathcal{K}^{[1]}, \mathcal{K}^{[2]})\right)^2 - \mathbb{E}_{S_n}\left[\left(\hat{\mathcal{E}}(S_n, \mathcal{K}^{[1]}, \mathcal{K}^{[2]})\right)^2\right]\right| = 2\left|\mathbb{E}_{S_n}\left[\left\|\frac{1}{n}\sum_{i=1}^{n}\boldsymbol{\varphi}^{[1]}(\boldsymbol{x}_i^{[1]})\boldsymbol{\varphi}^{[2]}(\boldsymbol{x}_i^{[2]})^\top\right\| - \|\boldsymbol{\Phi}\|_*\right]\right|
$$

$$
\leq 2\mathbb{E}_{S_n}\left[\left\|\frac{1}{n}\sum_{i=1}^{n}\boldsymbol{\varphi}^{[1]}(\boldsymbol{x}_i^{[1]})\boldsymbol{\varphi}^{[2]}(\boldsymbol{x}_i^{[2]})^\top - \boldsymbol{\Phi}\right\|_*\right] \leq 4\mathbb{E}\left[\left\|\sum_{i=1}^{n}\epsilon_i\mathbf{X}_i\right\|_*\right] ,
$$

where $\epsilon_1, \cdots, \epsilon_n$ are independent Rademacher random variables. From $\mathbf{X}_i = (\boldsymbol{\varphi}^{[1]}(\boldsymbol{x}_i^{[1]})\boldsymbol{\varphi}^{[2]}(\boldsymbol{x}_i^{[2]})^\top - \boldsymbol{\Phi})/n$, we have

$$\mathbf{X}_i\mathbf{X}_i^\top = \frac{\mathcal{K}^{[1]}(\boldsymbol{x}_i^{[1]},\boldsymbol{x}_i^{[1]}) \cdot \boldsymbol{\varphi}^{[2]}(\boldsymbol{x}_i^{[2]})\boldsymbol{\varphi}^{[2]}(\boldsymbol{x}_i^{[2]})^\top - \boldsymbol{\varphi}^{[1]}(\boldsymbol{x}_i^{[1]})\boldsymbol{\varphi}^{[2]}(\boldsymbol{x}_i^{[2]})^\top\boldsymbol{\Phi}^\top - \boldsymbol{\Phi}\boldsymbol{\varphi}^{[2]}(\boldsymbol{x}_i^{[2]})\boldsymbol{\varphi}^{[1]}(\boldsymbol{x}_i^{[1]})^\top + \boldsymbol{\Phi}\boldsymbol{\Phi}^\top}{n^2} \,.$$

This follows that, by the sub-additivity of square root w.r.t nuclear norm and some algebraic calculations,

$$\left\| \sqrt{\sum_{i=1}^n \mathbf{X}_i\mathbf{X}_i^\top} \right\|_* \le \frac{1}{n}\left\| \sqrt{\sum_{i=1}^n \mathcal{K}^{[1]}(\boldsymbol{x}_i^{[1]},\boldsymbol{x}_i^{[1]}) \cdot \boldsymbol{\varphi}^{[2]}(\boldsymbol{x}_i^{[2]})\boldsymbol{\varphi}^{[2]}(\boldsymbol{x}_i^{[2]})^\top} \right\|_*$$
$$+ \frac{\sqrt{2}}{n}\left\| \sqrt{\sum_{i=1}^n \boldsymbol{\Phi}\boldsymbol{\varphi}^{[2]}(\boldsymbol{x}_i^{[2]})\boldsymbol{\varphi}^{[1]}(\boldsymbol{x}_i^{[1]})^\top} \right\|_* + \frac{1}{n}\left\| \sqrt{\boldsymbol{\Phi}\boldsymbol{\Phi}^\top} \right\|_* \le \left( \frac{1+\sqrt{2}}{\sqrt{n}} + \frac{1}{n} \right) r^2 \,,$$

where we use

$$\|\sqrt{\boldsymbol{\Phi}\boldsymbol{\Phi}^\top}\|_* = \|\boldsymbol{\Phi}\|_* = \left\| \mathbb{E}_\mathcal{D}\left[ \boldsymbol{\varphi}^{[1]}(\boldsymbol{x}^{[1]})\boldsymbol{\varphi}^{[2]}(\boldsymbol{x}^{[2]})^\top \right] \right\|_* \le \mathbb{E}_\mathcal{D}\left[ \left\| \boldsymbol{\varphi}^{[1]}(\boldsymbol{x}^{[1]})\boldsymbol{\varphi}^{[2]}(\boldsymbol{x}^{[2]})^\top \right\|_* \right] \le r^2 \,;$$

and

$$\left\| \sqrt{\sum_{i=1}^n \boldsymbol{\Phi}\boldsymbol{\varphi}^{[2]}(\boldsymbol{x}_i^{[2]})\boldsymbol{\varphi}^{[1]}(\boldsymbol{x}_i^{[1]})^\top} \right\|_* \le r\left\| \sqrt{\sum_{i=1}^n \boldsymbol{\varphi}^{[1]}(\boldsymbol{x}_i^{[1]})\mathbb{E}_\mathcal{D}\left[ \boldsymbol{\varphi}^{[1]}(\boldsymbol{x}^{[1]}) \right]^\top} \right\|_* \le \sqrt{n}r^{\frac{3}{2}}\left\| \sqrt{\mathbb{E}_\mathcal{D}[\boldsymbol{\varphi}^{[1]}(\boldsymbol{x}^{[1]})]} \right\|_* \le \sqrt{n}r^2 \,;$$

and, from linearity of expectation and relationship between nuclear norm and operator norm for rank-1 operator,

$$\left\| \sqrt{\sum_{i=1}^n \mathcal{K}^{[1]}(\boldsymbol{x}_i^{[1]},\boldsymbol{x}_i^{[1]}) \cdot \boldsymbol{\varphi}^{[2]}(\boldsymbol{x}_i^{[2]})\boldsymbol{\varphi}^{[2]}(\boldsymbol{x}_i^{[2]})^\top} \right\|_* \le r\left\| \sqrt{\sum_{i=1}^n \boldsymbol{\varphi}^{[2]}(\boldsymbol{x}_i^{[2]})\boldsymbol{\varphi}^{[2]}(\boldsymbol{x}_i^{[2]})^\top} \right\|_* \le \sqrt{n}r^2 .$$

Similarly, we can present the same upper bounds for $\sum_{i=1}^n \mathbf{X}_i^\top\mathbf{X}_i$. This follows that, from Lemmas A.6 and A.7,

$$\left| \mathcal{E}(\mathcal{D},\mathcal{K}^{[1]},\mathcal{K}^{[2]}) - \mathbb{E}_{S_n}\left[ \hat{\mathcal{E}}(S_n,\mathcal{K}_1,\mathcal{K}_2) \right] \right| \le C\left( \frac{1+\sqrt{2}}{\sqrt{n}} + \frac{1}{n} \right) r \,. \tag{21}$$

We complete the proof from Eqns. (20) and (21), and triangle inequality. $\qquad\square$

### A.4. Proof of Lemma 3.5

From (Bhatia, 2013), we have $\|\sqrt{\mathbf{A}} - \sqrt{\mathbf{B}}\|_F \le \sqrt{\|\mathbf{A}-\mathbf{B}\|_*}$, and this follows that, for KOM discrepancy,

$$\hat{\mathcal{E}}(S_n,\mathcal{K}^{[1]},\mathcal{K}^{[2]}) \le \frac{1}{\sqrt{n}}\left\| \sqrt{\mathbf{K}^{[1]}} - \sqrt{\mathbf{K}^{[2]}} \right\|_F \le \sqrt{\frac{1}{n}\left\| \mathbf{K}^{[1]} - \mathbf{K}^{[2]} \right\|_*} \,.$$

We further have, for normalized kernel matrices,

$$\sqrt{\frac{1}{n}\left\| \mathbf{K}^{[1]} - \mathbf{K}^{[2]} \right\|_*} \le \sqrt{\sqrt{\frac{1}{n}}\left\| \mathbf{K}^{[1]} - \mathbf{K}^{[2]} \right\|_F}$$
$$= \sqrt[4]{\frac{\|\mathbf{K}^{[1]}\|_F\|\mathbf{K}^{[2]}\|_F}{n}\left( \frac{\|\mathbf{K}^{[1]}\|_F}{\|\mathbf{K}^{[2]}\|_F} + \frac{\|\mathbf{K}^{[2]}\|_F}{\|\mathbf{K}^{[1]}\|_F} - \frac{2\,\mathrm{Tr}(\mathbf{K}^{[1]}\mathbf{K}^{[2]})}{\|\mathbf{K}^{[1]}\|_F\|\mathbf{K}^{[2]}\|_F} \right)} \le r\sqrt[4]{2(1-\hat{A}(\mathbf{K}^{[1]},\mathbf{K}^{[2]}))} \,,$$

which completes the proof. $\qquad\square$

## A.5. Proof of Lemma 3.6

Denote by $d = \dim(\varphi^{[1]})$ for simplicity. For sample $S_n = \{\boldsymbol{x}_1, \cdots, \boldsymbol{x}_n\}$, we write

$$\mathbf{P}^{[1]} = \left(\boldsymbol{\varphi}^{[1]}(\boldsymbol{x}_i^{[1]})\right)_{i=1}^n \quad \text{and} \quad \hat{\mathbf{P}}^{[2]} = \left(\hat{\boldsymbol{\varphi}}^{[2]}(\boldsymbol{x}_i^{[1]})\right)_{i=1}^n ,$$

and we also have their respective Gram matrices $\mathbf{K}^{[1]} = \mathbf{P}^{[1]\top}\mathbf{P}^{[1]}$ and $\hat{\mathbf{K}}^{[2]} = \hat{\mathbf{P}}^{[2]\top}\hat{\mathbf{P}}^{[2]}$. It remains to prove

$$\min_{\mathbf{U}\in\mathcal{U}_n}\left\{\left\|\sqrt{\mathbf{K}^{[1]}}\mathbf{U} - \sqrt{\hat{\mathbf{K}}^{[2]}}\right\|_F\right\} \leq \left\|\mathbf{P}^{[1]} - \hat{\mathbf{P}}^{[2]}\right\|_F .$$

If $n \leq d$, then there exists a matrix $\mathbf{V} \in \mathcal{U}_n$, from Lemma A.1 and the unitary invariance of Frobenius norm, such that

$$\left\|\mathbf{P}^{[1]} - \hat{\mathbf{P}}^{[2]}\right\|_F = \left\|\mathbf{P}^{[1]}\mathbf{V} - \hat{\mathbf{P}}^{[2]}\mathbf{V}\right\|_F = \left\|\mathbf{P}^{[1]\top}\mathbf{V} - \sqrt{\hat{\mathbf{K}}^{[2]}}\right\|_F ,$$

We also have

$$\left\|\mathbf{P}^{[1]} - \hat{\mathbf{P}}^{[2]}\right\|_F \geq \min_{\mathbf{W}\in\mathcal{U}_n}\left\{\left\|\sqrt{\mathbf{K}^{[1]}}\mathbf{W} - \sqrt{\hat{\mathbf{K}}^{[2]}}\right\|_F\right\} ,$$

for some $\mathbf{W} \in \mathcal{U}_n$ with $\sqrt{\mathbf{K}^{[1]}}\mathbf{W} = \mathbf{P}^{[1]\top}\mathbf{V}$.

If $n > d$, then we have, similarly to the proof of Theorem A.2,

$$\left\|\mathbf{P}^{[1]} - \hat{\mathbf{P}}^{[2]}\right\|_F \geq \min_{\mathbf{W}\in\mathcal{U}_d}\left\{\left\|\mathbf{P}^{[1]\top}\mathbf{W} - \hat{\mathbf{P}}^{[2]\top}\right\|_F\right\} = \sqrt{\operatorname{Tr}(\mathbf{K}^{[1]}) + \operatorname{Tr}(\hat{\mathbf{K}}^{[2]}) - 2\left\|\mathbf{P}^{[1]}\hat{\mathbf{P}}^{[2]\top}\right\|_*} ,$$

and this follows that, from the unitary invariance of the nuclear norm,

$$\left\|\mathbf{P}^{[1]}\hat{\mathbf{P}}^{[2]\top}\right\|_* = \left\|\begin{bmatrix}\mathbf{P}^{[1]} \\ \mathbf{0}_{(n-d)\times n}\end{bmatrix}\begin{bmatrix}\hat{\mathbf{P}}^{[2]\top} & \mathbf{0}_{n\times(n-d)}\end{bmatrix}\right\|_* = \left\|\sqrt{\mathbf{K}^{[1]}}\sqrt{\hat{\mathbf{K}}^{[2]}}\right\|_* .$$

We finally have, from Lemma 3.2,

$$\min_{\mathbf{W}\in\mathcal{U}_d}\left\{\left\|\mathbf{P}^{[1]\top}\mathbf{W} - \hat{\mathbf{P}}^{[2]\top}\right\|_F^2\right\} = \left(\hat{\mathcal{E}}(S_n, \mathcal{K}^{[1]}, \mathcal{K}^{[2]})\right)^2 \leq \frac{1}{n}\sum_{i=1}^n\left\|\boldsymbol{\varphi}^{[1]}(\boldsymbol{x}_i^{[1]}) - \hat{\boldsymbol{\varphi}}^{[2]}(\boldsymbol{x}_i^{[2]})\right\|_2^2 ,$$

which completes the proof. $\square$

# B. Proof of Theorem 3.7

We begin with two useful lemmas as follows:

**Lemma B.1.** *The sub-gradient of $f(\boldsymbol{p})$ in Eqn. (6) is given by*

$$\boldsymbol{v} - diag(\mathbf{M}^\top\mathbf{U}\mathbf{V}^\top)/\sqrt{\boldsymbol{p}} \in \partial f(\boldsymbol{p}) ,$$

*where $\mathbf{U}$ and $\mathbf{V}$ are left and right singular vectors matrices of $\mathbf{M}diag(\sqrt{\boldsymbol{p}})$.*

*Proof.* We set $\mathbf{X} = \mathbf{M}\sqrt{\boldsymbol{p}}$ with the singular value decomposition $\mathbf{X} = \mathbf{U}\boldsymbol{\Sigma}\mathbf{V}^\top$. Following (Watson, 1992), the sub-gradient of $\|\mathbf{X}\|_*$ is given by

$$\partial\|\mathbf{X}\|_* = \left\{\mathbf{U}\mathbf{V}^\top + \mathbf{W}|\mathbf{W} \in \mathbb{R}^{d_1\times d_2}, \mathbf{U}^\top\mathbf{W} = \mathbf{0}, \mathbf{W}\mathbf{V} = \mathbf{0}, \|\mathbf{W}\|_2 \leq 1\right\} .$$

Obviously, $\mathbf{U}\mathbf{V}^\top$ is a sub-gradient by taking $\mathbf{W} = \mathbf{0}$, and we have

$$\frac{\partial\|\mathbf{X}\|_*}{\partial\boldsymbol{p}} = \frac{\partial\|\mathbf{X}\|_*}{\partial\mathbf{X}}\frac{\partial\mathbf{X}}{\partial diag(\sqrt{\boldsymbol{p}})}\frac{\partial diag(\sqrt{\boldsymbol{p}})}{\partial\boldsymbol{p}} = diag(\mathbf{M}^\top\mathbf{U}\mathbf{V}^\top)/\sqrt{\boldsymbol{p}} ,$$

which completes the proof. $\square$

This proposition shows that a singular value decomposition is required to compute the sub-gradient of $f(\boldsymbol{p})$ in each iteration, which is computationally expensive. However, $\mathbf{M}\mathrm{diag}(\sqrt{\boldsymbol{p}})$ has a low rank structure with the rank no more than $\min\{T_e, d_1, d_2\}$ because evolving stage $T_e$ is usually small compared to $d_1$ and $d_2$ in our setting. This could drastically reduce the computational cost on singular value decomposition.

***Proof of Theorem 3.7.*** From Eqn. (6), we have

$$f(\boldsymbol{p}) = \min_{\mathbf{W} \in \mathcal{U}_{T_e}} \{g(\boldsymbol{p}, \mathbf{W})\} \quad \text{with} \quad g(\boldsymbol{p}, \mathbf{W}) = \frac{1}{2}\left\|\sqrt{\tilde{\mathbf{K}}^{[1]}} - \sqrt{\boldsymbol{\Phi}^{[2]}\mathrm{diag}(\boldsymbol{p})\boldsymbol{\Phi}^{[2]\top}}\right\|_F^2 - \frac{1}{2}\mathrm{Tr}\left(\tilde{\mathbf{K}}^{[1]}\right),$$

where $\tilde{\mathbf{K}}^{[1]} = \left[\sum_{k=1}^{d_1} \Phi_{i,k}^{[1]}\Phi_{j,k}^{[1]}/d_1\right]_{T_e \times T_e}$ and $\boldsymbol{\Phi}^{[2]} = [\Phi_{i,j}^{[2]}]_{T_e \times d_2}$. Hence, $f(\boldsymbol{p})$ is a convex function w.r.t. $\boldsymbol{p}$ from the convexity of $\min_{\mathbf{W} \in \mathcal{U}_{T_e}} g(\boldsymbol{p}, \mathbf{W})$ in (Boyd & Vandenberghe, 2004).

In Algorithm 1, we select

$$h(\boldsymbol{p}) = \sum_{i=1}^{d_2} p_i(\ln p_i - 1),$$

and we have the corresponding Bregman divergence

$$D_h(\boldsymbol{p}\|\boldsymbol{q}) = h(\boldsymbol{p}) - h(\boldsymbol{q}) - \langle \nabla h(\boldsymbol{q}), \boldsymbol{p} - \boldsymbol{q}\rangle \quad \text{for} \quad \boldsymbol{p}, \boldsymbol{q} \in \Delta.$$

From the Fenchel conjugate, we also have

$$h^*(\boldsymbol{\theta}) = \sup_{\boldsymbol{p} \in \mathbb{R}^{d_2}} \{\boldsymbol{\theta}^\top \boldsymbol{p} - h(\boldsymbol{p})\} \quad \text{and} \quad \nabla h^*(\boldsymbol{\theta}) = \arg\max_{\boldsymbol{p} \in \mathbb{R}^{d_2}} \{\boldsymbol{\theta}^\top \boldsymbol{p} - h(\boldsymbol{p})\},$$

and define $D_{h^*}(\boldsymbol{p}\|\boldsymbol{q}) = h^*(\boldsymbol{p}) - h^*(\boldsymbol{q}) - \langle \nabla h^*(\boldsymbol{q}), \boldsymbol{p} - \boldsymbol{q}\rangle$ similarly. In Algorithm 1, we rewrite mirror descent iteration as

$$\boldsymbol{p}^{(t)} = \nabla h^*\left(\nabla h(\boldsymbol{p}^{(t-1)}) - \tau_{k-1}\boldsymbol{g}^{(t-1)}\right) \quad \text{with} \quad \boldsymbol{g}^{(t-1)} \in \partial f(\boldsymbol{p}^{(t-1)}).$$

Let $\boldsymbol{p}^* \in \arg\min_{\boldsymbol{p} \in \Delta} f(\boldsymbol{p})$ and $\boldsymbol{\theta}^* = \nabla h(\boldsymbol{p}^*)$. For Bregman divergence (Banerjee et al., 2005), we have

$$D_{h^*}(\boldsymbol{\theta}^{(t)}\|\boldsymbol{\theta}^*) = D_{h^*}(\boldsymbol{\theta}^{(t-1)}\|\boldsymbol{\theta}^*) + \left(\boldsymbol{\theta}^{(t)} - \boldsymbol{\theta}^{(t-1)}\right)^\top \left(\nabla h^*(\boldsymbol{\theta}^{(t-1)} - \nabla h^*(\boldsymbol{\theta}^*))\right) + D_{h^*}(\boldsymbol{\theta}^{(t)}\|\boldsymbol{\theta}^{(t-1)}),$$

and

$$\left(\boldsymbol{\theta}^{(t)} - \boldsymbol{\theta}^{(t-1)}\right)^\top \left(\nabla h^*(\boldsymbol{\theta}^{(t-1)} - \nabla h^*(\boldsymbol{\theta}^*))\right) = -\tau_{t-1}\boldsymbol{g}^{(t-1)\top}\left(\boldsymbol{p}^{(t-1)} - \boldsymbol{p}^*\right).$$

For convex $f$ and $\boldsymbol{g}^{(t-1)} \in \partial f(\boldsymbol{p}^{(t-1)})$, we have $f(\boldsymbol{x}^{(t)}) - f(\boldsymbol{x}^*) \leq \boldsymbol{g}^{(t)\top}(\boldsymbol{p}^{(t)} - \boldsymbol{p}^*)$, and

$$\tau_{t-1}\left[f(\boldsymbol{p}^{(t-1)} - f(\boldsymbol{p}^*))\right] \leq D_{h^*}(\boldsymbol{\theta}^{(t-1)}\|\boldsymbol{\theta}^*) - D_{h^*}(\boldsymbol{\theta}^{(t)}\|\boldsymbol{\theta}^*) + D_{h^*}(\boldsymbol{\theta}^{(t-1)}\|\boldsymbol{\theta}^{(t)}).$$

Summing from $t = 0$ to $T_m$, we have

$$\sum_{t=1}^{T_m} \tau_t\left[f(\boldsymbol{p}^{(t)}) - f(\boldsymbol{p}^*)\right] \leq D_{h^*}(\boldsymbol{\theta}^{(1)}\|\boldsymbol{\theta}^*) + \frac{1}{2}\sum_{t=1}^{T_m} \tau_t^2\left\|\boldsymbol{g}^{(t)}\right\|_\star^2 \leq \ln d_2 + \frac{1}{2}\sum_{t=1}^{T_m}\tau_t^2\left\|\boldsymbol{g}^{(t)}\right\|_\star^2, \tag{22}$$

where the norm $\|\cdot\|_\star$ is defined on $h^*$ and $D_{h^*}(\boldsymbol{\theta}^{(1)}\|\boldsymbol{\theta}^*) = D_h(\boldsymbol{p}^{(1)}\|\boldsymbol{p}^*)$.

For each $t \in [T_m]$, we project $\boldsymbol{p}_t$ onto $\mathcal{P} = \{\boldsymbol{p} \in \Delta : \|\boldsymbol{p}\|_\infty \geq \epsilon\}$ for a small $\epsilon > 0$. We consider the singular value decomposition $\mathbf{M}\mathrm{diag}(\sqrt{\boldsymbol{p}^{(t)}}) = \mathbf{U}_t\boldsymbol{\Sigma}_t\mathbf{V}_t^\top$, and denote by $\boldsymbol{u}_t = \mathrm{diag}(\mathbf{M}^\top\mathbf{U}_t\mathbf{V}_t^\top)$. This follows that

$$\|\boldsymbol{u}_t\|_\infty = \max_{i \in [d_2]}\left\{(\mathbf{M}\mathbf{U}_t\mathbf{V}_t^\top)_{i,i}\right\} \leq \max_{i \in [d_2]}\{\|\mathbf{M}_i\|_2\} \leq T_e\sqrt{d_2},$$

from Lemma B.1 and orthogonality of $\mathbf{U}_t\mathbf{V}_t^\top$. Hence, we have, from the conjugation between $\ell_1$ and $\ell_\infty$ norm,

$$\left\|\boldsymbol{g}^{(t)}\right\|_* \leq \|\boldsymbol{v}\|_\infty + \frac{\|\boldsymbol{u}_t\|_\infty}{\epsilon} \leq T_e\left(1 + \frac{\sqrt{d_2}}{\epsilon}\right), \tag{23}$$

and this follows that, from Eqns. (22)-(23) and by selecting stepsize $\tau_t = \tau$,

$$\frac{1}{T_m}\sum_{t=1}^{T_m} f(\boldsymbol{p}^{(t)}) - f(\boldsymbol{p}^*) \leq \frac{\ln d_2}{\tau T_m} + \frac{\tau T_e}{2}\left(1 + \frac{\sqrt{d_2}}{\epsilon}\right) .$$

This completes the proof by setting $\tau = \sqrt{2\ln d_2 / \left(T_m T_e(1 + \sqrt{d_2}/\epsilon)\right)}$. $\qquad\square$

## C. Analysis for OPFES

### C.1. Proof of Proposition 4.1

Let $d_1$ and $d_2$ be the dimensionalities of random features of $\mathcal{K}^1$ and $\mathcal{K}^2$, respectively. Denote by

$$\mathbf{Z}^{[1]} = \left(\boldsymbol{z}_{T_1+t}^{[k]}\right)_{t=1}^{T_e} \in \mathbb{R}^{d_k \times T_e} \quad \text{and} \quad \mathbf{Z}^{[1]} = \left(\boldsymbol{z}_{T_1+t}^{[k]}\right)_{t=1}^{T_e} \in \mathbb{R}^{d_k \times T_e} .$$

For $d_1 \geq d_2$, we can rewrite the optimization for Eqn. (11) as

$$\mathbf{U}^* \in \operatorname*{arg\,min}_{\mathbf{U} \in \mathcal{U}_{d_2 \times d_1}} \left\{\left\|\mathbf{U}\mathbf{Z}^{[1]} - \mathbf{Z}^{[2]}\right\|_F\right\} ,$$

where $\mathcal{U}_{d_2 \times d_1} = \{\mathbf{U} \in \mathbb{R}^{d_2 \times d_1} : \mathbf{U}\mathbf{U}^\top = \mathbf{I}_{d_2}\}$ is the set of semi-orthogonal matrices. From the proof of Lemma 3.2, we have an equivalent optimization as

$$\max_{\mathbf{U} \in \mathcal{U}_{d_2 \times d_1}} \left\{\operatorname{Tr}\left(\mathbf{U}\mathbf{M}^{(T_1+T_e)}\operatorname{diag}(\sqrt{\boldsymbol{p}^{(T_M)}})\right)\right\} .$$

Let $\mathbf{M}^{(T_1+T_e)}\operatorname{diag}(\sqrt{\boldsymbol{p}^{(T_M)}}) = \mathbf{V}\boldsymbol{\Sigma}\mathbf{W}^\top$ be the singular value decomposition with left and right singular vector matrices $\mathbf{V}^\top \in \mathcal{U}_{d_2 \times d_1}$ and $\mathbf{W} \in \mathcal{U}_{d_2}$, respectively. Denote by $\mathbf{S} = \mathbf{W}^\top\mathbf{U}\mathbf{V}$, and we have, from cyclic invariance of matrix trace,

$$\max_{\mathbf{U} \in \mathcal{U}_{d_2 \times d_1}} \left\{\operatorname{Tr}\left(\mathbf{U}\mathbf{V}\boldsymbol{\Sigma}\mathbf{W}^\top\right)\right\} = \max_{\mathbf{S} \in \mathcal{U}_{d_2}} \left\{\operatorname{Tr}\left(\mathbf{S}\boldsymbol{\Sigma}\right)\right\} .$$

We get the maximum when $\mathbf{S} = \mathbf{I}_{d_2}$ and the optimal solution set for $\mathbf{U}$ is given by

$$\mathbf{U}_* \in \left\{\mathbf{W}\mathbf{V}^\top + \mathbf{Q}(\mathbf{I} - \mathbf{V}\mathbf{V}^\top) \colon \mathbf{Q} \in \mathbb{R}^{d_1 \times d_2}\right\} . \tag{24}$$

For $d_1 < d_2$, we have the equivalent optimization of Eqn. (11) as

$$\mathbf{U}^* \in \operatorname*{arg\,min}_{\mathbf{U}^\top \in \mathcal{U}_{d_1 \times d_2}} \left\{\left\|\mathbf{U}\mathbf{Z}^{[1]} - \mathbf{Z}^{[2]}\right\|_F\right\} .$$

We also take the singular value decomposition of $\mathbf{M}^{(T_1+T_e)}\operatorname{diag}(\sqrt{\boldsymbol{p}^{(T_M)}})$ with $\mathbf{V} \in \mathcal{U}_{d_1}$ and $\mathbf{W}^\top \in \mathcal{U}_{d_1 \times d_2}$, respectively. For $\mathbf{S} = \mathbf{W}^\top\mathbf{U}\mathbf{V} \in \mathbb{R}^{d_1 \times d_1}$, we similarly have the optimal solution set for $\mathbf{U}$ as

$$\mathbf{U}_* \in \left\{\mathbf{W}\mathbf{V}^\top + (\mathbf{I} - \mathbf{W}\mathbf{W}^\top)\mathbf{Q} \colon \mathbf{Q} \in \mathbb{R}^{d_2 \times d_1}\right\} . \tag{25}$$

From Eqns. (24) and (25), $\mathbf{U} = \mathbf{W}\mathbf{V}^\top$ is always an optimal solution for Eqn. (11) by setting $\mathbf{Q} = \mathbf{0}$. $\qquad\square$

### C.2. Proof of Lemma 4.2

For $t \in [T_1 + 1, T_1 + T_e]$, we have, by simple calculations and Cauchy-Schwarz inequality,

$$\left|\langle\boldsymbol{w}_{T_1}^{[1]}, \boldsymbol{z}^{[1]}(\boldsymbol{x}_t^{[1]})\rangle - \langle\boldsymbol{w}_{T_1+T_e}^{[2]}, \boldsymbol{z}^{[2]}(\boldsymbol{x}_t^{[2]})\rangle\right| = \left|\left\langle\boldsymbol{w}_{T_1}^{[1]}, \boldsymbol{z}^{[1]}(\boldsymbol{x}_t^{[1]})\right\rangle - \left\langle\mathbf{U}_*^\top\boldsymbol{w}_{T_1}^{[1]}, \boldsymbol{z}^{[2]}(\boldsymbol{x}_t^{[2]})\right\rangle\right|$$

$$\leq \left|\left\langle\boldsymbol{w}_{T_1}^{[1]}, \boldsymbol{z}^{[1]}(\boldsymbol{x}_t^{[1]}) - \mathbf{U}_*\boldsymbol{z}^{[2]}(\boldsymbol{x}_t^{[2]})\right\rangle\right| \leq \left\|\boldsymbol{w}_{T_1}^{[1]}\right\|_2 \left\|\boldsymbol{z}^{[1]}(\boldsymbol{x}_t^{[1]}) - \mathbf{U}_*\boldsymbol{z}^{[2]}(\boldsymbol{x}_t^{[2]})\right\|_2 .$$

This follows that, by summing $t$ from $T_1 + 1$ to $T_1 + T_e$,

$$
\frac{1}{T_e}\left|\langle \boldsymbol{w}_{T_1}^{[1]}, \boldsymbol{z}^{[1]}(\boldsymbol{x}_i^{[1]})\rangle - \langle \boldsymbol{w}_{T_1+T_e}^{[2]}, \boldsymbol{z}^{[2]}(\boldsymbol{x}_i^{[2]})\rangle\right| \leq \frac{\left\|\boldsymbol{w}_{T_1}^{[1]}\right\|_2}{T_e} \sum_{i=T_1+1}^{T_1+T_e} \left\|\mathbf{U}_* \boldsymbol{z}^{[1]}(\boldsymbol{x}_i^{[1]}) - \boldsymbol{z}^{[2]}(\boldsymbol{x}_i^{[2]})\right\|_2
$$

$$
\leq \frac{\sqrt{2}}{\lambda\sqrt{T_e}} \min_{\mathbf{U}\in\mathcal{U}_{d_2\times d_1}} \left\{\left\|\mathbf{U}_* \left(\boldsymbol{z}^{[1]}(\boldsymbol{x}_t^{[1]})\right)_{t=T_1+1}^{T_1+T_e} - \left(\boldsymbol{z}^{[2]}(\boldsymbol{x}_t^{[2]})\right)_{t=T_1+1}^{T_1+T_e}\right\|_F\right\} = \sqrt{2}\hat{\mathcal{E}}(S_{T_e}^{[e]}, \mathcal{K}^{[1]}, \mathcal{K}^{[2]})/\lambda \,,
$$

where the second inequality holds from Lemma C.5 and Cauchy-Schwarz inequality, and the equality holds from the unitary invariance of Frobenius norm. $\qquad\square$

## C.3. Proof of Theorem 4.3

We first introduce some useful lemmas.

**Lemma C.1** (Hoeffding's bounds (Hoeffding, 1963)). *Let $X_1, X_2, \cdots, X_n$ be independent random variables in $[a,b]$, and $\bar{X} = \sum_{i=1}^n X_i/n$. For $t > 0$, we have*

$$
\Pr\left[\bar{X} - \mathbb{E}[\bar{X}] \geq t\right] \leq \exp\left(-\frac{2nt^2}{(b-a)^2}\right) \,.
$$

**Lemma C.2** (Generalization bound of kernel methods (Mohri et al., 2018)). *Given $S = \{(\boldsymbol{x}_1, y_1), \cdots, (\boldsymbol{x}_n, y_n)\}$ drawn i.i.d from the distribution $\mathcal{D}$. Let $\mathcal{K} : \mathcal{X} \times \mathcal{X} \to \mathbb{R}$ be a kernel bounded by $r^2$, and $\boldsymbol{\varphi}(\cdot)$ is the feature mapping of $\mathcal{K}$ and let $\mathcal{H} = \{\boldsymbol{x} \to \langle \boldsymbol{w}, \boldsymbol{\varphi}(\boldsymbol{x})\rangle : \|\boldsymbol{w}\|_{\mathbb{H}_\mathcal{K}} \leq \Lambda\}$ for some $\Lambda \geq 0$. For loss function $|\ell(\boldsymbol{w}, (\boldsymbol{x}, y))| \leq M$ and $\delta \in (0,1)$, the following holds with probability at least $1 - \delta$ for $h \in \mathcal{H}$,*

$$
R(h) \leq \hat{R}_S(h) + 2r\Lambda\sqrt{\frac{1}{n}} + M\sqrt{\frac{1}{2n}\ln\frac{1}{\delta}} \,,
$$

*where $R(h) = \mathbb{E}_{(\boldsymbol{x},y)\sim\mathcal{D}}[\ell(\boldsymbol{w}, (\boldsymbol{x}, y))]$ and $\hat{R}_S(h) = \sum_{i=1}^n \ell(\boldsymbol{w}, (\boldsymbol{x}_i, y_i))/n$.*

**Lemma C.3** (Online to batch conversion (Cesa-Bianchi et al., 2004)). *Let $S_T = \{(\boldsymbol{x}_1, y_1), \cdots, (\boldsymbol{x}_T, y_T)\}$ be a sample drawn i.i.d. from $\mathcal{D}$, $\ell$ a loss bounded by $M$ and $h_1, \cdots, h_T$ the sequence of hypotheses generated by an online algorithm. For $\delta \in (0,1)$, the following holds with a probability at least $1 - \delta$,*

$$
\mathbb{E}_{(\boldsymbol{x},y)\sim\mathcal{D}}\left[\ell\left(\frac{1}{T}\sum_{t=1}^T h_t(x_t), y_t\right)\right] \leq \frac{1}{T}\sum_{i=1}^T \ell(h_t(x_t), y_t) + M\sqrt{\frac{2\ln(1/\delta)}{T}} \,.
$$

**Lemma C.4.** *Let $S_T = \{(\boldsymbol{x}_1, y_1), \cdots, (\boldsymbol{x}_T, y_T)\}$ be a sample and $B$ be a closed convex set with projection $\Pi_B(\boldsymbol{w}) = \arg\min_{\boldsymbol{w}'\in B}\|\boldsymbol{w} - \boldsymbol{w}'\|$. For a strongly convex loss function $\ell$ with bounded gradient w.r.t. $\boldsymbol{w}$, i.e., $\|\nabla_{\boldsymbol{w}}\ell(\boldsymbol{w}, (x_t, y_t))\| \leq G$ for $\boldsymbol{w} \in B$ and $t \in [T]$. For the update rule $\boldsymbol{w}_t = \Pi_B(\boldsymbol{w}_{t-1} - \nabla\ell(\boldsymbol{w}_t, (x_t, y_t))/\lambda t)$ with $\boldsymbol{w}_0, \boldsymbol{w} \in B$, we have*

$$
\frac{1}{T}\sum_{t=1}^T \ell(\boldsymbol{w}_t, (x_t, y_t)) - \frac{1}{T}\sum_{t=1}^T \ell(\boldsymbol{w}, (x_t, y_t)) \leq \frac{G^2(1 + \ln T)}{2\lambda T} \,.
$$

*Proof.* Denote by $\nabla_t = \nabla_{\boldsymbol{w}}\ell(\boldsymbol{w}_t, (x_t, y_t))$ for simplicity. For $\lambda$-strongly convex functions, we have

$$
\begin{aligned}
\|\boldsymbol{w}_{t+1} - \boldsymbol{w}\|^2 &\leq \|\boldsymbol{w}_t - \boldsymbol{w}\|^2 - 2\tau_t\langle\nabla_t, \boldsymbol{w}_t - \boldsymbol{w}\rangle + \tau_t^2\|\nabla_t\|^2 \\
&\leq \|\boldsymbol{w}_t - \boldsymbol{w}\|^2 - 2\tau_t\left(\ell(\boldsymbol{w}_t, (x_t, y_t)) - \ell(\boldsymbol{w}, (x_t, y_t)) + \frac{\lambda}{2}\|\boldsymbol{w}_t - \boldsymbol{w}\|^2\right) + \tau_t^2\|\nabla_t\|^2 \\
&\leq (1 - \lambda\tau_t)\|\boldsymbol{w}_t - \boldsymbol{w}\|^2 - 2\tau_t\left(\ell(\boldsymbol{w}_t, (x_t, y_t)) - \ell(\boldsymbol{w}, (x_t, y_t))\right) + \tau_t^2\|\nabla_t\|^2 \,.
\end{aligned}
$$

This follows that

$$
\ell(\boldsymbol{w}_t, (x_t, y_t)) - \ell(\boldsymbol{w}, (x_t, y_t)) \leq \frac{\tau_t^{-1} - \lambda}{2}\|\boldsymbol{w}_t - \boldsymbol{w}\|^2 - \frac{1}{2\tau_t}\|\boldsymbol{w}_{t+1} - \boldsymbol{w}\|^2 + \frac{\tau_t G^2}{2} \,.
$$

We have, by summing $t = 1$ to $T$, and setting $\tau_t = 1/(\lambda t)$ with $1/\tau_0 = 0$,

$$2 \sum_{t=1}^{T} \left( \ell(\boldsymbol{w}_t, (x_t, y_t)) - \ell(\boldsymbol{w}, (x_t, y_t)) \right) \leq \sum_{t=1}^{T} \| \boldsymbol{w}_t - \boldsymbol{w} \|^2 \left( \frac{1}{\tau_t} - \frac{1}{\tau_{t-1}} - \lambda \right) + G^2 \sum_{t=1}^{T} \tau_t = G^2 \sum_{t=1}^{T} \frac{1}{\lambda t} \leq \frac{G^2}{\lambda} (1 + \ln T) ,$$

which completes the proof. $\qquad \square$

**Lemma C.5.** *Let $S_T = \{(\boldsymbol{x}_1, y_1), \cdots, (\boldsymbol{x}_T, y_T)\}$ be a sample. For kernel function $\mathcal{K}(\boldsymbol{x}, \boldsymbol{x}) \leq r^2$. we have $T$ classifiers $h_1, \cdots, h_T$ generated by online kernel learning with $\ell_t(h) = \max\{1 - y_i h(\boldsymbol{x}_i), 0\} + \lambda \|h\|_{\mathbb{H}}^2 / 2$. We have*

$$\frac{1}{T} \sum_{t=1}^{T} \ell_t(h_t) - \frac{1}{T} \sum_{t=1}^{T} \ell_t(h_*) \leq \frac{4r^2(1 + \ln T)}{\lambda T} ,$$

*where $h_* = \arg\min_{h \in \mathbb{H}} \{ \sum_{t=1}^{T} \ell_t(h)/T \}$.*

*Proof.* By setting $\tau_t = 1/(\lambda t)$ and $h_0 = 0$, we rewrite the update rule as

$$h_t = \left( 1 - \frac{1}{t} \right) h_{t-1} - \frac{1}{\lambda t} g_t \quad \text{with} \quad g_t = \mathbb{I}[y_t h(\boldsymbol{x}_t) < 1] y_t \boldsymbol{\varphi}(\boldsymbol{x}_t) , \tag{26}$$

where $\boldsymbol{\varphi}(\boldsymbol{x}_t)$ is the feature mapping of $\mathcal{K}$. Hence, we have

$$h_t = \frac{1}{\lambda t} \sum_{i=1}^{t} g_i \quad \text{and} \quad \|h_t\|_{\mathbb{H}} \leq \frac{r}{\lambda} ,$$

from $\prod_{j=i+1}^{t} (1 - 1/j) = i/t$ for $i \leq t - 1$. We complete the proof from Eqn. (26) and Lemma C.4. $\qquad \square$

From Lemma C.5, we have the following corollary.

**Corollary C.6.** *For online kernel learning in the previous stage (Figure 1), we have*

$$\left\| \boldsymbol{w}_{T_1}^{[1]} \right\|_2 \leq \frac{r}{\lambda} \quad \text{and} \quad \left\| \frac{1}{T_1} \sum_{t=1}^{T_1} \boldsymbol{w}_t^{[1]} \right\|_2 \leq \frac{r}{\lambda} .$$

**Lemma C.7.** *Given $S_n = \{(\boldsymbol{x}_1, y_1), \cdots, (\boldsymbol{x}_T, y_T)\}$, and for a kernel $\mathcal{K}$ bounded by $r^2$ and a classifier $h_0 \in \mathbb{H}$, let $h_1, \cdots, h_T$ be classifiers generated by online kernel learning with $\ell(h, (\boldsymbol{x}, y)) = \max\{1 - yh(\boldsymbol{x}), 0\} + \lambda \|h\|_{\mathbb{H}}^2 / 2$ and $\lambda > 0$. For $h_* = \arg\min_{h \in \mathbb{H}} \sum_{t=1}^{T} \ell_t(h)$, we have*

$$\frac{1}{T} \sum_{t=1}^{T} \ell_t(h_t) - \frac{1}{T} \sum_{t=1}^{T} \ell_t(h_*) \leq \frac{2r \|h_0 - h_*\|_{\mathbb{H}}}{\sqrt{T}} .$$

*Proof.* For the norm in RKHS, we have

$$\|h_{t+1} - h_*\|_{\mathbb{H}}^2 = \|h_t - \eta \nabla \ell_t(h_t) - h_*\|_{\mathbb{H}}^2 = \|h_t - h_*\|_{\mathbb{H}}^2 + \eta^2 \|\nabla \ell_t(h_t)\|_{\mathbb{H}}^2 - 2\eta \nabla \ell_t(h_t)^{\top}(h_t - h_*) ,$$

and this follows that, from convex loss function $\ell_t(h_t) - \ell_t(h_*) \leq \nabla \ell_t(h_t)^{\top}(h_t - h_*)$,

$$\ell_t(h_t) - \ell_t(h_*) \leq \frac{\|h_t - h_*\|_{\mathbb{H}}^2 - \|h_{t+1} - h_*\|_{\mathbb{H}}^2}{2\eta} + \frac{\eta}{2} \|\nabla \ell_t(h_t)\|^2 .$$

We have, by summing from $t = 0$ to $T - 1$,

$$\sum_{t=1}^{T} (\ell_t(h_t) - \ell_t(h_*)) \leq \frac{\|h_0 - h_*\|_{\mathbb{H}}^2}{2\eta} + 2\eta r^2 T ,$$

which completes the proof by setting $\eta_t = \|h_0 - h_*\|_{\mathbb{H}} / (r\sqrt{T})$. $\qquad \square$

**Lemma C.8.** *Given samples $S_1 = \{(\boldsymbol{x}_i, y_i)\}_{i=1}^{n_1}$ and $S_2 = \{(\boldsymbol{x}_i, y_i)\}_{i=n_1+1}^{n_1+n_2}$ with $\|\boldsymbol{x}\|_2 \le r$, denote by*

$$\boldsymbol{w}_1^* \in \underset{\boldsymbol{w} \in \mathbb{R}^d}{\arg\min} \left\{ \hat{R}_1(\boldsymbol{w}) = \frac{1}{n_1} \sum_{i=1}^{n_1} \ell(\boldsymbol{w}, (\boldsymbol{x}_i, y_i)) + \frac{\lambda}{2} \|\boldsymbol{w}\|_2^2 \right\},$$

$$\boldsymbol{w}_2^* \in \underset{\boldsymbol{w} \in \mathbb{R}^d}{\arg\min} \left\{ \hat{R}_2(\boldsymbol{w}) = \frac{1}{n_2} \sum_{i=n_1+1}^{n_1+n_2} \ell(\boldsymbol{w}, (\boldsymbol{x}_i, y_i)) + \frac{\lambda}{2} \|\boldsymbol{w}\|_2^2 \right\}.$$

*For $\delta \in (0, 1)$, the following holds with probability at least $1 - \delta$,*

$$\hat{R}_1(\boldsymbol{w}_1^*) - \hat{R}_2(\boldsymbol{w}_2^*) \le \frac{r^2}{\lambda} \sqrt{\left( \frac{1}{n_1} + \frac{1}{n_2} \right) \ln \left( \frac{1}{\delta} \right)}.$$

*Proof.* We introduce a new function

$$f(S_1, S_2) = \hat{R}_1(\boldsymbol{w}_1^*) - \hat{R}_2(\boldsymbol{w}_2^*),$$

and consider the sample $S_1' = S_1 \backslash \{(\boldsymbol{x}_k, y_k)\} \cup \{\boldsymbol{x}_k', y_k'\}$ for $k \in [n_1]$. From Cauchy-Schwarz inequality and 1-Lipschitz hinge loss, we have

$$|f(S_1', S_2) - f(S_1, S_2)| = \left| \hat{R}_1(\boldsymbol{w}_1^*) - \hat{R}_1'(\boldsymbol{w}_1'^*) \right| \le r \|\boldsymbol{w}_1^* - \boldsymbol{w}_1'^*\|_2 + \frac{\lambda}{2} \|\boldsymbol{w}_1^* - \boldsymbol{w}_1'^*\|_2 \cdot \|\boldsymbol{w}_1^* + \boldsymbol{w}_1^{*'}\|_2 \le 2r \|\boldsymbol{w}_1^* - \boldsymbol{w}_1'^*\|_2,$$

where the norm of optimal classifiers satisfy

$$\|\boldsymbol{w}_1^*\|_2 \le \frac{r}{\lambda}, \quad \|\boldsymbol{w}_1'^*\|_2 \le \frac{r}{\lambda} \quad \text{and} \quad \|\boldsymbol{w}_1^* + \boldsymbol{w}_1'^*\|_2 \le \frac{2r}{\lambda},$$

from the KKT condition as in the proof of Theorem 3.3. From strong convexity, we have

$$\hat{R}_1(\boldsymbol{w}_1'^*) \ge \hat{R}_1(\boldsymbol{w}_1^*) + \frac{\lambda}{2} \|\boldsymbol{w}_1^* - \boldsymbol{w}_1'^*\|_2^2 \quad \text{and} \quad \hat{R}_1'(\boldsymbol{w}_1^*) \ge \hat{R}_1'(\boldsymbol{w}_1'^*) + \frac{\lambda}{2} \|\boldsymbol{w}_1^* - \boldsymbol{w}_1'^*\|_2^2, \tag{27}$$

and this follows that,

$$\|\boldsymbol{w}_1^* - \boldsymbol{w}_1'^*\|_2^2 \le \frac{1}{\lambda} \left( \hat{R}_1(\boldsymbol{w}_1'^*) - \hat{R}_1(\boldsymbol{w}_1^*) - \hat{R}_1'(\boldsymbol{w}_1^*) + \hat{R}_1'(\boldsymbol{w}_1'^*) \right)$$

$$= \frac{1}{\lambda n_1} \left( \ell(\boldsymbol{w}_1'^*, (\boldsymbol{x}_k, y_k)) - \ell(\boldsymbol{w}_1^*, (\boldsymbol{x}_k, y_k)) + \ell(\boldsymbol{w}_1^*, (\boldsymbol{x}_k', y_k')) - \ell(\boldsymbol{w}_1'^*, (\boldsymbol{x}_k', y_k')) \right) \le \frac{r}{\lambda n_1} \|\boldsymbol{w}_1^* - \boldsymbol{w}_1'^*\|_2.$$

Hence, we have, from Eqn. (27),

$$|f(S_1, S_2) - f(S_1', S_2)| \le \frac{2r^2}{\lambda n_1}.$$

We could make a similar analysis for $S_2$. This completes the proof from Lemma A.4. $\qquad \square$

***Proof of Theorem 4.3.*** For $k = 1$ and $k = 2$, we introduce some notations as follows:

$$\hat{R}_{T_1}(\boldsymbol{w}^{[k]}) = \frac{1}{T_1} \sum_{t=1}^{T_1} \ell(\boldsymbol{w}^{[k]}, (\boldsymbol{x}_t^{[k]}, y_t)) + \frac{\lambda}{2} \left\| \boldsymbol{w}^{[k]} \right\|_2^2,$$

$$\hat{R}_{T_e}(\boldsymbol{w}^{[k]}) = \frac{1}{T_e} \sum_{t=T_1+1}^{T_1+T_e} \ell(\boldsymbol{w}^{[k]}, (\boldsymbol{x}_t^{[k]}, y_t)) + \frac{\lambda}{2} \left\| \boldsymbol{w}^{[k]} \right\|_2^2,$$

$$\hat{R}_{T_2}(\boldsymbol{w}^{[k]}) = \frac{1}{T_2} \sum_{t=T_1+T_e+1}^{T_1+T_e+T_2} \ell(\boldsymbol{w}^{[k]}, (\boldsymbol{x}_t^{[k]}, y_t)) + \frac{\lambda}{2} \left\| \boldsymbol{w}^{[k]} \right\|_2^2.$$

Denote by $\ell_t(\boldsymbol{w}^{[k]}) = \ell(\boldsymbol{w}^{[k]}, (\boldsymbol{x}_t^{[k]}, y_t)) + \lambda \|\boldsymbol{w}^{[k]}\|_2^2$. From the i.i.d assumption, it is natural to consider samples on new feature space $\mathcal{X}^{[2]}$ in the previous stage and on old feature space $\mathcal{X}^{[1]}$ in the current stages respectively.

We have, from Lemma C.5 and strong convexity,

$$\hat{\mathcal{L}}_{T_2}^{[2]} - \mathcal{L}_{T_2}^{[2]}(\boldsymbol{w}_*^{[2]}) \le 2r\sqrt{\frac{2}{\lambda T_2}\left(\hat{R}_{T_2}^{[2]}(\boldsymbol{w}_{T_1+T_e}^{[2]}) - \hat{R}_{T_2}^{[2]}(\boldsymbol{w}_*^{[2]})\right)}, \tag{28}$$

and we also have $\hat{R}_{T_2}(\boldsymbol{w}_{T_1+T_e}^{[2]}) - \hat{R}_{T_2}(\boldsymbol{w}_{T_2*}^{[2]}) = Z_1 + Z_2 + Z_3 + Z_4 + Z_5 + Z_6 + Z_7$, where

$$Z_1 = \hat{R}_{T_2}(\boldsymbol{w}_{T_1+T_e}^{[2]}) - \hat{R}_{T_e}(\boldsymbol{w}_{T_1+T_e}^{[2]}), \quad Z_2 = \hat{R}_{T_e}(\boldsymbol{w}_{T_1+T_e}^{[2]}) - \hat{R}_{T_e}(\boldsymbol{w}_{T_1}^{[1]}), \quad Z_3 = \hat{R}_{T_e}(\boldsymbol{w}_{T_1}^{[1]}) - R_{T_1}(\boldsymbol{w}_{T_1}^{[1]}),$$

$$Z_4 = R_{T_1}(\boldsymbol{w}_{T_1}^{[1]}) - \frac{1}{T_1}\sum_{t=1}^{T_1}\ell_t(\boldsymbol{w}_t^{[1]}), \quad Z_5 = \frac{1}{T_1}\sum_{t=1}^{T_1}\ell_t(\boldsymbol{w}_t^{[1]}) - \hat{R}_{T_1}(\boldsymbol{w}_{T_1*}^{[1]}), \quad Z_6 = \hat{R}_{T_1}(\boldsymbol{w}_{T_1*}^{[1]}) - \hat{R}_{T_2}(\boldsymbol{w}_{T_2*}^{[1]}),$$

$$Z_7 = \hat{R}_{T_2}(\boldsymbol{w}_{T_2*}^{[1]}) - \hat{R}_{T_2}(\boldsymbol{w}_{T_2*}^{[2]}).$$

From the i.i.d assumption for evolving and current stage, the following holds with probability at least $1 - \delta/6$,

$$\begin{aligned}
Z_1 &\le \mathbb{E}[Z_1] + \max_{(\boldsymbol{x}^{[2]},y)\in\mathcal{X}^{[2]}\times\mathcal{Y}}\left\{\ell\left(\boldsymbol{w}_{T_1+T_e}^{[2]},(\boldsymbol{x}^{[2]},y)\right)\right\}\sqrt{\left(\frac{1}{2T_e}+\frac{1}{2T_2}\right)\ln\frac{6}{\delta}} \\
&= \left(\frac{r^2}{\lambda}+1\right)\sqrt{\left(\frac{1}{2T_e}+\frac{1}{2T_2}\right)\ln\frac{6}{\delta}},
\end{aligned} \tag{29}$$

from Lemma C.1 and Corollary C.6. Similarly, the following holds with the probability as least $1 - \delta/6$,

$$Z_3 \le \mathbb{E}[Z_3] + \max_{(\boldsymbol{x}^{[2]},y)\in\mathcal{X}^{[2]}\times\mathcal{Y}}\left\{\ell\left(\boldsymbol{w}_{T_1+T_e}^{[2]},(\boldsymbol{x}^{[2]},y)\right)\right\}\sqrt{\frac{\ln(1/\delta_2)}{2T_e}} = \left(\frac{r^2}{\lambda}+1\right)\sqrt{\frac{\ln(1/\delta_2)}{2T_e}}. \tag{30}$$

From Lemma 4.2 and Theorem 3.4, the following holds with the probability at least $1 - \delta/6$,

$$\begin{aligned}
Z_2 &= \frac{1}{T_e}\sum_{t=T_1+1}^{T_1+T_e}\left(\ell\left(\boldsymbol{w}_{T_1}^{[1]},(\boldsymbol{x}_t^{[1]},y_t)\right) - \ell\left(\boldsymbol{w}_{T_1+T_e}^{[2]},(\boldsymbol{x}_t^{[2]},y_t)\right)\right) \\
&\le \frac{1}{T_e}\sum_{t=T_1+1}^{T_1+T_e}\left|\langle\boldsymbol{w}_{T_1}^{[1]},\boldsymbol{z}^{[1]}(\boldsymbol{x}_i^{[1]})\rangle - \langle\boldsymbol{w}_{T_1+T_e}^{[2]},\boldsymbol{z}^{[2]}(\boldsymbol{x}_i^{[2]})\rangle\right| \\
&\le \frac{r\hat{\mathcal{E}}(S_{T_e}^{[e]},\mathcal{K}^{[1]},\mathcal{K}^{[2]})}{\lambda} \le \frac{r}{\lambda}\left(\mathcal{E}(\mathcal{D},\mathcal{K}^{[1]},\mathcal{K}^{[2]}) + c_1 r\sqrt{\frac{1}{T_e}\ln\frac{6}{\delta}}\right).
\end{aligned} \tag{31}$$

From Lemma C.3 , the following holds with the probability at least $1 - \delta/6$,

$$Z_4 \le \left(\frac{r^2}{\lambda}+1\right)\sqrt{\frac{\ln(6/\delta)}{2T_1}}, \tag{32}$$

and we have, from Lemma C.5,

$$Z_5 \le \frac{4r^2\ln(1+T_1)}{\lambda T_1} \le \frac{4r^2}{\lambda\sqrt{T_1}}. \tag{33}$$

From Lemma C.8, the following holds with a probability at least $1 - \delta/6$,

$$Z_6 \le \frac{r^2}{\lambda}\sqrt{\left(\frac{1}{T_1}+\frac{1}{T_2}\right)\ln\frac{6}{\delta}}. \tag{34}$$

From Theorem 3.3 and Theorem 3.4, the following holds with a probability at least $1 - \delta/6$,

$$
\begin{aligned}
Z_7 &\leq \frac{1}{T_2} \sum_{t=T_1+T_e+1}^{T_1+T_e+T_2} \left| \langle \boldsymbol{w}_{T_2*}^{[2]}, \boldsymbol{x}_t^{[2]} \rangle - \langle \boldsymbol{w}_{T_2*}^{[1]}, \boldsymbol{x}_t^{[1]} \rangle \right| + \frac{\lambda}{2} \left\| \boldsymbol{w}_{T_2*}^{[1]} - \boldsymbol{w}_{T_2*}^{[2]} \right\|_2 \left\| \boldsymbol{w}_{T_2*}^{[1]} + \boldsymbol{w}_{T_2*}^{[2]} \right\|_2 \\
&\leq \frac{r}{\lambda} \hat{\mathcal{E}}(S_{T_2}^{[2]}, \mathcal{K}^{[1]}, \mathcal{K}^{[2]}) + \frac{r}{\lambda} \sqrt{2r \hat{\mathcal{E}}(S_{T_2}^{[2]}, \mathcal{K}^{[1]}, \mathcal{K}^{[2]})} + r \left\| \boldsymbol{w}_{T_2*}^{[1]} - \boldsymbol{w}_{T_2*}^{[2]} \right\|_2 \\
&\leq \frac{r}{\lambda} \hat{\mathcal{E}}(S_{T_2}^{[2]}, \mathcal{K}^{[1]}, \mathcal{K}^{[2]}) + \frac{2r}{\lambda} \sqrt{2r \hat{\mathcal{E}}(S_{T_2}^{[2]}, \mathcal{K}^{[1]}, \mathcal{K}^{[2]})} \\
&\leq \frac{r}{\lambda} \left( \mathcal{E}\left( \mathcal{D}, \mathcal{K}^{[1]}, \mathcal{K}^{[2]} \right) + 2\sqrt{2r \mathcal{E}\left( \mathcal{D}, \mathcal{K}^{[1]}, \mathcal{K}^{[2]} \right)} \right) + O\left( \sqrt[4]{\frac{\ln(6/\delta)}{T_2}} \right).
\end{aligned}
\tag{35}
$$

From Eqns. (29)-(35) and union bounds, the following holds with probability at least $1 - \delta$,

$$
\hat{R}_{T_2}(\boldsymbol{w}_{T_1+T_e}^{[2]}) - \hat{R}_{T_2}(\boldsymbol{w}_{T_2*}^{[2]}) \leq \frac{r^2}{\lambda} \left[ c\left( \frac{1}{\sqrt{T_1}} + \frac{1}{\sqrt{T_e}} + \frac{1}{\sqrt[4]{T_2}} \right) \ln\left( \frac{6}{\delta} \right) + \frac{\mathcal{E}}{r} + \sqrt{\frac{2\mathcal{E}}{r}} \right],
\tag{36}
$$

for some constant $c > 0$ with $(1 - \delta/6)^6 \geq 1 - \delta$. We complete the proof by combining Eqn. (28) and Eqn. (36). $\qquad\square$

### C.4. Proof of Theorem 4.4

For simplicity, we reindex the time-step of samples in the current stage as $t = 1, \cdots, T_2$. Denote by $\ell_{t,i}$ the loss for the $i$-th base learner at the $t$-th iteration with $i \in [2]$, and $\mathcal{L}_{t,i}$ is the corresponding cumulative loss. We define the potential function

$$
\Phi_t = \frac{1}{\gamma} \ln \left( \sum_{i=1}^{2} \exp(-\gamma \mathcal{L}_{t,i}) \right),
$$

and this follows that, from $e^{-x} \leq 1 - x + x^2$ and $\ln(1+x) \leq x$,

$$
\begin{aligned}
\Phi_t - \Phi_{t-1} &= \frac{1}{\gamma} \left( \frac{\exp(-\gamma \mathcal{L}_{t,i})}{\sum_{i=1}^{2} \exp(-\gamma \mathcal{L}_{t-1,i})} \right) = \frac{1}{\gamma} \left( \sum_{i=1}^{2} \omega_{t,i} \exp(-\gamma \ell_{t,i}) \right) \\
&\leq \frac{1}{\gamma} \left( \sum_{i=1}^{2} \omega_{t,i}(1 - \gamma \ell_{t,i} + \gamma^2 \ell_{t,i}^2) \right) = \frac{1}{\gamma} \ln \left( 1 - \gamma \langle \boldsymbol{\omega}_t, \boldsymbol{\ell}_t \rangle + \gamma^2 \sum_{i=1}^{2} \omega_{t,i} \ell_{t,i}^2 \right) \leq \langle \boldsymbol{\omega}_t, \boldsymbol{\ell}_t \rangle + \gamma^2 \sum_{i=1}^{2} \omega_{t,i} \ell_{t,i}^2,
\end{aligned}
$$

where $\boldsymbol{\omega}_t = (\omega_{t,1}, \omega_{t,2})$, $\boldsymbol{\ell}_t = (\ell_{t,1}, \ell_{t,2})$, and the last equality holds from Eqn. (12). We have, by summing over $t \in [T_2]$,

$$
\begin{aligned}
\sum_{t=1}^{T_2} \langle \boldsymbol{\omega}_t, \boldsymbol{\ell}_t \rangle &\leq \Phi_0 - \Phi_{T_2} + \gamma \sum_{t=1}^{T_2} \sum_{i=1}^{2} \omega_{t,i} \ell_{t,i}^2 \\
&\leq \frac{\ln 2}{\gamma} - \frac{1}{\gamma} \ln \left( \exp(-\gamma \mathcal{L}_{T_2,i*}) \right) + \gamma \sum_{t=1}^{T_2} \sum_{i=1}^{2} \omega_{t,i} \ell_{t,i}^2 \leq \frac{\ln 2}{\gamma} + \mathcal{L}_{T_2,i} + \gamma \sum_{t=1}^{T_2} \sum_{i=1}^{2} \omega_{t,i} \ell_{t,i}^2,
\end{aligned}
$$

where $\mathcal{L}_{T_2,i*} = \min_{i \in \{1,2\}} \mathcal{L}_{T_2,i}$. We have, by rearranging and from Theorem 4.3,

$$
\sum_{t=1}^{T_2} \langle \boldsymbol{\omega}_t, \boldsymbol{\ell}_t \rangle - \min_{i=1,2} \mathcal{L}_{T_2,i} \leq \frac{\ln 2}{\gamma} + \gamma \sum_{t=1}^{T_2} \sum_{i=1}^{2} \omega_{t,i} \ell_{t,i}^2 \leq \frac{\ln 2}{\gamma} + \gamma T_2 \left( 1 + \frac{3r^2}{2\lambda} \right),
$$

which completes the proof by setting $\gamma = \sqrt{\ln 2 / ((1 + 3r^2/2\lambda)T_2)}$. $\qquad\square$

