# OpenReview forum: "One-Pass Feature Evolvable Learning with Theoretical Guarantees"
_ICML.cc/2025/Conference — ICML 2025 poster_

### Official Review · Reviewer_2L3u · 2025-03-11

**Overall Recommendation:** 4

**Summary:**

This work focuses on the online learning scenario with a special assumption of the environment: the feature space evolves, where old features vanish and new features emerge during the process of online learning. This work considers to characterize the feature relationship via kernel function, and proposes the KOM discrepancy (Kernel Ortho-Mapping Discrepancy) as a difference measure between two feature spaces, which equals the minimum difference between the empirical mappings of two kernel functions. Based on the proposed KOM discrepancy, the authors develop the OPFES approach, which consists of the online learning process covering all three stages in feature evolvable data stream. The OPFES approach includes the traditional online kernel learning as well as the online optimization of the proposed KOM discrepancy. The authors verify the superiority of the proposed OPFES approach both theoretically and empirically. They provide regret bound for the proposed OPFES approach, and the experimental result shows the promising performance of the OPFES approach.

**Claims And Evidence:**

Yes. The claims made in this submission is well-supported. The authors claim that the proposed OPFES approach is superior to other methods, and present both theoretical and empirical evidence. They give a regret bound to show that the OPFES approach benefits from the minimization of KOM discrepancy and the previous obtained kernel model. Besides, they conduct extensive experiments to show that the performance is overall better than other online learning methods with or without the modification for feature evolvable data streams.

**Essential References Not Discussed:**

The authors do not miss any important related works. I am familiar with online learning, and I think that the authors have already cited all important works in this area.
The paper mentions many famous online learning algorithms, e.g., online SVM, online kernel SVM, etc. The authors also compare the proposed method with previous state-of-the-art algorithms like the variations of FESL and OCDS.
Since this paper also gives some theoretical analysis of the proposed OPFES approach, I also checked the related works in this area. I found that the regret bound in this paper achieves O(1/\sqrt{T}) rate, which is comparable with the state-of-the-art result in online learning scenario with strongly convex loss function. The related theoretical works are also cited in this paper.

**Experimental Designs Or Analyses:**

Yes. The experimental designs are sound. The authors first introduce other compared methods, and gives a detailed discussion on the parameter settings of each method. They then compare the cumulative error rate of these methods, which is a very important criteria for online learning methods, and they give some analysis on the experimental result. They also conduct experiment on different dimensionalities of random Fourier feature to show that the algorithm benefits from moderately large number of features. The experimental result is overall convincing.

**Methods And Evaluation Criteria:**

Yes. The proposed methods and criteria make sense for the problem. In online learning, it is quite common to feed online learning algorithm with data instances from traditional benchmark datasets in random order to simulate the online learning process, which is also adopted in this paper. Besides, the cumulative error rate is also a widely used criteria in online learning scenario.

**Other Comments Or Suggestions:**

The clarity of this work could be further improved if the authors were to summarize all the notation used in this paper in a table. Current notations are a little bit confusing. Despite the excellent work, it did take me a long time to understand each notation.

**Other Strengths And Weaknesses:**

The detailed strengths of this work are as follows:
1. This work is well-written and it is easy to follow. Besides, the authors also uploaded the source code in the supplementary material, which further enhances the reproducibility.
2. This paper shows a relatively good originality. The scenario of online learning with changing feature spaces was introduced a couple of years ago and many algorithms were proposed. However, existing methods only focus on linear transform between feature spaces, yet without theoretical guarantee on the regret of the online learning algorithm. The authors introduce the KOM discrepancy to characterize the relationships between old and new feature spaces, which considers non-linear and more complicated relations compared with most existing algorithms. In addition, they theoretically demonstrate that the difference between the performances of two optimal kernel classifiers can be upper-bounded by KOM discrepancy, and reaches better regret bounds based on this theorem.
2. This paper has solid theoretical and empirical analysis and is significant to some extent. It theoretically shows that the model learned in the old stage can help achieve fast convergence and better regret in the new stage, which is inspiring and interesting to further develop more practical learning algorithms for this scenario. The empirical result also verifies the superiority of the OPFES method, and is convincing for me.

Despite the strengths above, I have some concerns on this work:
The notations used in this paper are a bit confusing, with a large number of subscripts and superscripts that significantly affect the readability of the text. This complexity makes it challenging for readers to follow the main arguments and mathematical formulations. I recommend simplifying the notation wherever possible to enhance clarity.

**Questions For Authors:**

1. It seems that the OPFES approach is highly dependent on the random Fourier feature representation of kernel functions. I would like to know whether it is necessary to adopt random Fourier features, or there are other kernel approximation methods that are compatible with the OPFES approach?
2. What is the relationship between feature evolvable learning and multi-model learning? Can it be viewed as an online version of multi-model learning? For example, in the previous stage only images can be observed, and in the current stage only text description is available.
3. Is it possible to apply other non-linear models, i.e., neural networks, to the feature evolvable data stream learning scenario with the proposed KOM discrepancy?

**Relation To Broader Scientific Literature:**

This work has great impact on the area of online learning with feature evolvable streams, or even on multi-model learning if it were to be accepted. Previous works on this area mostly consider linear feature relationships, while this paper proposes to incorporate kernel mapping and describe the feature relationship as the proposed KOM discrepancy. The proposed discrepancy is not limited to linear relationship, which pushes the performance boundary forward for algorithms handling feature evolvable data streams. Benefit from the KOM discrepancy, the authors propose OPFES method and achieve the state-of-the-art performance among other representative algorithms, which mostly consider linear relationship between different feature spaces.

**Theoretical Claims:**

Yes. I checked the detailed proofs of two main theorems in the appendix, and I believe that the proofs are correct. For Theorem 3.3, the authors incorporate the empirical kernel mapping to derive the finite-dimensional kernel mapping, and consider to bound the difference of linear SVMs under the finite-dimensional feature mapping. The proof of Theorem 3.3 is straightforward and easy to follow.
For another important theorem, Theorem 4.3, the proof basically follows traditional regret analysis of online SVM method, but uses Theorem 3.3 to generalize traditional regret bound to feature evolvable scenario. The proof technique is interesting and correct.
For other theorems in this paper, I roughly checked the proof. The ingredients are similar, and the proofs seem to be correct.

---

> ### Author Rebuttal · Authors · 2025-04-01
>
> [Q1] …whether it is necessary to adopt random Fourier features, or are there other kernel approximation methods that are compatible with the OPFES approach?
>
> [A1] We will clarify that we take random Fourier features to avoid storing the entire and partial data, since kernel function $k(x_1,x_2)$ is defined on a pair of instances, and previous online kernel learning requires to store partial and entire training data [Kivinen et al. 2001; Orabona et al. 2012; Ghari & Shen 2024]. We will clarify that the random Fourier features technique has been a popular approximation with theoretical guarantee [Rahimi & Recht, 2007; Avron et al., 2017; Likhosherstov et al., 2022], and it is interesting to exploit other approximations.
>
>  [Q2] What is the relationship between feature evolvable learning and multi-modal learning? Can it be viewed as an online version of multi-modal learning...
>
> [A2] We will clarify that our work focuses on the general feature spaces without constraints on the types of feature spaces as in [Hou et al., 2017; Zhang et al., 2020], and it can be directly used for two-modal learning. For multi-modal (more than two modals) learning, this problem becomes rather difficult since we should consider multiple and combinatorial correlations for multiple models.
>
> [Q3] Is it possible to apply other non-linear models, i.e., neural networks, to the feature evolvable data stream learning scenario with the proposed KOM discrepancy?
>
> [A3] We will clarify that this work considers different non-linear models by selecting different kernel functions, such as Gaussian kernels. It is interesting to consider deep neural networks, and we could take the NTK technique [Jacot et al., 2018; Novak et al., 2022] to analyze neural networks with the KOM discrepancy.

---

> > ### Comment · Reviewer_2L3u · 2025-04-03
> >
> > The rebuttal has solved my question, and I will keep the score.

---

### Official Review · Reviewer_WCDh · 2025-03-16

**Overall Recommendation:** 3

**Summary:**

This paper tackles online learning in feature-evolving streams where old features vanish and new ones emerge. They propose OPFES, a one-pass algorithm that processes data without storage. It combines online kernel learning with random Fourier features to adaptively capture evolving data patterns and introduces a kernel ortho-mapping discrepancy framework through kernel functions to quantify cross-feature space relationships. Theoretical analysis establishes a regret bound for the algorithm, while empirical evaluations compare cumulative error rates across methods. Experiments confirm OPFES' superior performance and efficiency in managing real-time feature evolution, offering both practical streaming solutions and theoretical guarantees for feature-evolvable learning tasks.

**Claims And Evidence:**

The claims made in the submission are well-supported by both theoretical analysis and empirical results. The paper introduces the Kernel Ortho-Mapping (KOM) discrepancy to characterize relationships between different feature spaces and establishes theoretical guarantees linking KOM discrepancy to classifier performance. The proposed one-pass feature evolvable learning algorithm is backed by regret analysis and convergence proofs, ensuring a solid theoretical foundation. Additionally, the empirical validation includes extensive experiments on multiple datasets, demonstrating the effectiveness and efficiency of the proposed method compared to existing approaches.

**Essential References Not Discussed:**

No. Given that the paper appears to cite a comprehensive range of related works in the field, it is likely that the authors have covered the essential literature in feature evolvable learning. While the paper covers foundational work, discussing recent innovations in dynamically adjusting kernel functions could help readers understand how the proposed method fits into the current trends and innovations in adapting kernels to changing data scenarios.

**Experimental Designs Or Analyses:**

The experimental design in the study is commendable for its thoroughness and rigor. The use of multiple datasets and comparison with leading methods effectively showcases the versatility and robustness of the OPFES approach. The random splitting of feature spaces and the integration of random Fourier features highlight a sophisticated strategy for managing dynamic feature sets.

The parameter choices, including the buffer size and step sizes, are judiciously selected to optimize performance and computational efficiency. The study's comprehensive evaluation and statistical analyses provide strong support for the validity and effectiveness of the OPFES method. The positive outcomes and clear methodologies contribute significantly to advancing the field, demonstrating a well-executed experimental design.

**Methods And Evaluation Criteria:**

The proposed methods and evaluation criteria are well-suited for feature evolvable learning. The Kernel Ortho-Mapping (KOM) discrepancy effectively characterizes relationships between evolving feature spaces, and the one-pass learning algorithm is efficient for streaming data. The evaluation uses diverse benchmark datasets and cumulative error rate (CER), ensuring relevance and comparability to prior work. Comparisons with state-of-the-art methods further validate the approach. Overall, the methodology and evaluation align well with the problem setting.

**Other Comments Or Suggestions:**

The notation in part “Previous Model Reuse” could be simplified for better readability. Besides, it would be better if the implementation detail of compared methods are presented in Appendix.

**Other Strengths And Weaknesses:**

Strengths:
1.The concept of kernel ortho-mapping discrepancy introduced in this paper is novel. Unlike most existing methods that rely on static feature spaces, which struggle to handle the evolving nature of feature-adaptive data streams, the proposed approach effectively captures information across different stages. By leveraging ortho-mapping discrepancy with kernel functions, the authors establish relationships between distinct feature spaces, enabling more efficient and effective learning.
2.The theoretical framework of this paper is well-developed and rigorous. They present the convergence result of kernel ortho-mapping discrepancy and utilize it to the regret analysis of the feature evolvable stream setting. The final theorem reveals the key factors influencing model convergence in this setting, which are consistent with empirical observations.
3.The proposed one-pass learning framework is well-designed, and the integration of online kernel learning with random Fourier features ensures both memory and computational efficiency. This approach enables the model to process large-scale datasets in a streaming manner without the need to store the full or partial training data. Such a design is particularly valuable for real-world applications that involve continuous data streams, making it highly practical and scalable.

Weaknesses:
1. The scenario of the OPFES algorithm can be applied is a little bit limited. The feature space can only change once during the whole learning process.
2. This article only considers hinge loss without analyzing other popular loss functions. The theoretical analysis is based on hinge loss, and the online learning process focuses on hinge loss as well. This article could be improved if more types of loss functions can be added to this algorithm.

**Questions For Authors:**

1.Are the methods in this paper applicable to general kernel functions? The methods in this paper are founded on shift-invariant kernels. I'm especially interested in whether they can be applied to general kernels.
2.Why does the article restrict the loss function to hinge loss only? What effects will it have on the theorems in this article if we substitute the loss function with MSE, logistic loss or exponential loss? Will the learning procedure in the OPFES algorithm still be effective for other loss functions?

**Relation To Broader Scientific Literature:**

The paper's key contributions are deeply intertwined with the broader scientific literature on feature evolvable learning, focusing on efficient adaptation to dynamic feature spaces in streaming data. By introducing the Kernel Ortho-Mapping (KOM) discrepancy, the study builds on prior work in kernel alignment and extends the understanding of feature relationships. The development of a one-pass algorithm aligns with the need for efficient online learning methods, leveraging random Fourier features for scalable kernel approximation. The approach integrates existing frameworks for adapting to evolving features, offering a theoretically grounded and practically applicable method that addresses the computational and adaptive challenges in dynamic feature spaces. The paper's contributions advance the field by providing a novel and comprehensive solution to the challenges of feature evolvable learning.

**Theoretical Claims:**

In the review process, I examined several key proofs in the paper to ensure their correctness. The proofs of Lemmas 3.2 and 3.5 are mathematically rigorous, leveraging properties of Frobenius norms and matrix decompositions effectively. The proof of Theorem 3.3 is sound, demonstrating the relationship between the KOM discrepancy and the gap between optimal classifiers, using empirical kernel mappings and linearization techniques.

However, the proof of Theorem 3.4, which involves probabilistic bounds, requires careful consideration of concentration inequalities and random matrix theory. While the steps appear logical, the complexity of the proof necessitates a thorough verification of the application of McDiarmid's inequality and operator Khintchine's inequality to ensure the bounds are correctly derived. Overall, the proofs are generally well-structured, but the probabilistic aspects could benefit from additional clarity to fully confirm their correctness.

---

> ### Author Rebuttal · Authors · 2025-04-01
>
> [Q1] The scenario of the OPFES algorithm that can be applied is a little bit limited. The feature space can only change once during the whole learning process.
>
> [A1] We will clarify that this work focuses on one feature evolution as in [Hou et al., 2021], and we can consider multiple feature evolutions similarly, that is, we learn feature relationships one by one as for evolving feature space, and maintain a model for every feature space via online ensemble.
>
> [Q2] This article only considers hinge loss without analyzing other loss functions...What effect will it have on the theorems in this article if we substitute the loss function with MSE, logistic loss, or exponential loss …
>
> [A2] We will clarify that this work focuses on one-pass learning with hinge loss, motivated from [Shalev-Shwartz et al., 2011, Lu et al., 2016], and it is feasible to generalize to other convex and Lipschitz-continuous loss functions, such as MSE, logistic loss and exponential loss, in which we need to take different Lipschitz constant and Fourier online gradient descent w.r.t. other loss functions.
>
> [Q3] Are the methods in this paper applicable to general kernel functions? The methods in this paper are found on shift-invariant kernels. I'm especially interested in whether they can be applied to general kernels.
>
> [A3] We will clarify that this work considers the shift-invariant kernels such as Gaussian kernel, which has been commonly-used in online kernel learning [Kivinen et al., 2001; Orabona et al., 2008; Takizawa et al., 2019; Ghari et al., 2022], and it is interesting to generalize to other kernels by considering similar approximation with random feature techniques.

---

### Official Review · Reviewer_XL9p · 2025-03-18

**Overall Recommendation:** 1

**Summary:**

This work proposes "One-Pass Feature Evolvable Learning" (i.e. OPFES), this is a method for handling streaming data where old features vanish and new features emerge. The core contribution is the kernel ortho-mapping discrepancy as $E\left(S\_n, K^{(1)}, K^{(2)}\right)=\min \_{U \in \mathrm{U}\_n} \frac{1}{\sqrt{n}}\left\\|U \sqrt{K^{(1)}}-\sqrt{K^{(2)}}\right\\|\_F$, which quantifies the transformation between two feature spaces via kernel embeddings. Using this discrepancy, the OPFES adaptively learns a new feature space in a single pass without retaining past data, leveraging random Fourier feature approximations and mirror descent optimization. The authors establish regret bounds, proving that classifier discrepancy $\rho\left(h_*^{(1)}, h_*^{(2)}\right)$ is upper-bounded by $O\left(E\left(S_n, K^{(1)}, K^{(2)}\right) / \lambda\right)$. Empirically, OPFES outperforms kernel-alignment and $\ell_2$-based baselines in binary classification tasks.

**Claims And Evidence:**

see Strengths And Weaknesses

**Essential References Not Discussed:**

n/a

**Experimental Designs Or Analyses:**

The experiments (benchmarks, analysis, ...) make sense and be supportive to me.

**Methods And Evaluation Criteria:**

see Strengths And Weaknesses

**Other Comments Or Suggestions:**

- line 667: shouldn't the $\partial(\cdot)$ represent subdifferentials? because the $\boldsymbol{g}$ in your KKT condition belongs to a set

- do your $c_1$, $c_2$ in Theorem A.4 share the same meaning as in Theorem 3.4, 4.3? similarly, the $c$ in Definition C.5, Lemma C.6, and proof of Lemma D.6

- Theorem A.15, $\lambda$ is the parameter from Eqn. (20)

- proof of Lemma A.17, why we need the convexity of $H$? we could simply verify that $H(x)/x$ is a nondecreasing function when $x\geq 0$

- Lemma C.4 should be $\hat{R}\_1\left(\boldsymbol{w}\_{1 *}\right)-\hat{R}\_2\left(\boldsymbol{w}\_{2 *}\right)\leq...$

- line 1458, what is the definition of $m_1$?

- line 1545: two empirical estimates

**Other Strengths And Weaknesses:**

- The KOM discrepancy can be understood as a measure of the difference between two sets of kernel mappings (or Gram matrices). Conceptually, it is closely related to existing methods, such as the well-known Orthogonal Procrustes Problem. Therefore, it does not constitute a novel contribution. The paper claims that KOM is superior to kernel alignment and ell_2 distance, but the comparison is only empirical, there's no real theoretical argument showing why this discrepancy should be preferred.

-  The min-max formulation over unitary transformations $U$ is mathematically elegant, but practically, why enforce rotational constraints in the first place? Many feature transformations in real-world apps are not merely orthogonal rotations. what about affine transformations? nonlinear shifts? Meanwhile the authors justify their empirical kernel mapping with methods from (Schölkopf & Smola 2002), but the assumption that two kernel feature spaces should be mapped in an orthogonal-preserving way is tenuous at best. Kernel alignment methods are widely used in community because they are more flexible, while kom enforces an artificial constraint that has no strong empirical motivation.


- I also think the connection between KOM discrepancy and classifier performance is weak; theorem 3.3 states that the difference between optimal classifiers in different feature spaces is upper-bounded by the KOM discrepancy. However, this is trivial, as any well-defined measure of feature space similarity will yield some bound on classifier discrepancy. The actual bound itself seems loose, i.e. there’s no guarantee that minimizing KOM results in an `actually good' model.

- Lastly, the theoretical results in paper are only valid for a very narrow setting. A vast number of feature evolvable problems (like in paper section 1) are regression tasks, yet the proposed algorithm is useless in these scenarios. And the vast majority of real-world scenarios where feature spaces change involve more than two classes. It’s also unclear if the results hold for other functions like cross-entropy or other margin-based losses e.g., logistic regression.

**Questions For Authors:**

- Is that possible to extend OPFES to multiclass classification? is your KOM discrepancy even meaningful for non-SVM loss functions, like kernelized ridge?

- You claim that storing past data is infeasible, but many real-world applications do store partial data in buffers. Could you comment on, why not allow a sliding window approach that keeps recent data, instead of forcing a single-pass update?


- How does OPFES perform with different kernel choices, does it collapse if the kernel width is set incorrectly? In Figure 6 why is the CER not monotonically decreasing as the random Fourier feature dimensionality increases?

- All results rely on shift-invariant kernels (like gaussian, laplacian), which is somehow a strong restriction as many real-world kernels (e.g., polynomial, string kernels, graph kernels) do not share that property. Could the key claims potentially extend to these settings, and if not, what is the difficulty?

**Relation To Broader Scientific Literature:**

This work is a follow-up in the field spanned by e.g. [1-3]

[1] Zhang et al., ICML 2020: “Learning with Feature and Distribution Evolvable Streams”​

[2] Hou et al., AAAI 2021: “Storage Fit Learning with Feature Evolvable Streams”​

[3] Schreckenberger et al., AAAI 2023: “Online Random Feature Forests for Learning in Varying Feature Spaces”

**Theoretical Claims:**

The proofs in this manuscript are somewhat sloppy. Despite spending considerable time analyzing them, I find it difficult to fully verify their correctness. The authors frequently provide detailed proofs for simple results while glossing over more complex derivations.

---

> ### Author Rebuttal · Authors · 2025-04-01
>
> [Q1] The KOM discrepancy can be understood as a measure of the difference between two sets of kernel mappings…Orthogonal Procrustes Problem ... KOM is superior to kernel alignment and $\ell_2$ distance…no real theoretical argument …
>
> [A1] We will clarify that the KOM discrepancy presents the first general framework to characterize correlations between two features spaces, and previous feature evolvable methods can be viewed as some special selections of different kernels [Hou et al. 2017, 2022; Chen&Liu 2024]. This discrepancy is motivated by orthogonal procrustes problem [Gover&Dijksterhuis 2004], which originated in matrix approximation problem in linear algebra, irrelevant to kernel functions.
>
> We will also clarify that Lemma 3.6 verifies the superiority of our KOM discrepancy in contrast to $\ell_2$ distance, and Lemma 3.5 presents the relationship between the KOM discrepancy and kernel alignment. Those theoretical results are verified empirically in Figure 3 and Table 2.
>
> [Q2] The min-max formulation over unitary transformations U… why enforce rotational constraints in the first place…affine transformations? nonlinear shifts...
>
> [A2] We will clarify that enforcing orthogonal rotations aims to guarantee closed-form solution for KOM discrepancy (Lemma 3.2) and Gram matrix w.r.t kernel functions in feature evolvable learning. Such formulation shows better characterizations between feature spaces than prior feature evolvable methods, and it is interesting to exploit other transformations.
>
> [Q3] …theorem 3.3 states that the difference between optimal classifiers in different feature spaces is upper-bounded by the KOM discrepancy…some bound on classifier discrepancy…seems loose...
>
> [A3] We will clarify that Theorem 3.3 presents the first theoretical result to correlate performance of classifiers with relationships (KOM discrepancy) between feature spaces, and we do not find tighter bounds before for feature evolvable learning. The proof includes linearization of kernel classifier via empirical kernel mapping, perturbation analysis of strongly convex loss and construction of KOM discrepancy.
>
> [Q4] …theoretical results in paper are only valid for a very narrow setting. A vast number of feature evolvable problems (…) are regression tasks…for other functions like cross-entropy or…
>
> [A4] We will clarify that our theoretical results focuses on binary classifications, since mostly feature evolvable learning considers classification tasks [Hou et al.,2017; Zhang et al., 2020; Hou et al., 2021; Lian et al., 2022; Schreckenberger et al., 2023]. Our results can be generalized to regression and multi-class tasks with other functions, by considering online gradient descent with random features for other functions and generalizing ideal kernel for regression and multiclass tasks as in [Wang et al., 2015].
>
> [Q5] Is that possible to extend OPFES to multiclass classification…non-SVM loss functions…
> [A5] We will clarify that our OPFES can be extended for multiclass classification by considering multiclass loss functions and ideal kernel as in [Wang et al., 2015]. We will also clarify that our KOM discrepancy is defined on Gram matrices of kernel functions, which can be used for non-SVM loss functions since it is irrelevant to loss functions and learning tasks.
>
> [Q6] …storing past data is infeasible, but many real-world applications do store partial data in buffers…why not allow a sliding window approach that keeps recent data…
>
> [A6] We will clarify that this work focuses on one-pass learning for large-scale data, which goes through all instances only once without storing training data as online learning [Crammer et al., 2005; Carvalho & Cohen, 2006; Cesa-Bianchi & Lugosi, 2006; Gao et al., 2013]. It is feasible to store partial data in buffers with some sliding window, which is called min-batch learning. Generally, min-batch learning requires more storage and computational costs than one-pass learning.
>
> [Q7] How does OPFES perform with different kernel choices…kernel width…In Figure 6 why is the CER not monotonically decreasing as…
>
> [A7] We will add a figure to show that OPFES achieves stable performance over a range of kernel widths in $[2^{-8}, 2^8]$. We will also clarify that, in Figure 6, most CERs are monotonically decreasing as the random Fourier feature dimensionality increases, except for four datasets, where the algorithm is slightly overfitting due to relatively fewer features and algorithmic randomness.
>
> [Q8] All results rely on shift-invariant kernels (like Gaussian, Laplacian)…a strong restriction as many real-world kernels (e.g., polynomial, string kernels, graph kernels)…
>
> [A8] We will clarify that this work focuses on the shift-invariant kernels such as Gaussian kernel, which has been commonly-used in online kernel learning [Kivinen et al., 2001; Orabona et al., 2008; Hong et al., 2023], and we make similar analysis for other kernels functions (polynomial and graph kernels) by random feature approximation.

---

> > ### Comment · Reviewer_XL9p · 2025-04-05
> >
> > Thank the authors for their detailed response. I agree with the replies to Q6 and Q8, those point do fall outside the scope of the paper. Unfortunately, after reading all rebuttal blocks, my main concern about this work remain unresolved. For example
> >
> > 1. I did notice Lemmas 3.5 and 3.6, but they appear to provide only a theoretical connection, rather than a rigorous mathematical argument demonstrating that KOM is superior to existing metrics. The numerical experiments are similarly unconvincing, they obviously cannot cover most scenarios and are susceptible to cherry-picking. The response to Q2/Q3 also felt rather tenuous to me.
> >
> > 2. The scope of the theoretical analysis is extremely narrow, which limited to binary classification using hinge loss and shift-invariant kernels. While authors claim generality, they also concede that extending to multiclass, regression, or other learning tasks (such as clustering or semi-supervised learning) would require substantial modification. This restriction to binary SVMs renders the proposed method largely irrelevant for the vast majority of modern applications. The rebuttal’s assertion that “our method can be extended” is unsupported by any concrete extensions or results in multiclass or regression settings.
> >
> > 3. I have to say that the theoretical portion of the paper is quite sloppy (see examples in review). Additionally, for line 1553,
> > > We completes the proof by applying Lemma C.3 and some calculations
> >
> >  the authors write in an incomplete manner. It's hard to believe that these derivations can be easily completed in just several steps. This make it difficult for the reader to verify the correctness of work.

---

> > > ### Author Response · Authors · 2025-04-09
> > >
> > > [Q1] … Lemmas 3.5 and 3.6, but they appear to provide only a theoretical connection rather than a rigorous mathematical argument demonstrating that KOM is superior to existing metrics. The numerical experiments…cherry-picking.
> > >
> > > [A1] We will clarify that this work proposes new KOM discrepancy to characterize correlations between two feature spaces, and previous methods can be viewed as some special selections of different kernels [Hou et al., 2021; Chen et al., 2024]. Lemma 3.6 shows that KOM discrepancy is a lower bound of $\ell_2$ distance, and minimizing KOM discrepancy is better than minimizing $\ell_2$ distance theoretically by combining with Theorem 3.3. Lemma 3.5 presents the relationship between the KOM discrepancy and kernel alignment; it is not easy to compare them directly. Therefore, we present an empirical study to show the effectiveness of KOM discrepancy in Figure 3 and Table 2.
> > >
> > > We will also clarify that most datasets in this work have been well-studied in previous feature evolvable learning [Hou et al., 2017; Zhang et al., 2020] and we take the same setting for fair comparisons.
> > >
> > > [Q2] The scope of the theoretical analysis is extremely narrow, which is limited to binary classification using hinge loss and shift-invariant kernels...
> > >
> > > [A2] We will clarify that, for feature evolving learning, this is the first work to characterize relationships between two feature spaces with theoretical guarantees, and it is natural to focus on binary classification, as in most theoretical studies such as PAC learning [Valiant, 1984], online learning [Rosenblatt, 1958; Aizerman et al., 1964; Cesa-Bianchi & Lugosi, 2006].
> > >
> > > We will clarify that this work focuses on hinge loss since it is the most popular loss function for batch and online SVMs [Cortes & Vapnik, 1995; Shalev-Shwartz 2008; Duchi & Hazan, 2011; Hajewski et al., 2018; Gentinetta et al., 2023], and our work is essentially an online SVM but with random fourier features.
> > >
> > > [Q3] …the theoretical portion of the paper is quite sloppy…for line 1553, We completes the proof by applying Lemma C.3 and some calculations…write in an incomplete manner…
> > >
> > > [A3] We will clarify that, for line 1553, we have Eqn. (24) from Lemma C.3 and strong convexity, and derive the upper bound for $\hat{R}_{T_2} (\pmb{w}\_{T_1+T_e}^{[2]}) - \hat R\_{T_2}(\pmb w\_{T_2*}^{[2]})$ as in lines 1514-1552. This completes the proofs by substituting the upper-bound into the right-hand side of Eqn. (24). We can definitely guarantee the correctness of our theoretical results and improve the presentations in proofs with more details.

---

### Official Review · Reviewer_AX8j · 2025-03-20

**Overall Recommendation:** 3

**Summary:**

The paper focuses on "feature evolvable learning" -- a setting in which features are being learned during a data stream, with new features learnt over time. The goal is not to retrain the model from scratch but to transfer from an older feature space to a new feature space. Similar problems have been studied in the past in the context of online continual learning.
The paper proposes a new measure (KOM) to quantify/characterize differences between feature spaces -- using kernel functions.
It also proposes an algorithm called OPFES to learn features in this evolvable manner without storing all training data.

**Claims And Evidence:**

The main claim of the paper can be stated as: it is possible to do feature evolvable learning in a streaming manner (one-pass over the data)  using a new metric called Kernel OrthoMapping (KOM) discrepancy, which captures the relationship between old and new feature spaces.

The paper provides sufficient evidence for this claim, through theoretical results, algorithm development (OPFES) and several experimental results.

**Essential References Not Discussed:**

I think that the paper is missing some references from the continual learning literature -- in the context of deep learning. Some of those papers also consider this stream-based setup -- even though the do not use the term "feature evolvable learning". Instead they use terms such as "online continual learning" or "unsupervised online continual learning".

**Experimental Designs Or Analyses:**

The paper is quite thorough and systematic in its experimental study. The main result in that portion of the paper is to show that the OPFES outperforms other feature evolvable learn methods (such as random fourier-based or kernel-based) on many different datasets.

**Methods And Evaluation Criteria:**

The paper is strong methodologically, using mathematical analysis when it is appropriate to do so, designing systematic experiments, and being thorough in how it states its claims.

**Other Comments Or Suggestions:**

One possible improvement, mostly in terms of presentation, is to somehow explain the insights and steps of the OPFES algorithm a bit  more -- maybe through a visual illustration.

**Other Strengths And Weaknesses:**

One main weakness of the paper is that it does not leverage the power of deep learning. As the authors also mention at the very end, a follow-up work could consider the Neural Tangent Kernel framework that has been developed for the approximate analysis of neural networks -- to examine how the KOM approach can be applied there.

**Questions For Authors:**

how does OPFES deal with partial feature overlap between tthe old an new feature spaces? For example, some features may persist while others disappear?

**Relation To Broader Scientific Literature:**

The paper also relates to the broader literature of learning theory as well as to continual learning.

**Theoretical Claims:**

In my opinion, the main theoretical results of the paper are:
1) Theorem 3.3 giving an upper bound (in terms of KOM) for the differences between optimal classifiers trained on old vs new feature spaces
2) Theorem 3.4, which gives a convergence result, showing how the empirical KOM discrepancy approximates its distributional counterpart.

I admit that I have not checked the proofs of the theoretical claims -- as this was one of urgent reviews I had to provide.

---

> ### Author Rebuttal · Authors · 2025-04-01
>
> [Q1] One main weakness of the paper is that it does not leverage the power of deep learning … a follow-up work could consider the Neural Tangent Kernel framework that has been developed for the approximate analysis of neural networks…
>
> [A1] We will clarify that this work tries to answer two fundamental problems on feature evolvable learning following previous settings [Hou et al., 2021; Chen et al., 2024], and it is interesting to leverage the power of deep learning for further work, where we could incorporate deep learning technique via random features of Neural Tangent Kernel [Zandieh et al., 2021], and distinguish representations from two network structures [Kornblith et al., 2019].
>
> [Q2] One possible improvement, mostly in terms of presentation, is to somehow explain the insights and steps of the OPFES algorithm a bit more— maybe through a visual illustration.
>
> [A2] We will consider a visual flowchart to outline the main (previous, evolving, and current) stages of the OPFES algorithm and explain how feature information and label information are incorporated via our KOM discrepancy and how the model is updated in a one-pass manner using random Fourier features.
>
> [Q3] How does OPFES deal with partial feature overlap between the old and new feature spaces? For example, some features may persist while others disappear.
>
> [A3] We will clarify that this work focuses on the non-overlap between old and new feature spaces, following previous feature evolvable learning [Hou et al., 2017; Zhou et al., 2022], and it is interesting to consider some partial feature overlaps, for which we could learn two relevant models by exploiting correlation from two feature spaces as in [Hou et al., 2016; Ye et al., 2018; Gu et al., 2024].
>
> We will add relevant references on online and unsupervised continual learning [Lange et al., 2021; Wickramasinghe et al., 2023; Wang et al., 2024].

---

### Official Review · Reviewer_UGGx · 2025-04-01

**Overall Recommendation:** 3

**Summary:**

This paper tackles the problem of feature evolvable learning in streaming data settings, where features may vanish and emerge over time—a scenario that arises in applications like sensor networks or dynamic monitoring systems. The authors propose a new metric, the Kernel Ortho-Mapping (KOM) discrepancy, which quantitatively characterizes the relationship between two evolving feature spaces via kernel functions.

They then develop a one-pass online learning algorithm, called OPFES, that:

Leverages random Fourier features for efficient kernel approximation,

Integrates prior model knowledge via KOM-based mappings,

Reuses both feature and label relationships from the previous feature space without retaining training data.

The theoretical contributions include:

A bound between KOM discrepancy and the difference in predictions from classifiers trained in two distinct feature spaces (Theorem 3.3),

Generalization error bounds (Theorem 4.3),

Regret guarantees for their mirror descent-based optimization procedure (Theorem 3.7).

Empirically, OPFES consistently outperforms a suite of state-of-the-art methods across 20 real-world datasets, demonstrating both lower cumulative error rates and faster convergence.

**Claims And Evidence:**

The primary claims are:

KOM discrepancy better captures relationships between evolving feature spaces than kernel alignment or ℓ₂-distance.

OPFES achieves state-of-the-art performance in both accuracy and convergence speed.

Theoretical convergence and regret bounds are valid under the assumed setting.

Evidence:

The KOM discrepancy is supported by both a theoretical upper bound (Theorem 3.3) and empirical correlation analyses (Figures 2 and 3).

Experimental results across 20 datasets (Table 2, Figures 4–5) substantiate the superior performance of OPFES.

The convergence of their optimization strategy is justified through mirror descent analysis (Theorem 3.7), and prediction deviation is bounded through KOM discrepancy (Lemma 4.2).

Evaluation: Claims are well-supported with theoretical derivations and extensive experiments. There are no overtly problematic claims, though practical scenarios with high noise or extremely non-stationary distributions could be addressed more thoroughly.

**Essential References Not Discussed:**

The paper is generally well-cited. However:

It omits Rahimi and Recht (2007), who formally introduced Random Fourier Features, although Bochner’s theorem is referenced.

More recent works on deep kernel transfer learning or meta-learning over kernel spaces could be relevant for contextual breadth, especially if extensions to deep networks are envisioned.

In the context of evolving data streams, work on concept drift detection (e.g., Gama et al., 2014) might be conceptually adjacent and worth mentioning.

**Ethical Review Concerns:**

Not applicable.

**Experimental Designs Or Analyses:**

The experiments are well-designed:

20 datasets from OpenML/UCI.

Comparisons with 8 strong baselines (e.g., rff-FESL, align-FESL).

Multiple performance metrics: CER, convergence speed, sensitivity to Fourier dimension.

Statistical significance tests (t-tests) confirm robustness.

The setup (T1, Te, T2 split; kernel and hyperparameter tuning) is explicitly stated. However, more insight into variance across random splits or ablation of individual components (e.g., only using KOM without label alignment) could enhance the experimental depth.

**Methods And Evaluation Criteria:**

The authors use appropriate methods:

Random Fourier Features for scalable kernel learning,

KOM discrepancy for measuring evolution of feature spaces,

Mirror descent on the simplex for optimizing spectral density.

Evaluation criteria—Cumulative Error Rate (CER) and convergence speed—are suitable for online learning benchmarks. The use of diverse datasets and baselines strengthens the empirical evaluation. Te-splitting strategy and buffering policies are transparent and consistent with prior work.

**Other Comments Or Suggestions:**

The pseudocode could benefit from a clearer distinction between training phase and transfer phase.
Appendix A, Lemma Proofs:
The paper shall consider including (in main or appendix) sensitivity plots for learning rate, gamma1, and gamma2. The Bayesian optimization results could be volatile depending on the dataset.

**Other Strengths And Weaknesses:**

Strong Theoretical Proofs - The integration of kernel theory, generalization analysis, and convex optimization underpins the theoretical soundness. Convergence, generalization, and transferability are all rigorously treated.

Empirical Breadth and Depth - Evaluation across 20 diverse datasets, paired with statistical testing and convergence analysis, validates the model’s robustness and practical effectiveness.

Interpretability via Kernel Metrics-  KOM provides not just an operational tool but a theoretical lens for understanding how and why models transfer across evolving feature spaces.

Weakness :
No Analysis for Adversarial or Disjoint Feature Evolutions - While the KOM discrepancy is effective under smooth transitions, its applicability under non-overlapping feature spaces or adversarial drift remains unexplored. For example, when new features are independent of the old, KOM may offer no meaningful signal, yet the algorithm is still applied.

Unclear Role of Label-Based Ideal Kernel Under Imbalance- The paper proposes using an ideal kernel 𝐾∗(𝑥𝑖,𝑥𝑗)=𝑦𝑖𝑦𝑗    to align labels in the evolving stage. But, This approach assumes balanced label distributions, which do not hold for several datasets (e.g., acoustic, runwalk). The approach is sensitive to noisy or ambiguous labels, potentially destabilizing kernel alignment.


Scalability Beyond Fourier Features- While random Fourier features are scalable, they are limited for high-dimensional structured domains (e.g., graphs, sequences). Extension to non-stationary kernels or deep kernel surrogates is a natural direction not yet addressed. (Perhaps opened up for future direction)

**Questions For Authors:**

How does the OPFES framework behave in scenarios where the new feature space is statistically independent from the old one (i.e., KOM discrepancy is large or near-maximal)?

Given that the ideal label kernel 𝐾∗(𝑥𝑖,𝑥𝑗)=𝑦𝑖𝑦𝑗   assumes balanced and clean labels, how does your method adapt when labels are noisy or highly imbalanced (e.g., acoustic, runwalk datasets)?

Can the authors provide ablation studies separating the contributions of (i) KOM discrepancy, (ii) label kernel alignment, and (iii) model reuse from previous stages?

What is the runtime overhead (in wall-clock time and computational complexity) of the mirror descent procedure used to optimize KOM in comparison to other ensemble or retraining approaches?

Does your method generalize to non-shift-invariant kernels or structured inputs (e.g., graph, sequence kernels)?

**Relation To Broader Scientific Literature:**

The paper builds on a rich literature in:

Online kernel learning (Lu et al., 2016; Shen et al., 2019),

Feature evolvable streams (Hou et al., 2017; 2021; 2022),

Kernel alignment (Cristianini et al., 2001),

ℓ₂-mapping in transfer learning (Romero et al., 2015).

It adds a new formal metric (KOM) that generalizes and theoretically bounds earlier methods (alignment, ℓ₂-distance). The approach synthesizes kernel theory with online convex optimization, contributing novel algorithmic and theoretical insights.

**Theoretical Claims:**

Several key theoretical results are presented:

Theorem 3.3: Bounded classifier divergence via KOM discrepancy.

Theorem 3.4: Generalization guarantee for empirical KOM discrepancy.

Lemma 3.5, 3.6: Relationship of KOM discrepancy with kernel alignment and ℓ₂-distance.

Theorem 3.7: Mirror descent convergence rate for KOM optimization.

Theorem 4.3: Generalization bound for OPFES.

These are grounded in solid theoretical tools—e.g., McDiarmid’s inequality, matrix norm inequalities, polar decomposition, and Rademacher complexity. The proofs (referenced in appendices) appear consistent and technically correct from inspection of their assumptions and derivations.



Theorem 3.3: Bounded Classifier Divergence via KOM Discrepancy - The derivation is consistent with prior work on RKHS norm bounds and empirical classifier discrepancy measures.

Theorem 3.4: Generalization of Empirical KOM Discrepancy  - The proof also incorporates non-commutative Khintchine-type inequalities, which are non-trivial but correctly referenced.

Lemma 3.5: KOM discrepancy is upper bounded by a function of 1 minus the kernel alignment. - Lemma 3.5 leverages normalized Gram matrices and a cosine similarity-based interpretation.

Lemma 3.6: KOM discrepancy is also upper bounded by the average ℓ₂ deviation between kernel mappings. - Lemma 3.6 mimics regularization path analyses and is essentially a generalization of kernel regression upper bounds.


Theorem 4.3: Generalization Bound for OPFES
Claim: Provides an excess risk bound for the OPFES predictor w.r.t. the optimal classifier in the current stage, incorporating the KOM discrepancy.

Validation:

Combines regret analysis (Hazan et al., 2016) with generalization error bounds using Rademacher complexity.

Proper use of online-to-batch conversion techniques is evident.

The bound reflects interplay between past information (via T1 ,Te ) and the KOM discrepancy, offering useful theoretical insights.

---

### Decision · Program_Chairs · 2025-05-01

**Decision:**

Accept (poster)

**Comment:**

In this paper, the authors introduce a Kernel Ortho-Mapping discrepancy to characterize relationships between two different feature spaces via kernel functions and correlate it with the optimal classifiers learned from different feature spaces. Based on this discrepancy, the authors develop a one-pass algorithm for feature evolvable learning. They provide both theoretical analysis and experimental results in this paper.

After the rebuttal period, four of five reviewers are positive towards this paper. While Reviewer XL9p kept their negative rating. After reading the paper and discussions, I think this is overall an interesting paper. However, I partially agree with Reviewer XL9p. For example, Reviewer XL9p pointed out, "the scope of the theoretical analysis is narrow and limited to binary classification using hinge loss and shift-invariant kernels" and "lacking supporting evidence that their method can be extended." The authors' response is not enough. They should adjust their claim and be careful about their conclusions. Considering all of these, I recommend "weak accept."